# GENIE: Higher-Order Denoising Diffusion Solvers

**Tim Dockhorn**[1,2,3,*]  **Arash Vahdat**[1]  **Karsten Kreis**[1]

[1]NVIDIA  [2]University of Waterloo  [3]Vector Institute

tim.dockhorn@uwaterloo.ca,  {avahdat,kkreis}@nvidia.com

## Abstract

Denoising diffusion models (DDMs) have emerged as a powerful class of generative models. A forward diffusion process slowly perturbs the data, while a deep model learns to gradually denoise. Synthesis amounts to solving a differential equation (DE) defined by the learnt model. Solving the DE requires slow iterative solvers for high-quality generation. In this work, we propose *Higher-Order Denoising Diffusion Solvers* (GENIE): Based on truncated Taylor methods, we derive a novel higher-order solver that significantly accelerates synthesis. Our solver relies on higher-order gradients of the perturbed data distribution, that is, higher-order score functions. In practice, only Jacobian-vector products (JVPs) are required and we propose to extract them from the first-order score network via automatic differentiation. We then distill the JVPs into a separate neural network that allows us to efficiently compute the necessary higher-order terms for our novel sampler during synthesis. We only need to train a small additional head on top of the first-order score network. We validate GENIE on multiple image generation benchmarks and demonstrate that GENIE outperforms all previous solvers. Unlike recent methods that fundamentally alter the generation process in DDMs, our GENIE solves the true generative DE and still enables applications such as encoding and guided sampling. Project page and code: https://nv-tlabs.github.io/GENIE.

## 1 Introduction

Denoising diffusion models (DDMs) offer both state-of-the-art synthesis quality and sample diversity in combination with a robust and scalable learning objective. DDMs have been used for image [1–5] and video [6, 7] synthesis, super-resolution [8, 9], deblurring [10, 11], image editing and inpainting [5, 12–14], text-to-image synthesis [15–17], conditional and semantic image generation [18–22], image-to-image translation [14, 23, 24] and for inverse problems in medical imaging [25–31]. They also enable high-quality speech synthesis [32–37], 3D shape generation [38–42], molecular modeling [43–46], maximum likelihood training [47–50], and more [51–56]. In DDMs, a diffusion process gradually perturbs the data towards random noise, while a deep neural network learns to denoise. Formally, the problem reduces to learning the *score function*, i.e., the gradient of the log-density of the perturbed data. The (approximate) inverse of the forward diffusion can be described by an ordinary or a stochastic differential equation (ODE or SDE, respectively), defined by the learned score function, and can therefore be used for generation when starting from random noise [47, 57].

A crucial drawback of DDMs is that the generative ODE or SDE is typically difficult to solve, due to the complex score function. Therefore, efficient and tailored samplers are required for fast synthesis. In this work, building on the generative ODE [47, 57, 58], we rigorously derive a novel second-order ODE solver using *truncated Taylor methods* [59]. These higher-order methods require higher-order gradients of the ODE—in our case this includes higher-order gradients of the

---

*Work done during internship at NVIDIA.

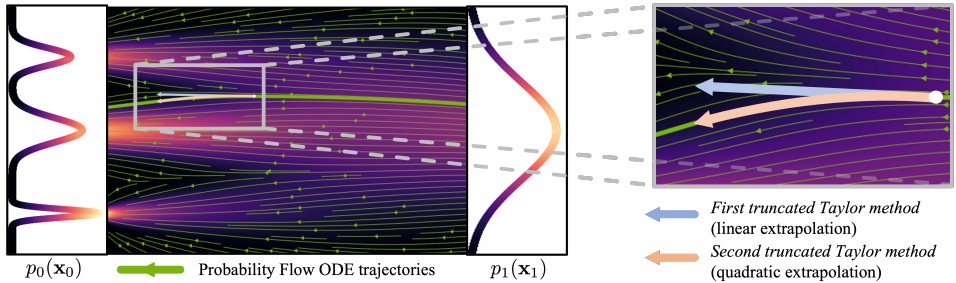

*First truncated Taylor method* (linear extrapolation)

*Second truncated Taylor method* (quadratic extrapolation)

$p_0(\mathbf{x}_0)$     ← Probability Flow ODE trajectories     $p_1(\mathbf{x}_1)$

Figure 1: Our novel *Higher-Order Denoising Diffusion Solver* (GENIE) relies on the second *truncated Taylor method* (TTM) to simulate a (re-parametrized) Probability Flow ODE for sampling from denoising diffusion models. The second TTM captures the local curvature of the ODE's gradient field and enables more accurate extrapolation and larger step sizes than the first TTM (Euler's method), which previous methods such as DDIM [58] utilize.

log-density of the perturbed data, i.e., higher-order score functions. Because such higher-order scores are usually not available, existing works typically use simple first-order solvers or samplers with low accuracy [1, 57, 58, 60], higher-order methods that rely on suboptimal finite difference or other approximations [61–63], or alternative approaches [64–66] for accelerated sampling. Here, we fundamentally avoid such approximations and directly model the higher-order gradient terms: Importantly, our novel *Higher-Order Denoising Diffusion Solver (GENIE)* relies on Jacobian-vector products (JVPs) involving second-order scores. We propose to calculate these JVPs by automatic differentiation of the regular learnt first-order scores. For computational efficiency, we then distill the entire higher-order gradient of the ODE, including the JVPs, into a separate neural network. In practice, we only need to add a small head to the first-order score network to predict the components of the higher-order ODE gradient. By directly modeling the JVPs we avoid explicitly forming high-dimensional higher-order scores. Intuitively, the higher-order terms in GENIE capture the local curvature of the ODE and enable larger steps when iteratively solving the generative ODE (Fig. 1).

Experimentally, we validate GENIE on multiple image modeling benchmarks and achieve state-of-the-art performance in solving the generative ODE of DDMs with few synthesis steps. In contrast to recent methods that fundamentally modify the generation process of DDMs by training conditional GANs [67] or by distilling the full sampling trajectory [68, 69], GENIE solves the true generative ODE. Therefore, we also show that we can still encode images in the DDM's latent space, as required for instance for image interpolation, and use techniques such as guided sampling [4, 57, 70].

We make the following contributions: **(i)** We introduce GENIE, a novel second-order ODE solver for fast DDM sampling. **(ii)** We propose to extract the required higher-order terms from the first-order score model by automatic differentiation. In contrast to existing works, we explicitly work with higher-order scores without finite difference approximations. To the best of our knowledge, GENIE is the first method that *explicitly* uses higher-order scores for generative modeling with DDMs. **(iii)** We propose to directly model the necessary JVPs and distill them into a small neural network. **(iv)** We outperform all previous solvers and samplers for the generative differential equations of DDMs.

## 2 Background

We consider continuous-time DDMs [1, 57, 71] whose forward process can be described by

$$p_t(\mathbf{x}_t|\mathbf{x}_0) = \mathcal{N}(\mathbf{x}_t; \alpha_t \mathbf{x}_0, \sigma_t^2 \boldsymbol{I}), \tag{1}$$

where $\mathbf{x}_0 \sim p_0(\mathbf{x}_0)$ is drawn from the empirical data distribution and $\mathbf{x}_t$ refers to diffused data samples at time $t \in [0, 1]$ along the diffusion process. The functions $\alpha_t$ and $\sigma_t$ are generally chosen such that the *logarithmic signal-to-noise ratio* [48] $\log \frac{\alpha_t^2}{\sigma_t^2}$ decreases monotonically with $t$ and the data diffuses towards random noise, i.e., $p_1(\mathbf{x}_1) \approx \mathcal{N}(\mathbf{x}_1; \mathbf{0}, \boldsymbol{I})$. We use *variance-preserving* [57] diffusion processes for which $\sigma_t^2 = 1 - \alpha_t^2$ (however, all methods introduced in this work are applicable to more general DDMs). The diffusion process can then be expressed by the (variance-preserving) SDE

$$d\mathbf{x}_t = -\tfrac{1}{2}\beta_t \mathbf{x}_t \, dt + \sqrt{\beta_t} \, d\mathbf{w}_t, \tag{2}$$

where $\beta_t = -\frac{d}{dt} \log \alpha_t^2$, $\mathbf{x}_0 \sim p_0(\mathbf{x}_0)$ and $\mathbf{w}_t$ is a standard Wiener process. A corresponding reverse diffusion process that effectively inverts the forward diffusion is given by [57, 72, 73]

$$d\mathbf{x}_t = -\tfrac{1}{2}\beta_t \left[\mathbf{x}_t + 2\nabla_{\mathbf{x}_t} \log p_t(\mathbf{x}_t)\right] dt + \sqrt{\beta_t} \, d\mathbf{w}_t, \tag{3}$$

and this reverse-time generative SDE is marginally equivalent to the generative ODE [47, 57]

$$d\mathbf{x}_t = -\tfrac{1}{2}\beta_t \left[\mathbf{x}_t + \nabla_{\mathbf{x}_t} \log p_t(\mathbf{x}_t)\right] dt, \tag{4}$$

where $\nabla_{\mathbf{x}_t} \log p_t(\mathbf{x}_t)$ is the *score function*. Eq. (4) is referred to as the *Probability Flow* ODE [57], an instance of continuous Normalizing flows [74, 75]. To generate samples from the DDM, one can sample $\mathbf{x}_1 \sim \mathcal{N}(\mathbf{x}_1; \mathbf{0}, \boldsymbol{I})$ and numerically simulate either the Probability Flow ODE or the generative SDE, replacing the unknown score function by a learned score model $\boldsymbol{s}_{\boldsymbol{\theta}}(\mathbf{x}_t, t) \approx \nabla_{\mathbf{x}_t} \log p_t(\mathbf{x}_t)$.

The DDIM solver [58] has been particularly popular to simulate DDMs due to its speed and simplicity. It has been shown that DDIM is *Euler's method* applied to an ODE based on a re-parameterization of the Probability Flow ODE [58, 69]: Defining $\gamma_t = \sqrt{\frac{1-\alpha_t^2}{\alpha_t^2}}$ and $\bar{\mathbf{x}}_t = \mathbf{x}_t \sqrt{1 + \gamma_t^2}$, we have

$$\frac{d\bar{\mathbf{x}}_t}{d\gamma_t} = \sqrt{1 + \gamma_t^2} \frac{d\mathbf{x}_t}{dt} \frac{dt}{d\gamma_t} + \mathbf{x}_t \frac{\gamma_t}{\sqrt{1 + \gamma_t^2}} = -\frac{\gamma_t}{\sqrt{1 + \gamma_t^2}} \nabla_{\mathbf{x}_t} \log p_t(\mathbf{x}_t), \tag{5}$$

where we inserted Eq. (4) for $\frac{d\mathbf{x}_t}{dt}$ and used $\beta(t) \frac{dt}{d\gamma_t} = \frac{2\gamma_t}{\gamma_t^2+1}$. Letting $\boldsymbol{s}_{\boldsymbol{\theta}}(\mathbf{x}_t, t) := -\frac{\boldsymbol{\epsilon}_{\boldsymbol{\theta}}(\mathbf{x}_t, t)}{\sigma_t}$ denote a parameterization of the score model, the approximate generative DDIM ODE is then given by

$$d\bar{\mathbf{x}}_t = \boldsymbol{\epsilon}_{\boldsymbol{\theta}}(\mathbf{x}_t, t) \, d\gamma_t, \tag{6}$$

where we used $\sigma_t = \sqrt{1 - \alpha_t^2} = \frac{\gamma_t}{\sqrt{\gamma_t^2+1}}$ (see App. A for a more detailed derivation of Eq. (6)). The model $\boldsymbol{\epsilon}_{\boldsymbol{\theta}}(\mathbf{x}_t, t)$ can be learned by minimizing the score matching objective [1, 76]

$$\min_{\boldsymbol{\theta}} \mathbb{E}_{t \sim \mathcal{U}[t_{\mathrm{cutoff}}, 1], \mathbf{x}_0 \sim p(\mathbf{x}_0), \boldsymbol{\epsilon} \sim \mathcal{N}(\mathbf{0}, \boldsymbol{I})} \left[ g(t) \| \boldsymbol{\epsilon} - \boldsymbol{\epsilon}_{\boldsymbol{\theta}}(\mathbf{x}_t, t) \|_2^2 \right], \quad \mathbf{x}_t = \alpha_t \mathbf{x}_0 + \sigma_t \boldsymbol{\epsilon}, \tag{7}$$

for small $0 < t_{\mathrm{cutoff}} \ll 1$. As is standard practice, we set $g(t) = 1$. Other weighting functions $g(t)$ are possible; for example, setting $g(t) = \frac{\beta_t}{2\sigma_t^2}$ recovers maximum likelihood learning [47–50].

## 3 Higher-Order Denoising Diffusion Solver

As discussed in Sec. 2, the so-known DDIM solver [58] is simply Euler's method applied to the DDIM ODE (cf. Eq. (6)). In this work, we apply a higher-order method to the DDIM ODE, building on the *truncated Taylor method* (TTM) [59]. The $p$-th TTM is simply the $p$-th order *Taylor polynomial* applied to an ODE. For example, for the general $\frac{d\mathbf{y}}{dt} = \boldsymbol{f}(\mathbf{y}, t)$, the $p$-th TTM reads as

$$\mathbf{y}_{t_{n+1}} = \mathbf{y}_{t_n} + h_n \frac{d\mathbf{y}}{dt} |_{(\mathbf{y}_{t_n}, t_n)} + \cdots + \frac{1}{p!} h_n^p \frac{d^p\mathbf{y}}{dt^p} |_{(\mathbf{y}_{t_n}, t_n)}, \tag{8}$$

where $h_n = t_{n+1} - t_n$ (see App. B.1 for a truncation error analysis with respect to the exact ODE solution). Note that the first TTM is simply Euler's method. Applying the second TTM to the DDIM ODE results in the following scheme:

$$\bar{\mathbf{x}}_{t_{n+1}} = \bar{\mathbf{x}}_{t_n} + h_n \boldsymbol{\epsilon}_{\boldsymbol{\theta}}(\mathbf{x}_{t_n}, t_n) + \frac{1}{2} h_n^2 \frac{d\boldsymbol{\epsilon}_{\boldsymbol{\theta}}}{d\gamma_t} |_{(\mathbf{x}_{t_n}, t_n)}, \tag{9}$$

where $h_n = \gamma_{t_{n+1}} - \gamma_{t_n}$. Recall that $\gamma_t = \sqrt{\frac{1-\alpha_t^2}{\alpha_t^2}}$, where the function $\alpha_t$ is a time-dependent hyperparameter of the DDM. The total derivative $d_{\gamma_t} \boldsymbol{\epsilon}_{\boldsymbol{\theta}} := \frac{d\boldsymbol{\epsilon}_{\boldsymbol{\theta}}}{d\gamma_t}$ can be decomposed as follows

$$d_{\gamma_t} \boldsymbol{\epsilon}_{\boldsymbol{\theta}}(\mathbf{x}_t, t) = \frac{\partial \boldsymbol{\epsilon}_{\boldsymbol{\theta}}(\mathbf{x}_t, t)}{\partial \mathbf{x}_t} \frac{d\mathbf{x}_t}{d\gamma_t} + \frac{\partial \boldsymbol{\epsilon}_{\boldsymbol{\theta}}(\mathbf{x}_t, t)}{\partial t} \frac{dt}{d\gamma_t}, \tag{10}$$

where $\frac{\partial \boldsymbol{\epsilon}_{\boldsymbol{\theta}}(\mathbf{x}_t, t)}{\partial \mathbf{x}_t}$ denotes the Jacobian of $\boldsymbol{\epsilon}_{\boldsymbol{\theta}}(\mathbf{x}_t, t)$ and

$$\frac{d\mathbf{x}_t}{d\gamma_t} = \frac{\partial \mathbf{x}_t}{\partial \bar{\mathbf{x}}_t} \frac{d\bar{\mathbf{x}}_t}{d\gamma_t} + \frac{\partial \mathbf{x}_t}{\partial \gamma_t} = \frac{1}{\sqrt{\gamma_t^2 + 1}} \boldsymbol{\epsilon}_{\boldsymbol{\theta}}(\mathbf{x}_t, t) - \frac{\gamma_t}{1 + \gamma_t^2} \mathbf{x}_t. \tag{11}$$

If not explicitly stated otherwise, we refer to the second TTM applied to the DDIM ODE, i.e., the scheme in Eq. (9), as *Higher-Order Denoising Diffusion Solver* (GENIE). Intuitively, the higher-order gradient terms used in the second TMM model the local curvature of the ODE. This translates into a Taylor formula-based extrapolation that is quadratic in time (cf. Eqs. (8) and (9)) and more accurate than linear extrapolation, as in Euler's method, thereby enabling larger time steps (see Fig. 1 for a visualization). In App. B, we also discuss the application of the third TTM to the DDIM ODE. We emphasize that TTMs are not restricted to the DDIM ODE and could just as well be applied to the Probability Flow ODE [57] (also see App. B) or neural ODEs [74, 75] more generally.

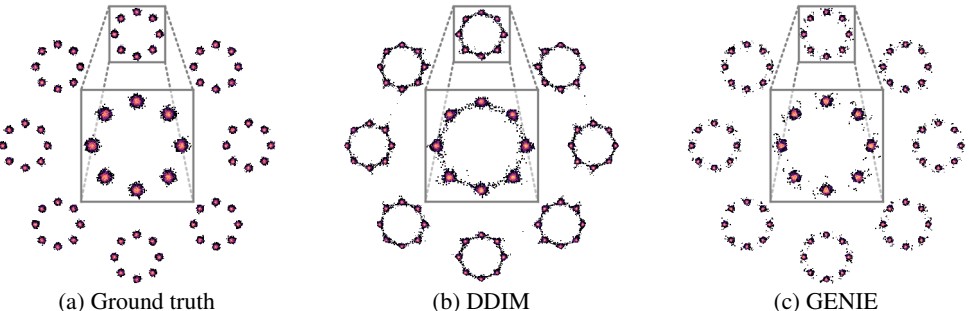

| (a) Ground truth | (b) DDIM | (c) GENIE |

Figure 2: Modeling a complex 2D toy distribution: Samples in *(b)* and *(c)* are generated via DDIM and GENIE, respectively, with 25 solver steps using the analytical score function of the ground truth distribution.

**The Benefit of Higher-Order Methods:** We showcase the benefit of higher-order methods on a 2D toy distribution (Fig. 2a) for which we know the score function as well as all higher-order derivatives necessary for GENIE analytically. We generate 1k different accurate "ground truth" trajectories $\mathbf{x}_t$ using DDIM with 10k steps. We compare these "ground truth" trajectories to *single* steps of DDIM and GENIE for varying step sizes $\Delta t$. We then measure the mean $L_2$-distance of the single steps $\hat{\mathbf{x}}_t(\Delta t)$ to the "ground truth" trajectories $\mathbf{x}_t$, and we repeat this experiment for three starting points $t \in \{0.1, 0.2, 0.5\}$. We see (Fig. 3 *(top)*) that GENIE can use larger step sizes to stay within a certain error tolerance for all starting points $t$. We further show samples for DDIM and GENIE, using 25 solver steps, in Fig. 2. DDIM has the undesired behavior of sampling low-density regions between modes, whereas GENIE looks like a slightly noisy version of the ground truth distribution (Fig. 2a).

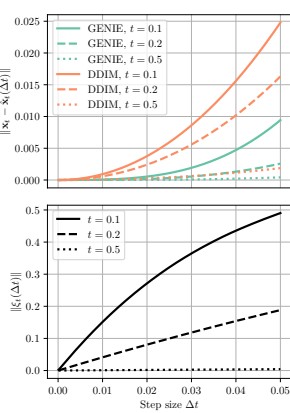

Figure 3: *Top*: Single step error using analytical score function. *Bottom*: Norm of difference $\xi_t(\Delta t)$ between analytical and approximate derivative computed via finite difference method.

**Comparison to Multistep Methods:** Linear multistep methods are an alternative higher-order method to solve ODEs. Liu et al. [63] applied the well-established Adams–Bashforth [AB, 77] method to the DDIM ODE. AB methods can be derived from TTMs by approximating higher-order derivatives $\frac{d^p \mathbf{y}}{dt^p}$ using the finite difference method [78]. For example, the second AB method is obtained from the second TTM by replacing $\frac{d^2 \mathbf{y}}{dt^2}$ with the first-order forward difference approximation $(f(\mathbf{y}_{t_n}, t_n) - f(\mathbf{y}_{t_{n-1}}, t_{n-1}))/h_{n-1}$. In Fig. 3 *(bottom)*, we visualize the mean $L_2$-norm of the difference $\xi_t(\Delta t)$ between the analytical derivative $d_{\gamma_t}\boldsymbol{\epsilon}_{\boldsymbol{\theta}}$ and its first-order forward difference approximation for varying step sizes $\Delta t$ for the 2D toy distribution. The approximation is especially poor at small $t$ for which the score function becomes complex (App. E for details on all toy experiments).

### 3.1 Learning Higher-Order Derivatives

The above observations inspire to apply GENIE to DDMs of more complex and high-dimensional data such as images. Regular DDMs learn a model $\boldsymbol{\epsilon}_{\boldsymbol{\theta}}$ for the first-order score; however, the higher-order gradient terms required for GENIE (cf. Eq. (10)) are not immediately available to us, unlike in the toy example above. Let us insert Eq. (11) into Eq. (10) and analyze the required terms more closely:

$$d_{\gamma_t}\boldsymbol{\epsilon}_{\boldsymbol{\theta}}(\mathbf{x}_t, t) = \frac{1}{\sqrt{\gamma_t^2 + 1}} \underbrace{\frac{\partial \boldsymbol{\epsilon}_{\boldsymbol{\theta}}(\mathbf{x}_t, t)}{\partial \mathbf{x}_t} \boldsymbol{\epsilon}_{\boldsymbol{\theta}}(\mathbf{x}_t, t)}_{\text{JVP}_1} - \frac{\gamma_t}{1 + \gamma_t^2} \underbrace{\frac{\partial \boldsymbol{\epsilon}_{\boldsymbol{\theta}}(\mathbf{x}_t, t)}{\partial \mathbf{x}_t} \mathbf{x}_t}_{\text{JVP}_2} + \frac{\partial \boldsymbol{\epsilon}_{\boldsymbol{\theta}}(\mathbf{x}_t, t)}{\partial t} \frac{dt}{d\gamma_t}. \quad (12)$$

We see that the full derivative decomposes into two JVP terms and one simpler time derivative term. The term $\frac{\partial \boldsymbol{\epsilon}_{\boldsymbol{\theta}}(\mathbf{x}_t, t)}{\partial \mathbf{x}_t}$ plays a crucial role in Eq. (12). It can be expressed as

$$\frac{\partial \boldsymbol{\epsilon}_{\boldsymbol{\theta}}(\mathbf{x}_t, t)}{\partial \mathbf{x}_t} = -\sigma_t \frac{\partial \mathbf{s}_{\boldsymbol{\theta}}(\mathbf{x}_t, t)}{\partial \mathbf{x}_t} \approx -\sigma_t \nabla_{\mathbf{x}_t}^{\top} \nabla_{\mathbf{x}_t} \log p_t(\mathbf{x}_t), \quad (13)$$

which means that GENIE relies on *second-order score functions* $\nabla_{\mathbf{x}_t}^{\top} \nabla_{\mathbf{x}_t} \log p_t(\mathbf{x}_t)$ under the hood.

Given a DDM, that is, given $\boldsymbol{\epsilon}_{\boldsymbol{\theta}}$, we could compute the derivative $d_{\gamma_t}\boldsymbol{\epsilon}_{\boldsymbol{\theta}}$ for the GENIE scheme in Eq. (9) using automatic differentiation (AD). This would, however, make a single step of GENIE at least twice as costly as DDIM, because we would need a forward pass through the $\boldsymbol{\epsilon}_{\boldsymbol{\theta}}$ network to compute $\boldsymbol{\epsilon}_{\boldsymbol{\theta}}(\mathbf{x}_t, t)$ itself, and another pass to compute the JVPs and the time derivative in Eq. (12). These

forward passes cannot be parallelized, since the vector-part of $\text{JVP}_1$ in Eq. (12) involves $\boldsymbol{\epsilon_\theta}$ itself, and needs to be known before computing the JVP. To accelerate sampling, this overhead is too expensive.

**Gradient Distillation:** To avoid this overhead, we propose to first distill $d_{\gamma_t}\boldsymbol{\epsilon_\theta}$ into a separate neural network. During distillation training, we can use the slow AD-based calculation of $d_{\gamma_t}\boldsymbol{\epsilon_\theta}$, but during synthesis we call the trained neural network. We build on the observation that the internal representations of the neural network modeling $\boldsymbol{\epsilon_\theta}$ (in our case a U-Net [79] architecture) can be used for downstream tasks [80, 81]: specifically, we provide the last feature layer from the $\boldsymbol{\epsilon_\theta}$ network together with its time embedding as well as $\mathbf{x}_t$ and the output $\boldsymbol{\epsilon_\theta}(\mathbf{x}_t, t)$ to a small prediction head $\boldsymbol{k_\psi}(\mathbf{x}_t, t)$ that models the different terms in Eq. (12) (see Fig. 4). The overhead generated by

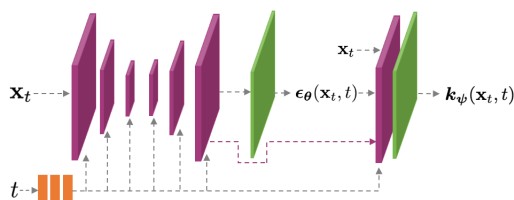

Figure 4: Our distilled model $\boldsymbol{k_\psi}$ that predicts the gradient $d_{\gamma_t}\boldsymbol{\epsilon_\theta}$ is implemented as a small additional output head on top of the first-order score model $\boldsymbol{\epsilon_\theta}$. Purple layers are used both in $\boldsymbol{\epsilon_\theta}$ and $\boldsymbol{k_\psi}$; green layers are specific for $\boldsymbol{\epsilon_\theta}$ and $\boldsymbol{k_\psi}$.

$\boldsymbol{k_\psi}$ is small, for instance less than 2% for our CIFAR-10 model (also see Sec. 5), and we found this approach to provide excellent performance. Note that in principle we could also train an independent deep neural network, which does not make use of the internal representations of $\boldsymbol{\epsilon_\theta}$ and could therefore theoretically be run in parallel to the $\boldsymbol{\epsilon_\theta}$ model. We justify using small prediction heads over independent neural networks because AD-based distillation training is slow: in each training iteration we first need to call the $\boldsymbol{\epsilon_\theta}$ network, then calculate the JVP terms, and only then can we call the distillation model. By modeling $d_{\gamma_t}\boldsymbol{\epsilon_\theta}$ via small prediction heads, while reusing the internal representation of the score model, we can make training relatively fast: we only need to train $\boldsymbol{k_\psi}$ for up to 50k iterations. In contrast, training score models from scratch takes roughly an order of magnitude more iterations. We leave training of independent networks to predict $d_{\gamma_t}\boldsymbol{\epsilon_\theta}$ to future work.

**Mixed Network Parameterization:** We found that learning $d_{\gamma_t}\boldsymbol{\epsilon_\theta}$ directly as single output of a neural network can be challenging. Assuming a single data point distribution $p_0(\mathbf{x}_0) = \delta(\mathbf{x}_0 = \mathbf{0})$, for which we know the diffused score function and all higher-order derivatives analytically, we found that the terms in Eq. (12) all behave very differently within the $t \in [0, 1]$ interval (for instance, the prefactor of $\text{JVP}_1$ in Eq. (12) approaches 1 as $t \to 0$, while $\text{JVP}_2$'s prefactor vanishes). As outlined in detail in App. C.2.3, this simple single data point assumption implies an effective *mixed network parameterization*, an approach inspired by the "mixed score parametrizations" in Vahdat et al. [49] and Dockhorn et al. [60]. In particular, we model

$$\boldsymbol{k_\psi} = -\frac{1}{\gamma_t}\boldsymbol{k_\psi}^{(1)} + \frac{\gamma_t}{1+\gamma_t^2}\boldsymbol{k_\psi}^{(2)} + \frac{1}{\gamma_t(1+\gamma_t^2)}\boldsymbol{k_\psi}^{(3)} \approx d_{\gamma_t}\boldsymbol{\epsilon_\theta}, \tag{14}$$

where $\boldsymbol{k_\psi}^{(i)}(\mathbf{x}_t, t)$, $i \in \{1, 2, 3\}$, are different output channels of the neural network (i.e. the additional head on top of the $\boldsymbol{\epsilon_\theta}$ network). The three terms in Eq. (14) exactly correspond to the three terms of Eq. (12), in the same order. We show the superior performance of this parametrization in Sec. 5.3.

**Learning Objective:** Ideally, we would like our model $\boldsymbol{k_\psi}$ to match $d_{\gamma_t}\boldsymbol{\epsilon_\theta}$ exactly, for all $t \in [0, T]$ and $\mathbf{x}_t$ in the diffused data distribution, which the generative ODE trajectories traverse. This suggests a simple (weighted) $L_2$-loss, similar to regular score matching losses for DDMs [1, 57]:

$$\min_{\boldsymbol{\psi}} \mathbb{E}_{t\sim\mathcal{U}[t_{\text{cutoff}},1], \mathbf{x}_0\sim p(\mathbf{x}_0), \boldsymbol{\epsilon}\sim\mathcal{N}(\mathbf{0},\boldsymbol{I})} \left[ g_{\text{d}}(t)\|\boldsymbol{k_\psi}(\alpha_t\mathbf{x}_0 + \sigma_t\boldsymbol{\epsilon}, t) - d_{\gamma_t}\boldsymbol{\epsilon_\theta}(\alpha_t\mathbf{x}_0 + \sigma_t\boldsymbol{\epsilon}, t)\|_2^2 \right] \tag{15}$$

for diffused data points $\alpha_t\mathbf{x}_0 + \sigma_t\boldsymbol{\epsilon}$ and $g_{\text{d}}(t) = \gamma_t^2$ to counteract the $1/\gamma_t$ in the first and third terms of Eq. (14). This leads to a roughly constant loss over different time values $t$. During training we compute $d_{\gamma_t}\boldsymbol{\epsilon_\theta}$ via AD; however, at inference time we use the learned prediction head $\boldsymbol{k_\psi}$ to approximate $d_{\gamma_t}\boldsymbol{\epsilon_\theta}$. In App. C.2.4, we provide pseudo code for training and sampling with heads $\boldsymbol{k_\psi}$. Note that our distillation objective is consistent and principled: if $\boldsymbol{k_\psi}$ matches $d_{\gamma_t}\boldsymbol{\epsilon_\theta}$ exactly, the resulting GENIE algorithm recovers the second TTM *exactly* (extended discussion in App. B.4).

**Alternative Learning Approaches:** As shown in Eq. (13), GENIE relies on second-order score functions. Recently, Meng et al. [82] directly learnt such higher-order scores with higher-order score matching objectives. Directly applying these techniques has the downside that we would need to explicitly form the higher-order score terms $\nabla_{\mathbf{x}_t}^\top\boldsymbol{\epsilon_\theta}(\mathbf{x}_t, t)$, which are very high-dimensional for data such as images. Low-rank approximations are possible, but potentially insufficient for high performance. In our approach, we are avoiding this complication by directly modeling the lower-dimensional JVPs. We

found that the methods from Meng et al. [82] can be modified to provide higher-order score matching objectives for the JVP terms required for GENIE and we briefly explored this (see App. D). However, our distillation approach with AD-based higher-order gradients worked much better. Nevertheless, this is an interesting direction for future research. To the best of our knowledge, GENIE is the first solver for the generative differential equations of DDMs that *directly* uses higher-order scores (in the form of the distilled JVPs) for generative modeling without finite difference or other approximations.

## 4   Related Work

**Accelerated Sampling from DDMs.** Several previous works address the slow sampling of DDMs: One line of work reduces and readjusts the timesteps [3, 64] used in time-discretized DDMs [1, 71]. This can be done systematically by grid search [32] or dynamic programming [83]. Bao et al. [65] speed up sampling by defining a new DDM with optimal reverse variances. DDIM [58], discussed in Sec. 2, was also introduced as a method to accelerate DDM synthesis. Further works leverage modern ODE and SDE solvers for fast synthesis from (continuous-time) DDMs: For instance, higher-order Runge–Kutta methods [57, 84] and adaptive step size SDE solvers [62] have been used. These methods are not optimally suited for the few-step synthesis regime, in which GENIE shines; see also Sec. 5. Most closely related to our work is Liu et al. [63], which simulates the DDIM ODE [58] using a higher-order linear multistep method [77]. As shown in Sec. 3, linear multistep methods can be considered an approximation of the TTMs used in GENIE. Furthermore, Tachibana et al. [61] solve the generative SDE via a higher-order Itô–Taylor method [59] and in contrast to our work, they propose to use an "ideal derivative trick" to approximate higher-order score functions. In App. B.2, we show that applying this ideal derivative approximation to the DDIM ODE does not have any effect: the "ideal derivatives" are zero by construction. Note that in GENIE, we in fact use the DDIM ODE, rather than, for example, the regular Probability Flow ODE [57], as the base ODE for GENIE.

Alternatively, sampling from DDMs can also be accelerated via learning: For instance, Watson et al. [66] learn parameters of a generalized family of DDMs by optimizing for perceptual output quality. Luhman and Luhman [68] and Salimans and Ho [69] distill a DDIM sampler into a student model, which enables sampling in as few as a single step. Xiao et al. [67] replace DDMs' Gaussian samplers with expressive generative adversarial networks, similarly allowing for few-step synthesis. GENIE can also be considered a learning-based approach, as we distill a derivative of the generative ODE into a separate neural network. However, in contrast to the mentioned methods, GENIE still solves the true underlying generative ODE, which has major advantages: for instance, it can still be used easily for classifier-guided sampling [4, 57, 70] and to efficiently encode data into latent space—a prerequisite for likelihood calculation [47, 57] and editing applications [17]. Note that the learnt sampler [66] defines a proper probabilistic generalized DDM; however, it isn't clear how it relates to the generative SDE or ODE and therefore how compatible the method is with applications such as classifier guidance.

Other approaches to accelerate DDM sampling change the diffusion itself [60, 85, 86] or train DDMs in the latent space of a Variational Autoencoder [49]. GENIE is complementary to these methods.

**Higher-Order ODE Gradients beyond DDMs.** TTMs [78] and other methods that leverage higher-order gradients are also applied outside the scope of DDMs. For instance, higher-order derivatives can play a crucial role when developing solvers [87] and regularization techniques [88, 89] for neural ODEs [74, 75]. Outside the field of machine learning, higher-order TTMs have been widely studied, for example, to develop solvers for stiff [90] and non-stiff [90, 91] systems.

**Concurrent Works.** Zhang and Chen [92] motivate the DDIM ODE from an exponential integrator perspective applied to the Probability Flow ODE and propose to apply existing solvers from the numerical ODE literature, namely, Runge–Kutta and linear multistepping, to the DDIM ODE directly. Lu et al. [93] similarly recognize the semi-linear structure of the Probability Flow ODE, derive dedicated solvers, and introduce new step size schedulers to accelerate DDM sampling. Karras et al. [94] propose new fast solvers, both deterministic and stochastic, specifically designed for the differential equations arising in DDMs. Both Zhang et al. [95] and Karras et al. [94] realize that the DDIM ODE has "straight line solution trajectories" for spherical normal data and single data points—this exactly corresponds to our derivation that the higher-order terms in the DDIM ODE are zero in such a setting (see App. B.2). Bao et al. [96] learn covariance matrices for DDM sampling using prediction heads somewhat similar to the ones in GENIE; in App. G.1, we thoroughly discuss the differences between GENIE and the method proposed in Bao et al. [96].

# 5 Experiments

**Datasets:** We run experiments on five datasets: CIFAR-10 [97] (resolution 32), LSUN Bedrooms [98] (128), LSUN Church-Outdoor [98] (128), (conditional) ImageNet [99] (64), and AFHQv2 [100] (512). On AFHQv2 we only consider the subset of cats; referred to as "Cats" in the remainder of this work.

**Architectures:** Except for CIFAR-10 (we use a checkpoint by Song et al. [57]), we train our own score models using architectures introduced by previous works [1, 4]. The architecture of our prediction heads is based on (modified) BigGAN residual blocks [57, 101]. To minimize computational overhead, we only use a single residual block. See App. C for training and architecture details.

**Evaluation:** We measure sample quality via Fréchet Inception Distance [FID, 102] (see App. F.1).

**Synthesis Strategy:** We simulate the DDIM ODE from $t=1$ up to $t=10^{-3}$ using evaluation times following a quadratic function (*quadratic striding* [58]). For variance-preserving DDMs, it can be beneficial to denoise the ODE solver output at the cutoff $t=10^{-3}$, i.e., $\mathbf{x}_0 = \frac{\mathbf{x}_t - \sigma_t \boldsymbol{\epsilon}_\theta(\mathbf{x}_t, t)}{\alpha_t}$ [57, 103]. Note that the denoising step involves a score model evaluation, and therefore "loses" a function evaluation that could otherwise be used as an additional step in the ODE solver. To this end, denoising the output of the ODE solver is left as a hyperparameter of our synthesis strategy.

**Analytical First Step (AFS):** Every additional neural network call becomes crucial in the low number of function evaluations (NFEs) regime. We found that we can improve the performance of GENIE and all other methods evaluated on our checkpoints by replacing the learned score with the (analytical) score of $\mathcal{N}(\mathbf{0}, \boldsymbol{I}) \approx p_{t=1}(\mathbf{x}_t)$ in the first step of the ODE solver. The "gained" function evaluation can then be used as an additional step in the ODE solver. Similarly to the denoising step mentioned above, AFS is treated as a hyperparameter of our **Synthesis Strategy**. AFS details in App. F.2.

**Accounting for Computational Overhead:** GENIE has a slightly increased computational overhead compared to other solvers due to the prediction head $\boldsymbol{k}_\psi$. The computational overhead is increased by 1.47%, 2.83%, 14.0%, and 14.4% on CIFAR-10, ImageNet, LSUN Bedrooms, and LSUN Church-Outdoor, respectively (see also App. C.2.5). This additional overhead is always accounted for implicitly: we divide the NFEs by the computational overhead and round to the nearest integer. For example, on LSUN Bedrooms, we compare baselines with 10/15 NFEs to GENIE with 9/13 NFEs.

## 5.1 Image Generation

In Fig. 5 we compare our method to the most competitive baselines. In particular, on the same score model checkpoints, we compare GENIE with DDIM [58], S-PNDM [63], and F-PNDM [63]. For these four methods, we only include the best result over the two hyperparameters discussed above, namely, the denoising step and AFS (see App. F.6 for tables with all results). We also include three competitive results from the literature [64–66] that use different checkpoints and sampling strategies: for each method, we include the best result for their respective set of hyperparameters. We do not compare in this figure with Knowledge Distillation [KD, 68], Progressive Distillation [PG, 69] and Denoising Diffusion GANs [DDGAN, 67] as they do not solve the generative ODE/SDE and use fundamentally different sampling approaches with drawbacks discussed in Sec. 4.

For NFEs $\in \{10, 15, 20, 25\}$, GENIE outperforms all baselines (on the same checkpoint) on all four datasets (see detailed results in App. F.6 and GENIE image samples in App. F.7). On CIFAR-10 and (conditional) ImageNet, GENIE also outperforms these baselines for NFEs=5, whereas DDIM outperforms GENIE slightly on the LSUN datasets (see tables in App. F.6). GENIE also performs better than the three additional baselines from the literature (which use different checkpoints and sampling strategies) with the exception of the Learned Sampler [LS, 66] on LSUN Bedrooms for NFEs=20. Though LS uses a learned striding schedule on LSUN Bedrooms (whereas GENIE simply uses quadratic striding), the LS's advantage is most likely due to the different checkpoint. In Tab. 1, we investigate the effect of optimizing the striding schedule, via learning (LS) or grid search (DDIM & GENIE), on CIFAR-10 and find that its significance decreases rapidly with increased NFEs (also see App. F.6 for details). In Tab. 1, we also show additional baseline results; however, we do not include commonly-used adaptive step size solvers in Fig. 5, as they are arguably not well-suited for this low NFE regime: for example, on the same CIFAR-10 checkpoint we use for GENIE, the adaptive SDE solver introduced in Jolicoeur-Martineau et al. [62] obtains an FID of 82.4 at 48 NFEs. Also on the same checkpoint, the adaptive Runge–Kutta 4(5) [84] method applied to the ProbabilityFlow ODE achieves an FID of 13.1 at 38 NFEs (solver tolerances set to $10^{-2}$).

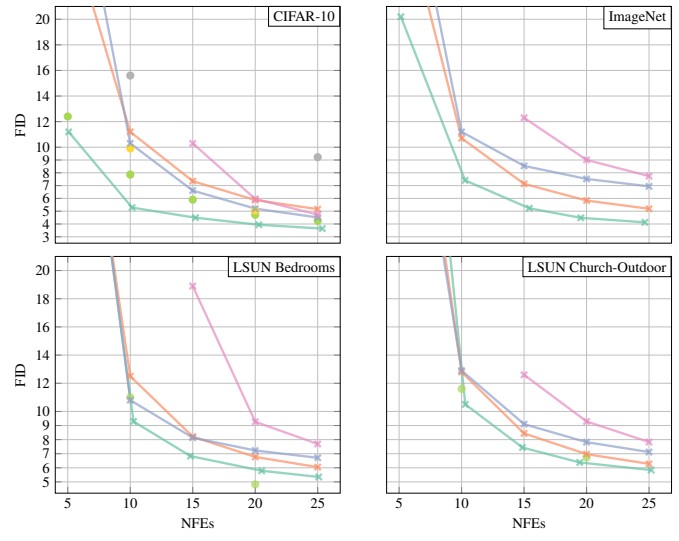

Figure 5: Unconditional performance on four popular benchmark datasets. The first four methods use the same score model checkpoints, whereas the last three methods all use different checkpoints. (†): numbers are taken from literature.

| Method | NFEs=5 | NFEs=10 | NFEs=15 | NFEs=20 | NFEs=25 |
|---|---|---|---|---|---|
| GENIE (ours) (∗) | **11.2** | **5.28** | **4.49** | **3.94** | **3.64** |
| GENIE (ours) | 13.9 | 5.97 | 4.49 | 3.94 | 3.67 |
| DDIM [58] (∗) | 27.6 | 11.2 | 7.35 | 5.87 | 5.16 |
| DDIM [58] | 29.7 | 11.2 | 7.35 | 5.87 | 5.16 |
| S-PNDM [63] | 35.9 | 10.3 | 6.61 | 5.20 | 4.51 |
| F-PNDM [63] | N/A | N/A | 10.3 | 5.96 | 4.73 |
| Euler–Maruyama | 325 | 230 | 164 | 112 | 80.3 |
| FastDDIM [64] (†) | - | 9.90 | - | 5.05 | - |
| Learned Sampler [66] (†/ ∗) | 12.4 | 7.86 | 5.90 | 4.72 | 4.25 |
| Learned Sampler [66] (†) | 14.3 | 8.15 | 5.94 | 4.89 | 4.47 |
| Analytic DDIM [65] (†) | - | 14.0 | - | - | 5.71 |
| CLD-SGM [60] | 334 | 306 | 236 | 162 | 106 |
| VESDE-PC [57] | 461 | 461 | 461 | 461 | 462 |

Table 1: Unconditional CIFAR-10 generative performance (measured in FID). Methods above the middle line use the same score model checkpoint; methods below all use different ones. (†): numbers are taken from literature. (∗): methods either learn an optimal striding schedule (Learned Sampler) or do a small grid search over striding schedules (DDIM & GENIE); also see App. F.6

The results in Fig. 5 suggest that higher-order gradient information, as used in GENIE, can be efficiently leveraged for image synthesis. Despite using small prediction heads our distillation seems to be sufficiently accurate: for reference, replacing the distillation heads with the derivatives computed via AD, we obtain FIDs of 9.22, 4.11, 3.54, 3.46 using 10, 20, 30, and 40 NFEs, respectively (NFEs adjusted assuming an additional computational overhead of 100%). As discussed in Sec. 3, linear multistep methods such as S-PNDM [63] and F-PNDM [63] can be considered (finite difference) approximations to TTMs as used in GENIE. These approximations can be inaccurate for large timesteps, which potentially explains their inferior performance when compared to GENIE. When compared to DDIM, the superior performance of GENIE seems to become less significant for large NFE: this is in line with the theory, as higher-order gradients contribute less for smaller step sizes (see the GENIE scheme in Eq. (9)). Approaches such as FastDDIM [64] and AnalyticDDIM [65], which adapt variances and discretizations of discrete-time DDMs, are useful; however, GENIE suggests that rigorous higher-order ODE solvers leveraging the continuous-time DDM formalism are still more powerful. To the best of our knowledge, the only methods that outperform GENIE abandon this ODE or SDE formulation entirely and train NFE-specific models [67, 69] which are optimized for the single use-case of image synthesis.

## 5.2 Guidance and Encoding

As discussed in Sec. 4, one major drawback of approaches such as KD [68], PG [69] and DDGAN [67] is that they abandon the ODE/SDE formalism, and cannot easily use methods such as classifier(-free) guidance [57, 70] or perform image encoding. However, these techniques can play an important role in synthesizing photorealistic images from DDMs [3, 4, 15, 17], as well as for image editing tasks [12, 17].

**Classifier-Free Guidance [70]:** We replace the unconditional model $\epsilon_{\boldsymbol{\theta}}(\mathbf{x}_t, t)$ with $\hat{\epsilon}_{\boldsymbol{\theta}}(\mathbf{x}_t, t, c, w) = (1 + w)\epsilon_{\boldsymbol{\theta}}(\mathbf{x}_t, t, c) - w\epsilon_{\boldsymbol{\theta}}(\mathbf{x}_t, t)$ in the DDIM ODE (cf Eq. (6)), where $\epsilon_{\boldsymbol{\theta}}(\mathbf{x}_t, t, c)$ is a conditional model and $w > 1.0$ is the "guidance scale". GENIE then requires the derivative

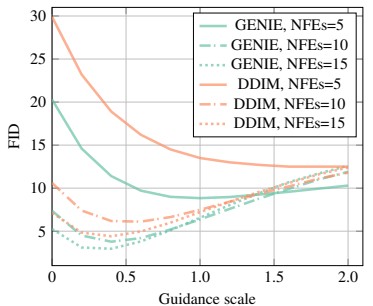

Figure 6: Sample quality as a function of guidance scale on ImageNet.

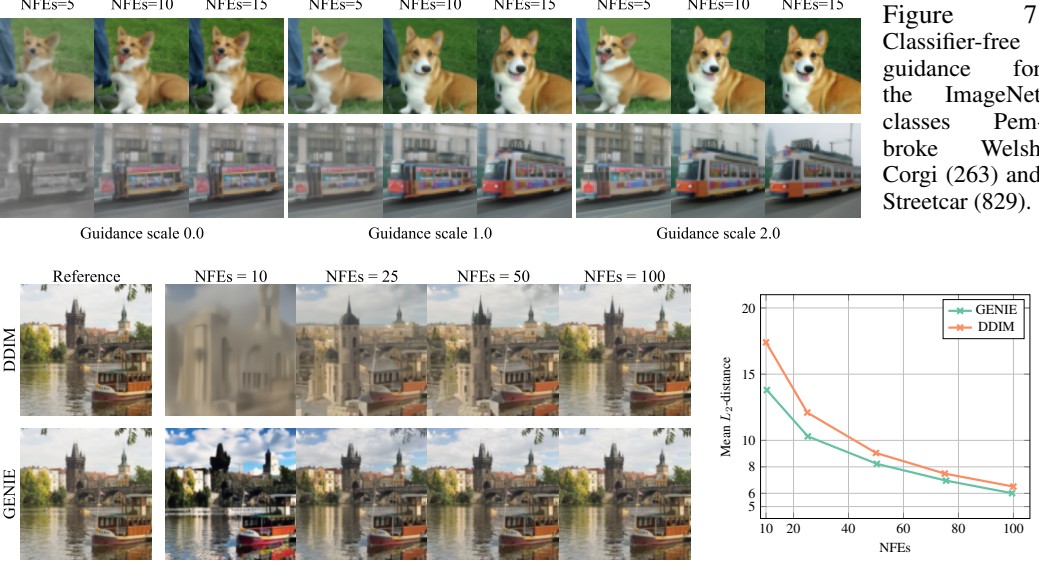

Figure 7: Classifier-free guidance for the ImageNet classes Pembroke Welsh Corgi (263) and Streetcar (829).

Figure 8: Encoding and subsequent decoding on LSUN Church-Outdoor. *Left:* Visual reconstruction. *Right:* $L_2$-distance to reference in Inception feature space [104], averaged over 100 images.

$$d_{\gamma_t}\hat{\boldsymbol{\epsilon}}_{\boldsymbol{\theta}}(\mathbf{x}_t, t, c, w) = (1+w)d_{\gamma_t}\boldsymbol{\epsilon}_{\boldsymbol{\theta}}(\mathbf{x}_t, t, c) - wd_{\gamma_t}\boldsymbol{\epsilon}_{\boldsymbol{\theta}}(\mathbf{x}_t, t). \quad (16)$$

for guidance. Hence, we need to distill $d_{\gamma_t}\boldsymbol{\epsilon}_{\boldsymbol{\theta}}(\mathbf{x}_t, t, c)$ and $d_{\gamma_t}\boldsymbol{\epsilon}_{\boldsymbol{\theta}}(\mathbf{x}_t, t)$, for which we could also share parameters [70]. We compare GENIE with DDIM on ImageNet in Fig. 6. GENIE clearly outperforms DDIM, in particular for few NFEs, and GENIE also synthesizes high-quality images (see Fig. 7).

**Image Encoding:** We can use GENIE also to solve the generative ODE in reverse to encode given images. Therefore, we compare GENIE to DDIM on the "encode-decode" task, analyzing reconstructions for different NFEs (used twice for encoding and decoding): We find that GENIE reconstructs images much more accurately (see Fig. 8). For more details on this experiment as well as the guidance experiment above, see App. F.4 and App. F.3, respectively. We also show latent space interpolations for both GENIE and DDIM in App. F.5.

## 5.3 Ablation Studies

We perform ablation studies over architecture and training objective for the prediction heads used in GENIE: In Tab. 2, "No mixed" refers to learning $d_{\gamma_t}\boldsymbol{\epsilon}_{\boldsymbol{\theta}}$ directly as single network output without mixed network

Table 2: CIFAR-10 ablation studies (measured in FID).

| Ablation | NFEs=5 | NFEs=10 | NFEs=15 | NFEs=20 | NFEs=25 |
|---|---|---|---|---|---|
| Standard | 13.9 | 6.04 | 4.49 | 3.94 | 3.67 |
| No mixed | 14.7 | 6.32 | 4.82 | 4.31 | 4.10 |
| No weighting | 14.8 | 7.45 | 5.89 | 5.17 | 4.80 |
| Bigger model | 13.7 | 5.58 | 4.46 | 4.05 | 3.77 |

parameterization; "No weighting" refers to setting $g_d(t) = 1$ in Eq. (15); "Standard" uses both the mixed network parameterization and the weighting function $g_d(t) = \gamma_t^2$. We can see that having both the mixed network parametrization and the weighting function is clearly beneficial. We also tested deeper networks in the prediction heads: for "Bigger model" we increased the number of residual blocks from one to two. The performance is roughly on par with "Standard", and we therefore opted for the smaller head due to the lower computational overhead.

## 5.4 Upsampling

Cascaded diffusion model pipelines [2] and DDM-based super-resolution [8] have become crucial ingredients in DDMs for large-scale image generation [105]. Hence, we also explore the applicability of GENIE in this setting. We train a $128 \times 128$ base model as well as a $128 \times 128 \rightarrow 512 \times 512$ diffusion upsampler [2, 8] on

Table 3: Cats (upsampler) generative performance (measured in FID).

| Method | NFEs=5 | NFEs=10 | NFEs=15 |
|---|---|---|---|
| GENIE (ours) | **5.53** | **4.90** | **4.83** |
| DDIM [58] | 9.47 | 6.64 | 5.85 |
| S-PNDM [63] | 14.6 | 11.0 | 8.83 |
| F-PNDM [63] | N/A | N/A | 11.7 |

Cats. In Tab. 3, we compare the generative performance of GENIE to other fast samplers for the upsampler (in isolation). We find that GENIE performs very well on this task: with only five NFEs GENIE outperforms all other methods at NFEs=15. We show upsampled samples for GENIE with NFEs=5 in Fig. 9. For more quantitative and qualitative results, we refer to App. F.6 and App. F.7, respectively. Training and inference details for the score model and the GENIE prediction head, for both base model and upsampler, can be found in App. C.

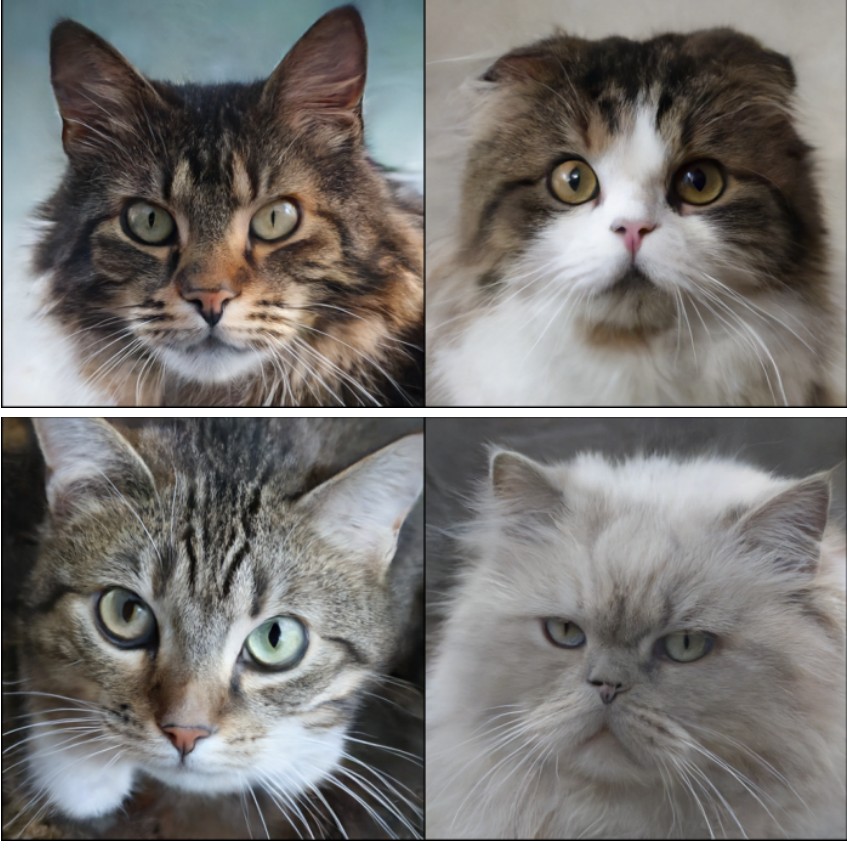

Figure 9: High-resolution images generated with the $128 \times 128 \rightarrow 512 \times 512$ GENIE upsampler using only five neural network calls. For the two images at the top, the upsampler is conditioned on test images from the Cats dataset. For the two images at the bottom, the upsampler is conditioned on samples from the $128 \times 128$ GENIE base model (generated using 25 NFEs); an upsampler neural network evaluation is roughly four times as expensive as a base model evaluation.

## 6 Conclusions

We introduced GENIE, a higher-order ODE solver for DDMs. GENIE improves upon the commonly used DDIM solver by capturing the local curvature of its ODE's gradient field, which allows for larger step sizes when solving the ODE. We further propose to distill the required higher-order derivatives into a small prediction head—which we can efficiently call during inference—on top of the first-order score network. A limitation of GENIE is that it is still slightly slower than approaches that abandon the differential equation framework of DDMs altogether, which, however, comes at the considerable cost of preventing applications such as guided sampling. To overcome this limitation, future work could leverage even higher-order gradients to accelerate sampling from DDMs even further (also see App. G.2).

**Broader Impact.** Fast synthesis from DDMs, the goal of GENIE, can potentially make DDMs an attractive method for promising interactive generative modeling applications, such as digital content creation or real-time audio synthesis, and also reduce DDMs' environmental footprint by decreasing the computational load during inference. Although we validate GENIE on image synthesis, it could also be utilized for other tasks, which makes its broader societal impact application-dependent. In that context, it is important that practitioners apply an abundance of caution to mitigate impacts given generative modeling can also be used for malicious purposes, discussed for instance in Vaccari and Chadwick [106], Nguyen et al. [107], Mirsky and Lee [108].

## Acknowledgements

This work was funded by NVIDIA. Tim Dockhorn acknowledges additional funding from the Vector Institute Research Grant, not in direct support of this work. We thank Yaoliang Yu for discussions.

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
