# Contents

# A  DDIM ODE

The DDIM ODE has previously been shown [58, 69] to be a re-parameterization of the Probability Flow ODE [57]. In this section, we show an alternative presentation to the ones given in Song et al. [58] and Salimans and Ho [69]. We start from the Probability Flow ODE for variance-preserving continuous-time DDMs [57], i.e.,

$$d\mathbf{x}_t = -\tfrac{1}{2}\beta_t \left[\mathbf{x}_t + \nabla_{\mathbf{x}_t} \log p_t(\mathbf{x}_t)\right] dt, \tag{17}$$

where $\beta_t = -\frac{d}{dt} \log \alpha_t^2$ and $\nabla_{\mathbf{x}_t} \log p_t(\mathbf{x}_t)$ is the *score function*. Replacing the unknown score function with a learned score model $\boldsymbol{s}_{\boldsymbol{\theta}}(\mathbf{x}_t, t) \approx \nabla_{\mathbf{x}_t} \log p_t(\mathbf{x}_t)$, we obtain the approximate Probability Flow ODE

$$d\mathbf{x}_t = -\tfrac{1}{2}\beta_t \left[\mathbf{x}_t + \boldsymbol{s}_{\boldsymbol{\theta}}(\mathbf{x}_t, t)\right] dt. \tag{18}$$

Let us now define $\gamma_t = \sqrt{\frac{1-\alpha_t^2}{\alpha_t^2}}$ and $\bar{\mathbf{x}}_t = \mathbf{x}_t \sqrt{1 + \gamma_t^2}$, and take the (total) derivative of $\bar{\mathbf{x}}_t$ with respect to $\gamma_t$:

$$\frac{d\bar{\mathbf{x}}_t}{d\gamma_t} = \frac{\partial \bar{\mathbf{x}}_t}{\partial \mathbf{x}_t} \frac{d\mathbf{x}_t}{d\gamma_t} + \frac{\partial \bar{\mathbf{x}}_t}{\partial \gamma_t} \tag{19}$$

$$= \sqrt{1 + \gamma_t^2}\, \frac{d\mathbf{x}_t}{d\gamma_t} + \frac{\gamma_t}{\sqrt{1 + \gamma_t^2}} \mathbf{x}_t. \tag{20}$$

The derivative $\frac{d\mathbf{x}_t}{d\gamma_t}$ can be computed as follows

$$\frac{d\mathbf{x}_t}{d\gamma_t} = \frac{d\mathbf{x}_t}{dt} \frac{dt}{d\gamma_t} \quad \text{(by chain rule)} \tag{21}$$

$$= -\frac{1}{2}\beta_t \left[\mathbf{x}_t + \boldsymbol{s}_{\boldsymbol{\theta}}(\mathbf{x}_t, t)\right] \frac{dt}{d\gamma_t} \quad \text{(inserting Eq. (18))} \tag{22}$$

$$= \frac{1}{2} \frac{d \log \alpha_t^2}{dt} \left[\mathbf{x}_t + \boldsymbol{s}_{\boldsymbol{\theta}}(\mathbf{x}_t, t)\right] \frac{dt}{d\gamma_t} \quad \text{(by definition of } \beta_t\text{)} \tag{23}$$

$$= \frac{1}{2} \frac{d \log \alpha_t^2}{d\gamma_t} \left[\mathbf{x}_t + \boldsymbol{s}_{\boldsymbol{\theta}}(\mathbf{x}_t, t)\right] \quad \text{(by chain rule)} \tag{24}$$

$$= \frac{1}{2} \frac{d \log \alpha_t^2}{d\alpha_t^2} \frac{d\alpha_t^2}{d\gamma_t} \left[\mathbf{x}_t + \boldsymbol{s}_{\boldsymbol{\theta}}(\mathbf{x}_t, t)\right] \quad \text{(by chain rule)} \tag{25}$$

$$= \frac{1}{2} \frac{1}{\alpha_t^2} \frac{d\alpha_t^2}{d\gamma_t} \left[\mathbf{x}_t + \boldsymbol{s}_{\boldsymbol{\theta}}(\mathbf{x}_t, t)\right]. \tag{26}$$

We can write $\alpha_t^2$ as a function of $\gamma_t$, i.e., $\alpha_t^2 = \left(\gamma_t^2 + 1\right)^{-1}$, and therefore

$$\frac{d\alpha_t^2}{d\gamma_t} = -\frac{2\gamma_t}{\left(\gamma_t^2 + 1\right)^2}. \tag{27}$$

Inserting Eq. (27) into Eq. (26), we obtain

$$\frac{d\mathbf{x}_t}{d\gamma_t} = -\frac{\gamma_t}{\gamma_t^2 + 1} \left[\mathbf{x}_t + \boldsymbol{s}_{\boldsymbol{\theta}}(\mathbf{x}_t, t)\right]. \tag{28}$$

Lastly, inserting Eq. (28) into Eq. (20), we have

$$\frac{d\bar{\mathbf{x}}_t}{d\gamma_t} = -\frac{\gamma_t}{\sqrt{\gamma_t^2 + 1}} \boldsymbol{s}_{\boldsymbol{\theta}}(\mathbf{x}_t, t) \tag{29}$$

Letting $\boldsymbol{s}_{\boldsymbol{\theta}}(\mathbf{x}_t, t) := -\frac{\boldsymbol{\epsilon}_{\boldsymbol{\theta}}(\mathbf{x}_t, t)}{\sigma_t}$, where $\sigma_t = \sqrt{1 - \alpha_t^2} = \frac{\gamma_t}{\sqrt{\gamma_t^2 + 1}}$, denote a particular parameterization of the score model, we obtain the approximate generative DDIM ODE as

$$\frac{d\bar{\mathbf{x}}_t}{d\gamma_t} = \frac{\gamma_t}{\sqrt{\gamma_t^2 + 1}} \frac{\boldsymbol{\epsilon}_{\boldsymbol{\theta}}(\mathbf{x}_t, t)}{\sigma_t} \tag{30}$$

$$= \boldsymbol{\epsilon}_{\boldsymbol{\theta}}(\mathbf{x}_t, t). \tag{31}$$

# B Synthesis from Denoising Diffusion Models via Truncated Taylor Methods

In this work, we propose Higher-Order Denoising Diffusion Solvers (GENIE). GENIE is based on the *truncated Taylor method* (TTM) [78]. As outlined in Sec. 3, the *p-th* TTM is simply the *p-th order Taylor polynomial* applied to an ODE. For example, for the general $\frac{d\mathbf{y}}{dt} = \boldsymbol{f}(\mathbf{y}, t)$, the *p*-th TTM reads as

$$\mathbf{y}_{t_{n+1}} = \mathbf{y}_{t_n} + h_n \frac{d\mathbf{y}}{dt}\big|_{(\mathbf{y}_{t_n}, t_n)} + \cdots + \frac{1}{p!} h_n^p \frac{d^p \mathbf{y}}{dt^p}\big|_{(\mathbf{y}_{t_n}, t_n)}, \tag{32}$$

where $h_n = t_{n+1} - t_n$. To generate samples from denoising diffusion models, we can, for example, apply the second TTM to the (approximate) Probability Flow ODE or the (approximate) DDIM ODE, resulting in the following respective schemes:

$$\mathbf{x}_{t_{n+1}} = \mathbf{x}_{t_n} + (t_{n+1} - t_n)\boldsymbol{f}(\mathbf{x}_{t_n}, t_n) + \frac{1}{2}(t_{n+1} - t_n)^2 \frac{d\boldsymbol{f}}{dt}\big|_{(\mathbf{x}_{t_n}, t_n)}, \tag{33}$$

where $\boldsymbol{f}(\mathbf{x}_t, t) = -\frac{1}{2}\beta(t)\left[\mathbf{x}_t - \frac{\boldsymbol{\epsilon_\theta}(\mathbf{x}_t, t)}{\sigma_t}\right]$, and

$$\bar{\mathbf{x}}_{t_{n+1}} = \bar{\mathbf{x}}_{t_n} + (\gamma_{t_{n+1}} - \gamma_{t_n})\boldsymbol{\epsilon_\theta}(\mathbf{x}_{t_n}, t_n) + \frac{1}{2}(\gamma_{t_{n+1}} - \gamma_{t_n})^2 \frac{d\boldsymbol{\epsilon_\theta}}{d\gamma_t}\big|_{(\mathbf{x}_{t_n}, t_n)}. \tag{34}$$

In this work, we generate samples from DDMs using the scheme in Eq. (34). We distill the derivative $d_{\gamma_t}\boldsymbol{\epsilon_\theta} := \frac{d\boldsymbol{\epsilon_\theta}}{d\gamma_t}$ into a small neural network $\boldsymbol{k_\psi}$. For training, $d_{\gamma_t}\boldsymbol{\epsilon_\theta}$ is computed via automatic differentiation, however, during inference, we can efficiently query the trained network $\boldsymbol{k_\psi}$.

## B.1 Theoretical Bounds for the Truncated Taylor Method

Consider the *p*-TTM for a general ODE $\frac{d\mathbf{y}}{dt} = \boldsymbol{f}(\mathbf{y}, t)$:

$$\mathbf{y}_{t_{n+1}} = \mathbf{y}_{t_n} + h_n \frac{d\mathbf{y}}{dt}\big|_{(\mathbf{y}_{t_n}, t_n)} + \cdots + \frac{1}{p!} h_n^p \frac{d^p \mathbf{y}}{dt^p}\big|_{(\mathbf{y}_{t_n}, t_n)}. \tag{35}$$

We represent, the exact solution $\mathbf{y}(t_{n+1})$ using the $(p+2)$-th Taylor expansion

$$\mathbf{y}(t_{n+1}) = \mathbf{y}(t_n) + h_n \frac{d\mathbf{y}}{dt}\big|_{(\mathbf{y}_{t_n}, t_n)} + \cdots + \frac{1}{p!} h_n^p \frac{d^p \mathbf{y}}{dt^p}\big|_{(\mathbf{y}_{t_n}, t_n)} + \frac{1}{(p+1)!} h_n^{p+1} \frac{d^{p+1} \mathbf{y}}{dt^{p+1}}\big|_{(\mathbf{y}_{t_n}, t_n)} + \mathcal{O}(h_n^{p+2}). \tag{36}$$

The local truncation error (LTE) introduced by the *p*-th TTM is given by the difference between the two equations above

$$\|\mathbf{y}_{t_{n+1}} - \mathbf{y}(t_{n+1})\| = \|\frac{1}{(p+1)!} h_n^{p+1} \frac{d^{p+1} \mathbf{y}}{dt^{p+1}}\big|_{(\mathbf{y}_{t_n}, t_n)} + \mathcal{O}(h_n^{p+2})\|. \tag{37}$$

For small $h_n$, the LTE is proportional to $h_n^{p+1}$. Consequently, using higher orders $p$ implies lower errors, as $h_n$ usually is a small time step.

In conclusion, this demonstrates that it is preferable to use higher-order methods with lower errors when aiming to accurately solve ODEs like the Probability Flow ODE or the DDIM ODE of diffusion models.

## B.2 Approximate Higher-Order Derivatives via the "Ideal Derivative Trick"

Tachibana et al. [61] sample from DDMs using (an approximation to) a higher-order Itô-Taylor method [59]. In their scheme, they approximate higher-order score functions with the "ideal derivative trick", essentially assuming simple single-point ($\mathbf{x}_0$) data distributions, for which higher-order score functions can be computed analytically (more formally, their approximation corresponds to ignoring the expectation over the full data distribution when learning the score function. They assume that for any $\mathbf{x}_t$, there is a single unique $\mathbf{x}_0$ from the input data to be predicted with the score model). In that case, further assuming the score model $\boldsymbol{\epsilon_\theta}(\mathbf{x}_t, t)$ is learnt perfectly (i.e., it perfectly predicts the noise that was used to generate $\mathbf{x}_t$ from $\mathbf{x}_0$), one has

$$\boldsymbol{\epsilon_\theta}(\mathbf{x}_t, t) \approx \frac{\mathbf{x}_t - \alpha_t \mathbf{x}_0}{\sigma_t}. \tag{38}$$

This expression can now be used to analytically calculate approximate spatial and time derivatives (also see App. F.1 and App. F.2 in Tachibana et al. [61]):

$$\frac{\partial \boldsymbol{\epsilon_\theta}(\mathbf{x}_t, t)}{\partial \mathbf{x}_t} \approx \frac{\partial}{\partial \mathbf{x}_t} \left( \frac{\mathbf{x}_t - \alpha_t \mathbf{x}_0}{\sigma_t} \right) = \frac{1}{\sigma_t} \boldsymbol{I}, \tag{39}$$

and

$$\frac{\partial \boldsymbol{\epsilon_\theta}(\mathbf{x}_t, t)}{\partial t} \approx \frac{\partial}{\partial t} \left( \frac{\mathbf{x}_t - \alpha_t \mathbf{x}_0}{\sigma_t} \right) = -\frac{\mathbf{x}_t - \alpha_t \mathbf{x}_0}{\sigma_t^2} \frac{d\sigma_t}{dt} - \frac{\mathbf{x}_0}{\sigma_t} \frac{d\alpha_t}{dt}. \tag{40}$$

Rearranging Eq. (38), we have

$$\mathbf{x}_0 \approx \frac{\mathbf{x}_t - \sigma_t \boldsymbol{\epsilon_\theta}(\mathbf{x}_t, t)}{\alpha_t}. \tag{41}$$

Inserting this expression, Eq. (40) becomes

$$\frac{\partial \boldsymbol{\epsilon_\theta}(\mathbf{x}_t, t)}{\partial t} \approx \frac{\frac{d \log \alpha_t^2}{dt}}{2\sigma_t} \left( \frac{\boldsymbol{\epsilon_\theta}(\mathbf{x}_t, t)}{\sigma_t} - \mathbf{x}_t \right). \tag{42}$$

We will now proceed to show that the "ideal derivative trick", i.e. using the approximations in Eqs. (39) and (42), results in $d_{\gamma_t} \boldsymbol{\epsilon_\theta} = \mathbf{0}$.

As in Sec. 3, the total derivative $d_{\gamma_t} \boldsymbol{\epsilon_\theta}$ is composed as

$$d_{\gamma_t} \boldsymbol{\epsilon_\theta}(\mathbf{x}_t, t) = \frac{\partial \boldsymbol{\epsilon_\theta}(\mathbf{x}_t, t)}{\partial \mathbf{x}_t} \frac{d\mathbf{x}_t}{d\gamma_t} + \frac{\partial \boldsymbol{\epsilon_\theta}(\mathbf{x}_t, t)}{\partial t} \frac{dt}{d\gamma_t}. \tag{43}$$

Inserting the "ideal derivative trick", the above becomes

$$d_{\gamma_t} \boldsymbol{\epsilon_\theta}(\mathbf{x}_t, t) \approx \frac{1}{\sigma_t} \left( \frac{1}{2} \frac{1}{\alpha_t^2} \frac{d\alpha_t^2}{d\gamma_t} \left[ \mathbf{x}_t - \frac{\boldsymbol{\epsilon_\theta}(\mathbf{x}_t, t)}{\sigma_t} \right] \right) + \left( \frac{\frac{d \log \alpha_t^2}{dt}}{2\sigma_t} \left( \frac{\boldsymbol{\epsilon_\theta}(\mathbf{x}_t, t)}{\sigma_t} - \mathbf{x}_t \right) \right) \frac{dt}{d\gamma_t}, \tag{44}$$

where we have inserted Eq. (26) for $\frac{d\mathbf{x}_t}{d\gamma_t}$ and used the usual parameterization $\boldsymbol{s_\theta}(\mathbf{x}_t, t) := -\frac{\boldsymbol{\epsilon_\theta}(\mathbf{x}_t, t)}{\sigma_t}$. Using $\frac{d \log \alpha_t^2}{dt} = \frac{1}{\alpha_t^2} \frac{d\alpha_t^2}{dt}$ and $\frac{d\alpha_t^2}{dt} \frac{dt}{d\gamma_t} = \frac{d\alpha_t^2}{d\gamma_t}$, we can see that the right-hand side of Eq. (44) is $\mathbf{0}$. Hence, applying the second TTM to the DDIM ODE and using the "ideal derivative trick" is equivalent to the first TTM (Euler's method) applied to the DDIM ODE. We believe that this is potentially a reason why the DDIM solver [58], Euler's method applied to the DDIM ODE, shows such great empirical performance: it can be interpreted as an approximate ("ideal derivative trick") second order ODE solver. On the other hand, our derivation also implies that the "ideal derivative trick" used in the second TTM for the DDIM ODE does not actually provide any benefit over the standard DDIM solver, because all additional second-order terms vanish. Hence, to improve upon regular DDIM, the "ideal derivative trick" is insufficient and we need to learn the higher-order score terms more accurately without such coarse approximations, as we do in our work.

Furthermore, it is interesting to show that we do not obtain the same cancellation effect when applying the "ideal derivative trick" to the Probability Flow ODE in Eq. (18): Let $\boldsymbol{f}(\mathbf{x}_t, t) = -\frac{1}{2}\beta(t) \left[ \mathbf{x}_t - \frac{\boldsymbol{\epsilon_\theta}(\mathbf{x}_t, t)}{\sigma_t} \right]$ (right-hand side of Probability Flow ODE), then

$$\frac{d\boldsymbol{f}}{dt}\Big|_{(\mathbf{x}_t, t)} = \frac{\beta'(t)}{\beta(t)} \boldsymbol{f}(\mathbf{x}_t, t) - \frac{1}{2}\beta(t) \frac{d}{dt} \left[ \mathbf{x}_t - \frac{\boldsymbol{\epsilon_\theta}(\mathbf{x}_t, t)}{\sigma_t} \right] \tag{45}$$

$$= \left[ \frac{\beta'(t)}{\beta(t)} - \frac{1}{2}\beta(t) \right] \boldsymbol{f}(\mathbf{x}_t, t) + \frac{1}{2}\beta(t) \left( \frac{\frac{d\boldsymbol{\epsilon_\theta}(\mathbf{x}_t, t)}{dt}}{\sigma_t} - \sigma_t^{-2} \frac{d\sigma_t}{dt} \boldsymbol{\epsilon_\theta}(\mathbf{x}_t, t) \right), \tag{46}$$

where $\beta'(t) := \frac{d\beta(t)}{dt}$. Using the "ideal derivative trick", we have $\frac{d\boldsymbol{\epsilon_\theta}}{dt} = d_{\gamma_t} \boldsymbol{\epsilon_\theta} \, d_t \gamma_t \approx \mathbf{0}$, and therefore the above becomes

$$\frac{d\boldsymbol{f}}{dt}\Big|_{(\mathbf{x}_t, t)} \approx \left[ \frac{\beta'(t)}{\beta(t)} - \frac{1}{2}\beta(t) \right] \boldsymbol{f}(\mathbf{x}_t, t) - \frac{\beta(t)}{2\sigma_t^2} \frac{d\sigma_t}{dt} \boldsymbol{\epsilon}(\mathbf{x}_t, t). \tag{47}$$

The derivative $\frac{d\sigma_t}{dt}$ can be computed as follows

$$\frac{d\sigma_t}{dt} = \frac{1}{2\sigma_t}\frac{d\sigma_t^2}{dt} \tag{48}$$

$$= \frac{1}{2\sigma_t}\frac{d}{dt}\left(1 - e^{-\int_0^t \beta(t')\,dt'}\right) \tag{49}$$

$$= \frac{\beta(t)e^{-\int_0^t \beta(t')\,dt'}}{2\sigma_t}. \tag{50}$$

Putting everything back together, we have

$$\frac{d\boldsymbol{f}}{dt}|_{(\mathbf{x}_t,t)} = \left[\frac{\beta'(t)}{2\sigma_t} + \frac{\beta^2(t)}{4\sigma_t} - \frac{\beta^2(t)e^{-\int_0^t \beta(t')\,dt'}}{4\sigma_t^3}\right]\boldsymbol{\epsilon}_{\boldsymbol{\theta}}(\mathbf{x}_t,t) + \left[-\frac{\beta'(t)}{2} + \frac{\beta^2(t)}{4}\right]\mathbf{x}_t, \tag{51}$$

which is clearly not $\mathbf{0}$ for all $\mathbf{x}_t$ and $t$. Hence, in contrast to the DDIM ODE, applying Euler's method to the Probability Flow ODE does not lead to an approximate (in the sense of the "ideal derivative trick") second order ODE solver.

Note that very related observations have been made in the concurrent works Karras et al. [94] and Zhang et al. [95]. These works notice that when the data distribution consist only of a single data point or a spherical Gaussian distribution, then the solution trajectories of the generative DDIM ODE are straight lines. In fact, this exactly corresponds to our observation that in such a setting we have $d_{\gamma_t}\boldsymbol{\epsilon}_{\boldsymbol{\theta}} = \mathbf{0}$, as shown above in the analysis of the "ideal derivatives approximation". Note in that context that our above derivation considers the "single data point" distribution assumption, but also applies to the setting where the data is a spherical normal distribution (only $\sigma_t$ would be different, which would not affect the derivation).

## B.3  3rd TTM Applied to the DDIM ODE

As promised in Sec. 3, we show here how to apply the third TTM to the DDIM ODE, resulting in the following scheme:

$$\bar{\mathbf{x}}_{t_{n+1}} = \bar{\mathbf{x}}_{t_n} + h_n\boldsymbol{\epsilon}_{\boldsymbol{\theta}}(\mathbf{x}_{t_n}, t_n) + \frac{1}{2}h_n^2\frac{d\boldsymbol{\epsilon}_{\boldsymbol{\theta}}}{d\gamma_t}|_{(\mathbf{x}_{t_n}, t_n)} + \frac{1}{6}h_n^3\frac{d^2\boldsymbol{\epsilon}_{\boldsymbol{\theta}}}{d\gamma_t^2}|_{(\mathbf{x}_{t_n}, t_n)}, \tag{52}$$

where $h_n = (\gamma_{t_{n+1}} - \gamma_{t_n})$. In the remainder of this section, we derive a computable formula for $\frac{d^2\boldsymbol{\epsilon}_{\boldsymbol{\theta}}}{d\gamma_t^2}$, only containing partial derivatives.

Using the chain rule, we have

$$\frac{d^2\boldsymbol{\epsilon}_{\boldsymbol{\theta}}}{d\gamma_t^2}|_{(\mathbf{x}_t,t)} = \frac{\partial d_\gamma\boldsymbol{\epsilon}_{\boldsymbol{\theta}}(\mathbf{x}_t,t)}{\partial \mathbf{x}_t}\frac{d\mathbf{x}_t}{d\gamma_t} + \frac{\partial d_\gamma\boldsymbol{\epsilon}_{\boldsymbol{\theta}}(\mathbf{x}_t,t)}{\partial t}\frac{dt}{d\gamma_t}, \tag{53}$$

where, using Eq. (43),

$$\frac{\partial d_\gamma\boldsymbol{\epsilon}_{\boldsymbol{\theta}}(\mathbf{x}_t,t)}{\partial \mathbf{x}_t} = \frac{\partial^2\boldsymbol{\epsilon}_{\boldsymbol{\theta}}(\mathbf{x}_t,t)}{\partial \mathbf{x}^2}\frac{d\mathbf{x}_t}{d\gamma_t} + \frac{\partial\boldsymbol{\epsilon}_{\boldsymbol{\theta}}(\mathbf{x}_t,t)}{\partial \mathbf{x}_t}\left(\frac{1}{\sqrt{\gamma_t^2+1}}\frac{\partial\boldsymbol{\epsilon}_{\boldsymbol{\theta}}(\mathbf{x}_t,t)}{\partial \mathbf{x}_t} - \frac{\gamma_t}{1+\gamma_t^2}\boldsymbol{I}\right) + \frac{\partial^2\boldsymbol{\epsilon}_{\boldsymbol{\theta}}(\mathbf{x}_t,t)}{\partial t\partial \mathbf{x}_t}\frac{dt}{d\gamma_t}, \tag{54}$$

and

$$\frac{\partial d_\gamma\boldsymbol{\epsilon}_{\boldsymbol{\theta}}(\mathbf{x}_t,t)}{\partial t} = \frac{\partial}{\partial t}\left(\frac{\partial\boldsymbol{\epsilon}_{\boldsymbol{\theta}}(\mathbf{x}_t,t)}{\partial \mathbf{x}_t}\frac{d\mathbf{x}_t}{d\gamma_t}\right) + \frac{\partial}{\partial t}\left(\frac{\partial\boldsymbol{\epsilon}_{\boldsymbol{\theta}}(\mathbf{x}_t,t)}{\partial t}\frac{dt}{d\gamma_t}\right). \tag{55}$$

The remaining terms in Eq. (55) can be computed as

$$\frac{\partial}{\partial t}\left(\frac{\partial\boldsymbol{\epsilon}_{\boldsymbol{\theta}}(\mathbf{x}_t,t)}{\partial t}\frac{dt}{d\gamma_t}\right) = \frac{\partial^2\boldsymbol{\epsilon}_{\boldsymbol{\theta}}(\mathbf{x}_t,t)}{\partial t^2}\frac{dt}{d\gamma_t} + \frac{\partial\boldsymbol{\epsilon}_{\boldsymbol{\theta}}(\mathbf{x}_t,t)}{\partial t}\frac{d\left(\frac{dt}{d\gamma_t}\right)}{dt}, \tag{56}$$

and

$$\frac{\partial}{\partial t}\left(\frac{\partial\boldsymbol{\epsilon}_{\boldsymbol{\theta}}(\mathbf{x}_t,t)}{\partial \mathbf{x}_t}\frac{d\mathbf{x}_t}{d\gamma_t}\right) = \frac{\partial^2\boldsymbol{\epsilon}_{\boldsymbol{\theta}}(\mathbf{x}_t,t)}{\partial t\,\partial \mathbf{x}_t}\frac{d\mathbf{x}_t}{d\gamma_t} + \frac{\partial\boldsymbol{\epsilon}_{\boldsymbol{\theta}}(\mathbf{x}_t,t)}{\partial \mathbf{x}_t}\frac{\partial\left(\frac{d\mathbf{x}_t}{d\gamma_t}\right)}{\partial t} \tag{57}$$

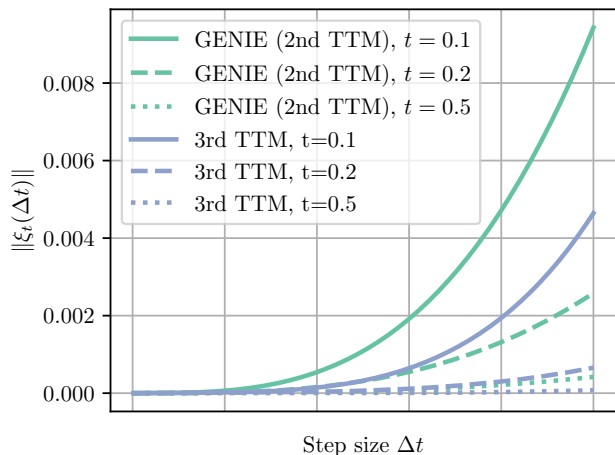

Figure 10: Single step error using analytical score function. See also Fig. 3 (*top*).

where, inserting Eq. (28) for $\frac{d\mathbf{x}_t}{d\gamma_t}$ as well as using the usual parameterization $s_{\boldsymbol{\theta}}(\mathbf{x}_t, t) := -\frac{\boldsymbol{\epsilon}_{\boldsymbol{\theta}}(\mathbf{x}_t, t)}{\sigma_t}$,

$$\frac{\partial\left(\frac{d\mathbf{x}_t}{d\gamma_t}\right)}{\partial t} = \frac{\partial}{\partial t}\left(-\frac{\gamma_t}{\gamma_t^2+1}\left[\mathbf{x}_t - \frac{\boldsymbol{\epsilon}_{\boldsymbol{\theta}}(\mathbf{x}_t, t)}{\sigma_t}\right]\right) \tag{58}$$

$$= \frac{\partial\left(\frac{1}{\sqrt{\gamma_t^2+1}}\right)}{\partial t}\boldsymbol{\epsilon}_{\boldsymbol{\theta}}(\mathbf{x}_t, t) + \frac{1}{\sqrt{\gamma_t^2+1}}\frac{\partial\boldsymbol{\epsilon}_{\boldsymbol{\theta}}(\mathbf{x}_t, t)}{\partial t} - \frac{\partial\left(\frac{\gamma_t}{1+\gamma_t^2}\right)}{\partial t}\mathbf{x}_t \quad \left(\text{using } \sigma_t = \frac{\gamma_t}{\sqrt{\gamma_t^2+1}}\right) \tag{59}$$

$$= \left(-\frac{\gamma_t}{(\gamma_t^2+1)^{3/2}}\boldsymbol{\epsilon}_{\boldsymbol{\theta}}(\mathbf{x}_t, t) + \frac{\gamma_t^2-1}{(\gamma_t^2+1)^2}\mathbf{x}_t\right)\frac{d\gamma_t}{dt} + \frac{1}{\sqrt{\gamma_t^2+1}}\frac{\partial\boldsymbol{\epsilon}_{\boldsymbol{\theta}}(\mathbf{x}_t, t)}{\partial t}. \tag{60}$$

We now have a formula for $\frac{d^2\boldsymbol{\epsilon}_{\boldsymbol{\theta}}}{d\gamma_t^2}$ containing only partial derivatives, and therefore we can compute $\frac{d^2\boldsymbol{\epsilon}_{\boldsymbol{\theta}}}{d\gamma_t^2}$ using automatic differentiation. Note that we could follow the same procedure to compute even higher derivatives of $\boldsymbol{\epsilon}_{\boldsymbol{\theta}}$.

We repeat the 2D toy distribution single step error experiment from Sec. 3 (see also Fig. 3 (*top*) and App. E for details). As expected, in Fig. 10 we can clearly see that the third TTM improves upon the second TTM.

In Fig. 11, we compare the second TTM to the third TTM applied to the DDIM ODE on CIFAR-10. Both for the second and the third TTM, we compute all partial derivatives using automatic differentiation (without distillation). It appears that for using 15 or less steps in the ODE solver, the second TTM performs better than the third TTM. We believe that this could potentially be due to our score model $s_{\boldsymbol{\theta}}(\mathbf{x}_t, t)$ not being accurate enough, in contrast to the above 2D toy distribution experiment, where we have access to the analytical score function. Furthermore, note that when we train $s_{\boldsymbol{\theta}}(\mathbf{x}_t, t)$ via score matching, we never regularize (higher-order) derivatives of the neural network, and therefore there is no incentive for them to be well-behaved. It would be interesting to see if, besides having more accurate score models, regularization techniques such as spectral regularization [109] could potentially alleviate this issue. Also the higher-order score matching techniques derived by Meng et al. [82] could help to learn higher-order derivates of the score functions more accurately. We leave this exploration to future work.

## B.4 GENIE is Consistent and Principled

GENIE is a consistent and principled approach to developing a higher-order ODE solver for sampling from diffusion models: GENIE's design consists of two parts: (1) We are building on the second Truncated Taylor Method (TTM), which is a well-studied ODE solver (see Kloeden and Platen [78])

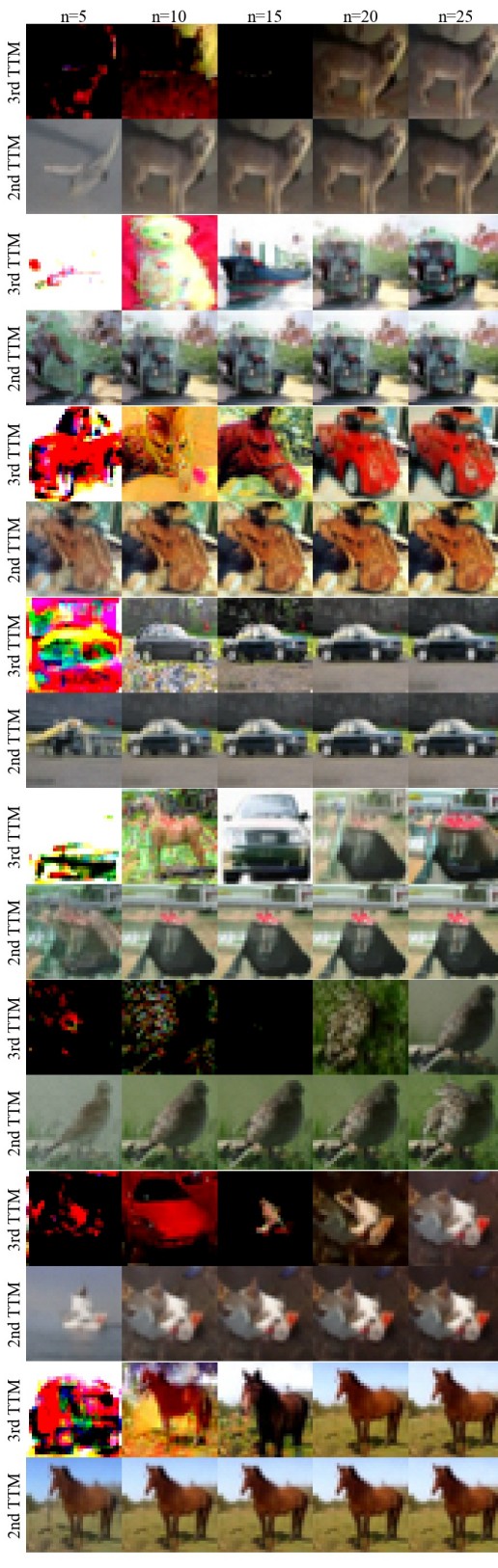

Figure 11: Qualitative comparison of the second and the third TTMs applied to the DDIM ODE on CIFAR-10 (all necessary derivatives calculated with automatic differentiation). The number of steps in the ODE solver is denoted as n.

with provable local and global truncation errors (see also App. B.1). Therefore, if during inference we had access to the ground truth second-order ODE derivatives, which are required for the second TTM, GENIE would simply correspond to the *exact* second TTM.

(2) In principle, we could calculate the exact second-order derivatives *during inference* using automatic differentiation. However, this is too slow for competitive sampling speeds, as it requires additional backward passes through the first-order score network. Therefore, in practice, we use the learned prediction heads $\mathbf{k}_\psi(\mathbf{x}_t, t)$.

Consequently, if $\mathbf{k}_\psi(\mathbf{x}_t, t)$ modeled the ground truth second-order derivatives exactly, i.e. $\mathbf{k}_\psi(\mathbf{x}_t, t) = d_{\gamma_t}\epsilon_\theta(\mathbf{x}_t, t)$ for all $\mathbf{x}_t$ and $t$, we would obtain a rigorous second-order solver based on the TTM, following (1) above.

In practice, distillation will not be perfect. However, given the above analysis, optimizing a neural network $\mathbf{k}_\psi(\mathbf{x}_t, t)$ towards $d_{\gamma_t}\epsilon_\theta(\mathbf{x}_t, t)$ is well motivated and theoretically grounded. In particular, *during training* we are calculating *exact* ODE gradients using automatic differentiation on the first-order score model as distillation targets. Therefore, in the limit of infinite neural network capacity and perfect optimization, we could in theory minimize our distillation objective function (Eq. (15)) perfectly and obtain $\mathbf{k}_\psi(\mathbf{x}_t, t) = d_{\gamma_t}\epsilon_\theta(\mathbf{x}_t, t)$.

Also recall that regular denoising score matching itself, on which all diffusion models rely, follows the exact same argument. In particular, denoising score matching also minimizes a "simple" (weighted) $L_2$-loss between a trainable score model $\mathbf{s}_\theta(\mathbf{x}_t, t)$ and the spatial derivative of the log-perturbation kernel, i.e., $\nabla_{\mathbf{x}_t} \log p_t(\mathbf{x}_t \mid \mathbf{x}_0)$. From this perspective, denoising score matching itself also simply tries to "distill" (spatial) derivatives into a model. If we perfectly optimized the denoising score matching objective, we would obtain a diffusion model that models the data distribution exactly, but in practice, similar to GENIE, we never achieve that due to imperfect optimization and finite-capacity neural networks. Nevertheless, denoising score matching similarly is a well-defined and principled method, precisely because of that theoretical limit in which the distribution can be reproduced exactly.

We would also like to point out that other, established higher-order methods for diffusion model sampling with the generative ODE, such as linear multistep methods [63], make approximations, too, which can be worse in fact. In particular, multistep methods *always* approximate higher-order derivatives in the TTM using finite differences which is crude for large step sizes, as can be seen in Fig. 3 (*bottom*). From this perspective, if our distillation is sufficiently accurate, GENIE can be expected to be more accurate than such multistep methods.

# C  Model and Implementation Details

## C.1  Score Models

We train *variance-preserving* DDMs [57] for which $\sigma_t^2 = 1 - \alpha_t^2$. We follow Song et al. [57] and set $\beta(t) = 0.1 + 19.9t$; note that $\alpha_t = e^{-\frac{1}{2}\int_0^t \beta(t')\,dt'}$. All score models are parameterized as either $s_\theta(\mathbf{x}_t, t) := -\frac{\epsilon_\theta(\mathbf{x}_t,t)}{\sigma_t}$ ($\epsilon$-prediction) or $s_\theta(\mathbf{x}_t, t) := -\frac{\alpha_t \mathbf{v}_\theta(\mathbf{x}_t,t)+\sigma_t \mathbf{x}_t}{\sigma_t}$ ($\mathbf{v}$-prediction), where $\epsilon_\theta(\mathbf{x}_t, t)$ and $\mathbf{v}_\theta(\mathbf{x}_t, t)$ are U-Nets [79]. The $\epsilon$-prediction model is trained using the following score matching objective [1]

$$\min_{\boldsymbol{\theta}} \; \mathbb{E}_{t\sim\mathcal{U}[t_{\mathrm{cutoff}},1],\mathbf{x}_0\sim p(\mathbf{x}_0),\epsilon\sim\mathcal{N}(\mathbf{0},\boldsymbol{I})} \left[ \|\epsilon - \epsilon_\theta(\mathbf{x}_t, t)\|_2^2 \right], \quad \mathbf{x}_t = \alpha_t \mathbf{x}_0 + \sigma_t \epsilon. \tag{61}$$

The $\mathbf{v}$-prediction model is trained using the following score matching objective [69]

$$\min_{\boldsymbol{\theta}} \; \mathbb{E}_{t\sim\mathcal{U}[t_{\mathrm{cutoff}},1],\mathbf{x}_0\sim p(\mathbf{x}_0),\epsilon\sim\mathcal{N}(\mathbf{0},\boldsymbol{I})} \left[ \|\tfrac{\epsilon-\sigma_t \mathbf{x}_t}{\alpha_t} - \mathbf{v}_\theta(\mathbf{x}_t, t)\|_2^2 \right], \quad \mathbf{x}_t = \alpha_t \mathbf{x}_0 + \sigma_t \epsilon, \tag{62}$$

which is referred to as "SNR+1" weighting [69]. The neural network $\mathbf{v}_\theta$ is now effectively tasked with predicting $\mathbf{v} := \alpha_t \epsilon - \sigma_t \mathbf{x}_0$.

**CIFAR-10:** On this dataset, we do not train our own score model, but rather use a checkpoint[2] provided by Song et al. [57]. The model is based on the DDPM++ architecture introduced in Song et al. [57] and predicts $\epsilon_\theta$.

**LSUN Bedrooms and LSUN Church-Outdoor:** Both datasets use exactly the same model structure. The model structure is based on the DDPM architecture introduced in Ho et al. [1] and predicts $\epsilon_\theta$.

**ImageNet:** This model is based on the architecture introduced in Dhariwal and Nichol [4]. We make a small change to the architecture and replace its sinusoidal time embedding by a Gaussian Fourier projection time embedding [57]. The model is class-conditional and we follow Dhariwal and Nichol [4] and simply add the class embedding to the (Gaussian Fourier projection) time embedding. The model predicts $\epsilon_\theta$.

**Cats (Base):** This model is based on the architecture introduced in Dhariwal and Nichol [4]. We make a small change to the architecture and replace its sinusoidal time embedding by a Gaussian Fourier projection time embedding [57]. The model predicts $\mathbf{v}_\theta$.

**Cats (Upsampler):** This model is based on the architecture introduced in Dhariwal and Nichol [4]. We make a small change to the architecture and replace its sinusoidal time embedding by a Gaussian Fourier projection time embedding [57]. The upsampler is conditioned on noisy upscaled lower-resolution images, which are concatenated to the regular channels that form the synthesized outputs of the diffusion model. Therefore, we expand the number of input channels from three to six. We use augmentation conditioning [105] to noise the lower-resolution image. In particular, we upscale $\alpha_{t'}\mathbf{x}_{\mathrm{low}} + \sigma_{t'}\mathbf{z}$, where $\mathbf{x}_{\mathrm{low}}$ is the clean lower-resolution image. During training $t'$ is sampled from $\mathcal{U}[t_{\mathrm{cutoff}}, 1]$. During inference, $t'$ is a hyper-parameter which we set to $0.1$ for all experiments.

We use two-independent Gaussian Fourier projection embeddings for $t$ and $t'$ and concatenate them before feeding them into the layers of the U-Net.

**Model Hyperparameters and Training Details**: All model hyperparameters and training details can be found in Tab. 4.

## C.2  Prediction Heads

We model the derivative $d_{\gamma_t}\epsilon_\theta$ using a small prediction head $\boldsymbol{k}_\psi$ on top of the first-order score model $\epsilon_\theta$. In particular, we provide the last feature layer from the $\epsilon_\theta$ network together with its time embedding as well as $\mathbf{x}_t$ and the output of $\epsilon(\mathbf{x}_t, t)$ to the prediction head (see Fig. 4 for a visualization). We found modeling $d_{\gamma_t}\epsilon_\theta$ to be effective even for our Cats models that learn to predict $\mathbf{v} = \alpha_t \epsilon - \sigma_t \mathbf{x}_0$ rather than $\epsilon$. Directly learning $d_{\gamma_t}\mathbf{v}_\theta$ and adapting the mixed network parameterization (see App. C.2.3) could potentially improve results further. We leave this exploration to future work.

We provide additional details on our architecture next.

---

[2]The checkpoint can be found at https://drive.google.com/file/d/16_-Ahc6ImZV5ClUcOvM5Iivf8OJ1VSif/view?usp=sharing.

Table 4: Model hyperparameters and training details. The CIFAR-10 model is taken from Song et al. [57]; all other models are trained by ourselves.

| Hyperparameter | CIFAR-10 | LSUN Bedrooms | LSUN Church-Outdoor | ImageNet | Cats (Base) | Cats (Upsampler) |
|---|---|---|---|---|---|---|
| **Model** | | | | | | |
| Data dimensionality (in pixels) | 32 | 128 | 128 | 64 | 128 | 512 |
| Residual blocks per resolution | 8 | 2 | 2 | 3 | 2 | 2 |
| Attention resolutions | 16 | 16 | 16 | 8 | (8, 16) | (8, 16) |
| Base channels | 128 | 128 | 128 | 192 | 96 | 192 |
| Channel multipliers | 1,2,3,4 | 1,1,2,2,4,4,4 | 1,1,2,2,4,4,4 | 1,2,3,4 | 1,2,2,3,3 | 1,1,2,2,3,3,4 |
| EMA rate | 0.9999 | 0.9999 | 0.9999 | 0.9999 | 0.9999 | 0.9999 |
| # of head channels | N/A | N/A | N/A | 64 | 64 | 64 |
| # of parameters | 107M | 148M | 148M | 283M | 200M | 80.2M |
| Base architecture | DDPM++ [57] | DDPM [1] | DDPM [1] | [4] | [4] | [4] |
| Prediction | $\epsilon$ | $\epsilon$ | $\epsilon$ | $\epsilon$ | $\mathbf{v}$ | $\mathbf{v}$ |
| **Training** | | | | | | |
| # of iterations | 400k | 300k | 300k | 400k | 400k | 150k |
| # of learning rate warmup iterations | 100k | 100k | 100k | 100k | 100k | 100k |
| Optimizer | Adam | Adam | Adam | Adam | Adam | Adam |
| Mixed precision training | ✗ | ✓ | ✓ | ✓ | ✓ | ✓ |
| Learning rate | $10^{-4}$ | $3 \cdot 10^{-4}$ | $3 \cdot 10^{-4}$ | $2 \cdot 10^{-4}$ | $10^{-4}$ | $10^{-4}$ |
| Gradient norm clipping | 1.0 | 1.0 | 1.0 | 1.0 | 1.0 | 1.0 |
| Dropout | 0.1 | 0.0 | 0.0 | 0.1 | 0.1 | 0.1 |
| Batch size | 128 | 256 | 256 | 1024 | 128 | 64 |
| $t_{\text{cutoff}}$ | $10^{-5}$ | $10^{-3}$ | $10^{-3}$ | $10^{-3}$ | $10^{-3}$ | $10^{-3}$ |

### C.2.1 Model Architecture

The architecture of our prediction heads is based on (modified) BigGAN residual blocks [57, 101]. To minimize computational overhead, we only use a single residual block.

In particular, we concatenate the last feature layer with $\mathbf{x}_t$ as well as $\epsilon_{\boldsymbol{\theta}}(\mathbf{x}_t, t)$ and feed it into a convolutional layer. For the upsampler, we also condition on the noisy up-scaled lower resolution image. We experimented with normalizing the feature layer before concatenation. The output of the convolutional layer as well as the time embedding are then fed to the residual block. Similar to U-Nets used in score models, we normalize the output of the residual block and apply an activation function. Lastly, the signal is fed to another convolutional layer that brings the number of channels to a desired value (in our case nine, three for each $\boldsymbol{k}_{\boldsymbol{\psi}}^{(i)}$, $i \in \{1, 2, 3\}$, in Eq. (66)).

All model hyperparameters can be found in Tab. 5. We also include the additional computational overhead induced by the prediction heads in Tab. 5; see App. C.2.5 for details on how we measured the overhead.

### C.2.2 Training Details

We train for 50k iterations using Adam [110]. We experimented with two base learning rates: $10^{-4}$ and $5 \cdot 10^{-5}$. We furthermore tried two "optimization setups": (linearly) warming up the learning rate in the first 10k iterations (score models are often trained by warming up the learning rate in the first 100k iterations) or, following Salimans and Ho [69], linearly decaying the learning rate to 0 in the entire 50k iterations of training; we respectively refer to these two setups as "warmup" and "decay". We measure the FID every 5k iterations and use the best checkpoint.

Note that we have to compute the Jacobian-vector products in Eq. (12) via automatic differentiation during training. We repeatedly found that computing the derivative $\frac{\partial \epsilon_{\boldsymbol{\theta}}(\mathbf{x}_t, t)}{\partial t}$ via automatic differentiation leads to numerical instability (NaN) for small $t$ when using mixed precision training. For simplicity, we turned off mixed precision training altogether. However, training performance could have been optimized by only turning off mixed precision training for the derivative $\frac{\partial \epsilon_{\boldsymbol{\theta}}(\mathbf{x}_t, t)}{\partial t}$.

All training details can be found in Tab. 5.

### C.2.3 Mixed Network Parameterization

Our mixed network parameterization is derived from a simple single data point assumption, i.e., $p_t(\mathbf{x}_t) = \mathcal{N}(\mathbf{x}_t; \mathbf{0}, \sigma_t^2 \boldsymbol{I})$. This assumption leads to $\epsilon_{\boldsymbol{\theta}}(\mathbf{x}_t, t) \approx \frac{\mathbf{x}_t}{\sigma_t}$ which we can plug into the three

Table 5: Model hyperparameters and training details for the prediction heads.

| Hyperparameter | CIFAR-10 | LSUN Bedrooms | LSUN Church-Outdoor | ImageNet | Cats (Base) | Cats (Upsampler) |
|---|---|---|---|---|---|---|
| **Model** | | | | | | |
| Data dimensionality | 32 | 128 | 128 | 64 | 128 | 512 |
| EMA rate | 0 | 0 | 0 | 0 | 0 | 0 |
| Number of channels | 128 | 128 | 128 | 196 | 196 | 92 |
| # of parameters | 526k | 526k | 526k | 1.17M | 1.17M | 302k |
| Normalize $\mathbf{x}_{\text{embed}}$ | ✗ | ✗ | ✓ | ✗ | ✗ | ✗ |
| **Training** | | | | | | |
| # of iterations | 20k | 40k | 35k | 15k | 20k | 20k |
| Optimizer | Adam | Adam | Adam | Adam | Adam | Adam |
| Optimization setup | Decay | Warmup | Warmup | Warmup | Warmup | Warmup |
| Mixed precision training | ✗ | ✗ | ✗ | ✗ | ✗ | ✗ |
| Learning rate | $5 \cdot 10^{-5}$ | $10^{-4}$ | $10^{-4}$ | $10^{-4}$ | $10^{-4}$ | $10^{-4}$ |
| Gradient norm clipping | 1.0 | 1.0 | 1.0 | 1.0 | 1.0 | 1.0 |
| Dropout | 0.0 | 0.0 | 0.0 | 0.0 | 0.0 | 0.0 |
| Batch size | 128 | 256 | 256 | 256 | 64 | 16 |
| $t_{\text{cutoff}}$ | $10^{-3}$ | $10^{-3}$ | $10^{-3}$ | $10^{-3}$ | $10^{-3}$ | $10^{-3}$ |
| **Inference** | | | | | | |
| Add. comp. overhead | 1.47% | 14.0% | 14.4% | 2.83% | 7.55% | 13.3% |

terms of Eq. (12):

$$\frac{1}{\sqrt{\gamma_t^2+1}}\frac{\partial \boldsymbol{\epsilon}_{\boldsymbol{\theta}}(\mathbf{x}_t,t)}{\partial \mathbf{x}_t}\boldsymbol{\epsilon}_{\boldsymbol{\theta}}(\mathbf{x}_t,t) \approx \frac{1}{\sqrt{\gamma_t^2+1}}\frac{\mathbf{x}_t}{\sigma_t^2} = \frac{1}{\gamma_t}\frac{\mathbf{x}_t}{\sigma_t}, \tag{63}$$

and

$$-\frac{\gamma_t}{1+\gamma_t^2}\frac{\partial \boldsymbol{\epsilon}_{\boldsymbol{\theta}}(\mathbf{x}_t,t)}{\partial \mathbf{x}_t}\mathbf{x}_t \approx -\frac{\gamma_t}{\sigma_t\left(1+\gamma_t^2\right)}\mathbf{x}_t = -\frac{\gamma_t}{1+\gamma_t^2}\frac{\mathbf{x}_t}{\sigma_t}, \tag{64}$$

and finally

$$\frac{\partial \boldsymbol{\epsilon}_{\boldsymbol{\theta}}(\mathbf{x}_t,t)}{\partial t}\frac{dt}{d\gamma_t} \approx -\frac{\mathbf{x}_t}{\sigma_t^2}\frac{d\sigma_t}{dt}\frac{dt}{d\gamma_t} = -\frac{\gamma_t^2+1}{\gamma_t^2}\mathbf{x}_t\frac{1}{\left(\gamma_t^2+1\right)^{3/2}} = -\frac{1}{\gamma_t(1+\gamma_t^2)}\frac{\mathbf{x}_t}{\sigma_t}, \tag{65}$$

where we have used $\sigma_t = \frac{\gamma_t}{\sqrt{\gamma_t^2+1}}$. This derivation therefore implies the following mixed network parameterization

$$\boldsymbol{k}_{\boldsymbol{\psi}} = -\frac{1}{\gamma_t}\boldsymbol{k}_{\boldsymbol{\psi}}^{(1)} + \frac{\gamma_t}{1+\gamma_t^2}\boldsymbol{k}_{\boldsymbol{\psi}}^{(2)} + \frac{1}{\gamma_t(1+\gamma_t^2)}\boldsymbol{k}_{\boldsymbol{\psi}}^{(3)} \approx d_{\gamma_t}\boldsymbol{\epsilon}_{\boldsymbol{\theta}}, \tag{66}$$

where $\boldsymbol{k}_{\boldsymbol{\psi}}^{(i)}(\mathbf{x}_t,t)$, $i \in \{1,2,3\}$, are different output channels of the neural network (i.e. the additional head on top of the $\boldsymbol{\epsilon}_{\boldsymbol{\theta}}$ network). To provide additional intuition, we basically replaced the $-\frac{\mathbf{x}_t}{\sigma_t}$ terms in Eqs. (63) to (65) by neural networks. However, we know that for approximately Normal data $\frac{\mathbf{x}_t}{\sigma_t} \approx \boldsymbol{\epsilon}_{\boldsymbol{\theta}}(\mathbf{x}_t,t)$, where $\boldsymbol{\epsilon}_{\boldsymbol{\theta}}(\mathbf{x}_t,t)$ predicts "noise" values $\boldsymbol{\epsilon}$ that were drawn from a standard Normal distribution and are therefore varying on a well-behaved scale. Consequently, up to the Normal data assumption, we can also expect our prediction heads $\boldsymbol{k}_{\boldsymbol{\psi}}^{(i)}(\mathbf{x}_t,t)$ in the parameterization in Eq. (66) to predict well-behaved output values, which should make training stable. This *mixed network parameterization* approach is inspired by the mixed score parameterization from Vahdat et al. [49] and Dockhorn et al. [60].

### C.2.4 Pseudocode

In this section, we provide pseudocode for training our prediction heads $\boldsymbol{k}_{\boldsymbol{\psi}}$ and using them for sampling with GENIE. In Alg. 1, the analytical $\frac{dt}{d\gamma_t}$ is an implicit hyperparameter of the DDM as it depends on $\alpha_t$. For our choice of $\alpha_t = e^{-\frac{1}{2}\int_0^t 0.1+19.9t'\,dt'}$ (see App. C.1), we have

$$\frac{dt}{d\gamma_t} = \frac{\frac{2\gamma_t}{19.9(\gamma_t^2+1)}}{\sqrt{\left(\frac{0.1}{19.9}\right)^2 + \frac{2\log(\gamma_t^2+1)}{19.9}}}, \tag{67}$$

where $\gamma_t = \sqrt{\frac{1-\alpha_t^2}{\alpha_t^2}}$.

In Alg. 2, we are free to use any time discretization $t_0 = 1.0 > t_1 > \cdots > t_N = t_{\text{cutoff}}$. When referring to "linear striding" in this work, we mean the time discretization $t_n = 1.0 - (1.0 - t_{\text{cutoff}}) \frac{n}{N}$. When referring to "quadratic striding" in this work, we mean the time discretization $t_n = \left(1.0 - (1.0 - \sqrt{t_{\text{cutoff}}}) \frac{n}{N}\right)^2$.

---

**Algorithm 1** Training prediction heads $\boldsymbol{k_\psi}$

---

**Input:** Score model $\boldsymbol{s_\theta} := -\frac{\boldsymbol{\epsilon_\theta}(\mathbf{x}_t, t)}{\sigma_t}$, number of training iterations $N$.
**Output:** Trained prediction head $\boldsymbol{k_\psi}$.

**for** $n = 1$ **to** $N$ **do**
    Sample $\mathbf{x}_0 \sim p_0(\mathbf{x}_0)$, $t \sim \mathcal{U}[t_{\text{cutoff}}, 1]$, $\boldsymbol{\epsilon} \sim \mathcal{N}(\mathbf{0}, \boldsymbol{I})$
    Set $\mathbf{x}_t = \alpha_t \mathbf{x}_0 + \sigma_t \boldsymbol{\epsilon}$
    Compute $\boldsymbol{\epsilon_\theta}(\mathbf{x}_t, t)$
    Compute the exact spatial Jacobian-vector product $\text{JVP}_s = \frac{\partial \boldsymbol{\epsilon_\theta}(\mathbf{x}_t, t)}{\partial \mathbf{x}_t} \left( \frac{1}{\sqrt{\gamma_t^2 + 1}} \boldsymbol{\epsilon_\theta}(\mathbf{x}_t, t) - \frac{\gamma_t}{1 + \gamma_t^2} \mathbf{x}_t \right)$
via automatic differentiation
    Compute the exact temporal Jacobian-vector product $\text{JVP}_t = \frac{\partial \boldsymbol{\epsilon_\theta}(\mathbf{x}_t, t)}{\partial t} \frac{dt}{d\gamma_t}$ via automatic differentiation
($\frac{dt}{d\gamma_t}$ can be computed analytically)
    Compute $\boldsymbol{k_\psi}(\mathbf{x}_t, t)$ using the mixed parameterization in Eq. (66)
    Update weights $\boldsymbol{\psi}$ to minimize $\gamma_t^2 \|\boldsymbol{k_\psi}(\mathbf{x}_t, t) - d_{\gamma_t} \boldsymbol{\epsilon_\theta}(\mathbf{x}_t, t)\|_2^2$, where $d_{\gamma_t} \boldsymbol{\epsilon_\theta}(\mathbf{x}_t, t) = \text{JVP}_s - \text{JVP}_t$
**end for**

---

**Algorithm 2** GENIE sampling

---

**Input:** Score model $\boldsymbol{s_\theta} := -\frac{\boldsymbol{\epsilon_\theta}(\mathbf{x}_t, t)}{\sigma_t}$, prediction head $\boldsymbol{k_\psi}$, number of sampler steps $N$, time discretization $\{t_n\}_{n=0}^N$.
**Output:** Generated GENIE output sample $\mathbf{y}$.
Sample $\mathbf{x}_{t_0} \sim \mathcal{N}(\mathbf{0}, \boldsymbol{I})$
Set $\bar{\mathbf{x}}_{t_0} = \sqrt{1 + \gamma_{t_0}^2} \mathbf{x}_{t_0}$                 ▷ *Note that* $\bar{\mathbf{x}}_{t_n} = \sqrt{1 + \gamma_{t_n}^2} \mathbf{x}_{t_n}$ *for all* $t_n$
**for** $n = 0$ **to** $N - 1$ **do**
    **if** AFS and $n = 0$ **then**
        $\bar{\mathbf{x}}_{t_{n+1}} = \bar{\mathbf{x}}_{t_n} + (\gamma_{t_{n+1}} - \gamma_{t_n}) \mathbf{x}_{t_n}$
    **else**
        $\bar{\mathbf{x}}_{t_{n+1}} = \bar{\mathbf{x}}_{t_n} + (\gamma_{t_{n+1}} - \gamma_{t_n}) \boldsymbol{\epsilon_\theta}(\mathbf{x}_{t_n}, t_n) + \frac{1}{2}(\gamma_{t_{n+1}} - \gamma_{t_n})^2 \boldsymbol{k_\psi}(\mathbf{x}_{t_n}, t_n)$
    **end if**
    $\mathbf{x}_{t_{n+1}} = \frac{\bar{\mathbf{x}}_{t_{n+1}}}{\sqrt{1 + \gamma_{t_{n+1}}^2}}$
**end for**
**if** Denoising **then**
    $\mathbf{y} = \frac{\mathbf{x}_{t_N} - \sigma_{t_N} \boldsymbol{\epsilon_\theta}(\mathbf{x}_{t_N}, t_N)}{\alpha_{t_N}}$
**else**
    $\mathbf{y} = \mathbf{x}_{t_N}$
**end if**

---

### C.2.5   Measuring Computational Overhead

Our prediction heads induce a slight computational overhead since their forward pass has to occur after the forward pass of the score model. We measure the overhead as follows: first, we measure the inference time of the score model itself. We do five forward passes to "warm-up" the model and then subsequently synchronize via `torch.cuda.synchronize()`. We then measure the total wall-clock time of 50 forward passes. We then repeat this process using a combined forward pass: first the score model and subsequently the prediction head. We choose the batch size to (almost) fill the entire GPU memory. In particular we chose batch sizes of 512, 128, 128, 64, 64, and 8, for CIFAR-10, LSUN Bedrooms, LSUN Church-Outdoor, ImageNet, Cats (base), and Cats (upsampler), respectively. The computational overhead for each model is reported in Tab. 5. This measurement was carried out on a single NVIDIA 3080 Ti GPU.

# D  Learning Higher-Order Gradients without Automatic Differentiation and Distillation

In this work, we learn the derivative $d_{\gamma_t}\boldsymbol{\epsilon_\theta}$, which includes a spatial and a temporal Jacobian-vector product, by distillation based on automatic differentiation (AD). We now derive an alternative learning objective for the spatial Jacobian-vector product (JVP) which does not require any AD. We start with the following (conditional) expectation

$$\mathbb{E}\left[\alpha_t^2\mathbf{x}_0\mathbf{x}_t^\top - \alpha_t\left[\mathbf{x}_0\mathbf{x}_t^\top + \mathbf{x}_t\mathbf{x}_0^\top\right] \mid \mathbf{x}_t, t\right] = -\mathbf{x}_t\mathbf{x}_t^\top + \sigma_t^4\boldsymbol{S}_2(\mathbf{x}_t, t) + \sigma_t^4\boldsymbol{s}_1(\mathbf{x}_t, t)\boldsymbol{s}_1(\mathbf{x}_t, t)^\top + \sigma_t^2\boldsymbol{I}, \tag{68}$$

where $\boldsymbol{s}_1(\mathbf{x}_t, t) := \nabla_{\mathbf{x}_t}\log p_t(\mathbf{x}_t)$ and $\boldsymbol{S}_2(\mathbf{x}_t, t) := \nabla_{\mathbf{x}_t}^\top\nabla_{\mathbf{x}_t}\log p_t(\mathbf{x}_t)$. The above formula is derived in Meng et al. [Theorem 1, 82]. Adding $\mathbf{x}_t\mathbf{x}_t^\top$ to Eq. (68) and subsequently dividing by $\sigma_t^2$, we have

$$\mathbb{E}\left[\frac{\alpha_t^2}{\sigma_t^2}\mathbf{x}_0\mathbf{x}_t^\top - \frac{\alpha_t}{\sigma_t^2}\left[\mathbf{x}_0\mathbf{x}_t^\top + \mathbf{x}_t\mathbf{x}_0^\top\right] + \frac{1}{\sigma_t^2}\mathbf{x}_t\mathbf{x}_t^\top \mid \mathbf{x}_t, t\right] = \sigma_t^2\boldsymbol{S}_2(\mathbf{x}_t, t) + \sigma_t^2\boldsymbol{s}_1(\mathbf{x}_t, t)\boldsymbol{s}_1(\mathbf{x}_t, t)^\top + \boldsymbol{I}, \tag{69}$$

where we could pull the $\frac{1}{\sigma_t^2}\mathbf{x}_t\mathbf{x}_t^\top$ term into the expectation because it is conditioned on $t$ and $\mathbf{x}_t$. Using $\mathbf{x}_t = \alpha_t\mathbf{x}_0 + \sigma_t\boldsymbol{\epsilon}$, we can rewrite the above as

$$\mathbb{E}\left[\boldsymbol{\epsilon}\boldsymbol{\epsilon}^\top \mid \mathbf{x}_t, t\right] = \sigma_t^2\boldsymbol{S}_2(\mathbf{x}_t, t) + \sigma_t^2\boldsymbol{s}_1(\mathbf{x}_t, t)\boldsymbol{s}_1(\mathbf{x}_t, t)^\top + \boldsymbol{I}. \tag{70}$$

For an arbitrary $\boldsymbol{v} := \boldsymbol{v}(\mathbf{x}_t, t)$, we then have

$$\mathbb{E}\left[\boldsymbol{\epsilon}\boldsymbol{\epsilon}^\top\boldsymbol{v} \mid \mathbf{x}_t, t\right] = \sigma_t^2\boldsymbol{S}_2(\mathbf{x}_t, t)\boldsymbol{v} + \sigma_t^2\boldsymbol{s}_1(\mathbf{x}_t, t)\boldsymbol{s}_1(\mathbf{x}_t, t)^\top\boldsymbol{v} + \boldsymbol{v}. \tag{71}$$

Therefore, we can develop a score matching-like learning objective for the (general) spatial JVP $\boldsymbol{o_\theta}(\mathbf{x}_t, t) \approx \boldsymbol{S}_2(\mathbf{x}_t, t)\boldsymbol{v}$ as

$$\mathbb{E}_{t\sim\mathcal{U}[t_{\text{cutoff}}, 1], \mathbf{x}_0\sim p(\mathbf{x}_0), \boldsymbol{\epsilon}\sim\mathcal{N}(\boldsymbol{0}, \boldsymbol{I})}\left[g_{\text{no-ad}}(t)\|\boldsymbol{o_\theta}(\mathbf{x}_t, t) + \boldsymbol{s_\theta}(\mathbf{x}_t, t)\boldsymbol{s_\theta}(\mathbf{x}_t, t)^\top\boldsymbol{v} + \frac{1}{\sigma_t^2}\boldsymbol{v} - \boldsymbol{\epsilon}\boldsymbol{\epsilon}^\top\boldsymbol{v}\|_2^2\right], \tag{72}$$

for some weighting function $g_{\text{no-ad}}(t)$. Setting $\boldsymbol{v}(\mathbf{x}_t, t) = -\sigma_t\left(\frac{1}{\sqrt{\gamma_t^2+1}}\boldsymbol{\epsilon_\theta}(\mathbf{x}_t, t) - \frac{\gamma_t}{1+\gamma_t^2}\mathbf{x}_t\right)$, would recover the spatial JVP needed for the computation of $d_{\gamma_t}\boldsymbol{\epsilon}$. In the initial phase of this project, we briefly experimented with learning the spatial JVP using this approach; however, we found that our distillation approach worked significantly better.

# E  Toy Experiments

For all toy experiments in Sec. 3, we consider the following ground truth distribution:

$$p_0(\mathbf{x}_0) = \frac{1}{8} \sum_{i=1}^{8} p_0^{(i)}(\mathbf{x}_0), \tag{73}$$

where

$$p_0^{(i)}(\mathbf{x}_0) = \frac{1}{8} \sum_{j=1}^{8} \mathcal{N}(\mathbf{x}_0, s_1 \boldsymbol{\mu}_i + s_1 s_2 \boldsymbol{\mu}_j, \sigma^2 \boldsymbol{I}). \tag{74}$$

We set $\sigma = 10^{-2}$, $s_1 = 0.9$, $s_2 = 0.2$, and

$$\boldsymbol{\mu}_1 = \begin{pmatrix} 1 \\ 0 \end{pmatrix}, \qquad \boldsymbol{\mu}_2 = \begin{pmatrix} -1 \\ 0 \end{pmatrix}, \qquad \boldsymbol{\mu}_3 = \begin{pmatrix} 0 \\ 1 \end{pmatrix}, \qquad \boldsymbol{\mu}_4 = \begin{pmatrix} 0 \\ -1 \end{pmatrix}$$

$$\boldsymbol{\mu}_5 = \begin{pmatrix} \frac{1}{\sqrt{2}} \\ \frac{1}{\sqrt{2}} \end{pmatrix}, \qquad \boldsymbol{\mu}_6 = \begin{pmatrix} \frac{1}{\sqrt{2}} \\ -\frac{1}{\sqrt{2}} \end{pmatrix}, \qquad \boldsymbol{\mu}_7 = \begin{pmatrix} -\frac{1}{\sqrt{2}} \\ \frac{1}{\sqrt{2}} \end{pmatrix}, \qquad \boldsymbol{\mu}_8 = \begin{pmatrix} -\frac{1}{\sqrt{2}} \\ -\frac{1}{\sqrt{2}} \end{pmatrix}.$$

The ground truth distribution is visualized in Fig. 2a. Note that we can compute the score functions (and all its derivatives) analytically for Gaussian mixture distributions.

In Fig. 2, we compared DDIM to GENIE for sampling using the analytical score function of the ground truth distribution with 25 solver steps. In Fig. 12, we repeated this experiment for 5, 10, 15, and 20 solver steps. We found that in particular for $n = 10$ both solvers generate samples in interesting patterns.

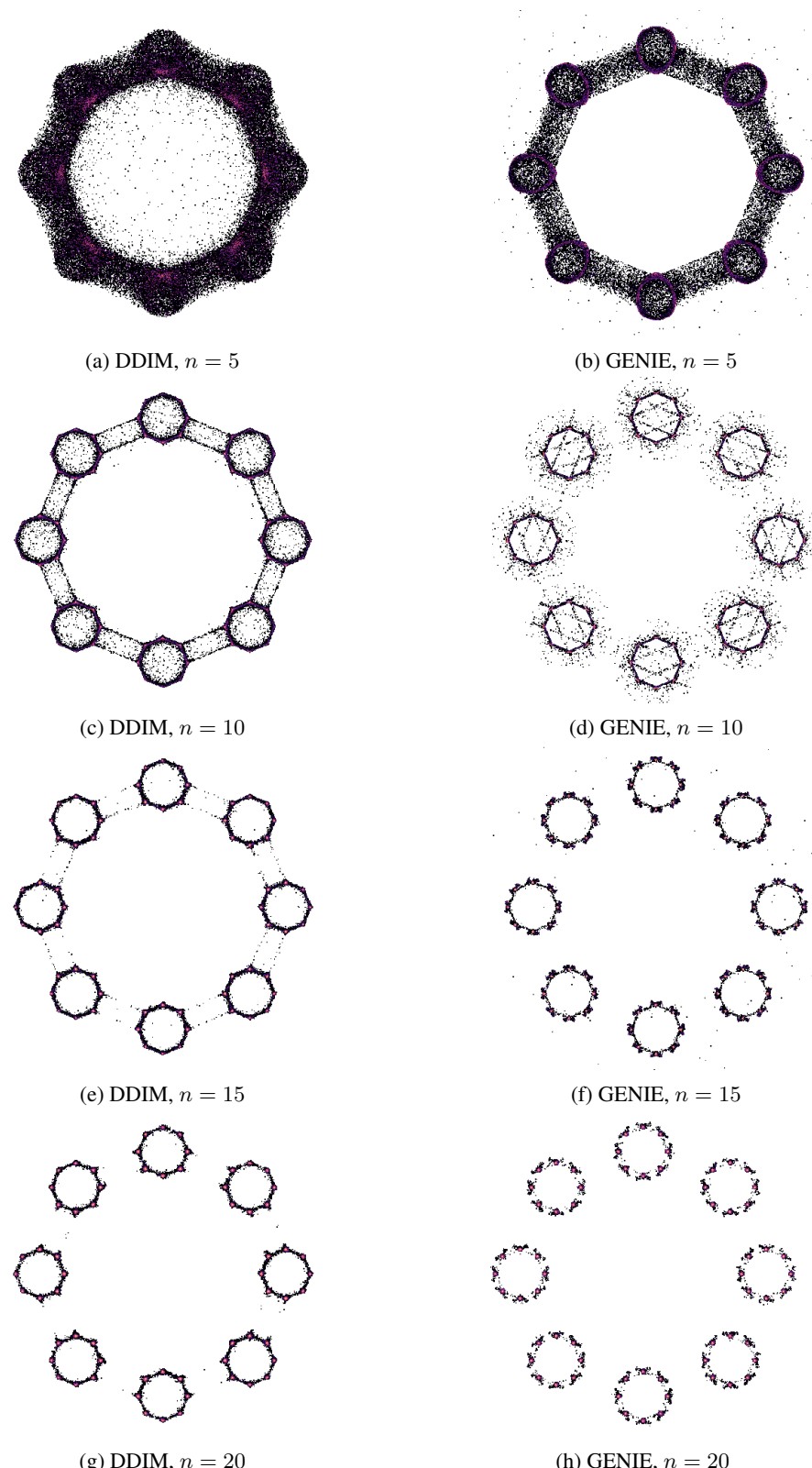

(a) DDIM, $n = 5$

(b) GENIE, $n = 5$

(c) DDIM, $n = 10$

(d) GENIE, $n = 10$

(e) DDIM, $n = 15$

(f) GENIE, $n = 15$

(g) DDIM, $n = 20$

(h) GENIE, $n = 20$

Figure 12: Modeling a complex 2D toy distribution: Samples are generated with DDIM and GENIE with $n$ solver steps using the analytical score function of the ground truth distribution (visualized in Fig. 2a). Zoom in for details.

## F   Image Experiments

### F.1   Evaluation Metrics, Baselines, and Datasets

**Metrics:** We quantitatively measure sample quality via Fréchet Inception Distance [FID, 102]. It is common practice to use 50k samples from the training set for reference statistics. We follow this practice for all datasets except for ImageNet and Cats. For ImageNet, we follow Dhariwal and Nichol [4] and use the entire training set for reference statistics. For the small Cats dataset, we use the training as well as the validation set for reference statistics.

**Baselines:** We run baseline experiments using two publicly available repositories. The `score_sde_pytorch` repository is licensed according to the Apache License 2.0; see also their license file here. The `CLD-SGM` repository is licensed according to the NVIDIA Source Code License; see also their license file here.

**Datasets:** We link here the websites of the datasets used in this experiment: CIFAR-10, LSUN datasets, ImageNet, and AFHQv2.

### F.2   Analytical First Step (AFS)

The forward process of DDMs generally converges to an analytical distribution. This analytical distribution is then used to sample from DDMs, defining the initial condition for the generative ODE/SDE. For example, for variance-preserving DDMs, we have $p_1(\mathbf{x}_1) \approx \mathcal{N}(\mathbf{x}_1; \mathbf{0}, \boldsymbol{I})$.

In this work, we try to minimize the computational complexity of sampling from DDMs, and therefore operate in a low NFE regime. In this regime, every additional function evaluation makes a significant difference. We therefore experimented with replacing the learned score with the (analytical score) of $\mathcal{N}(\mathbf{0}, \boldsymbol{I}) \approx p_1(\mathbf{x}_1)$ in the first step of the ODE solver. This "gained" function evaluation can then be used as an additional step in the ODE solver later.

In particular, we have

$$\epsilon_{\boldsymbol{\theta}}(\mathbf{x}_1, 1) \approx \mathbf{x}_1, \tag{75}$$

and $\frac{d\epsilon_{\boldsymbol{\theta}}(\mathbf{x}_1,1)}{d\gamma_1} \approx \mathbf{0}$ as shown below:

$$\frac{d\epsilon_{\boldsymbol{\theta}}(\mathbf{x}_1, 1)}{d\gamma_1} \approx \frac{d\mathbf{x}_t}{d\gamma_t}\big|_{t=1} \tag{76}$$

$$= -\frac{\gamma_t}{\gamma_t^2 + 1} \left[ \mathbf{x}_t + \boldsymbol{s}_{\boldsymbol{\theta}}(\mathbf{x}_t, t) \right]\big|_{t=1} \quad \text{(using Eq. (28))} \tag{77}$$

$$\approx \mathbf{0} \quad \text{(using normal assumption } \boldsymbol{s}_{\boldsymbol{\theta}}(\mathbf{x}_t, t) \approx -\mathbf{x}_t) \tag{78}$$

Given this, the AFS step becomes identical to the Euler update that uses the Normal score function for $\mathbf{x}_1$. This step is shown in the pseudocode in Alg. 2.

### F.3   Classifier-Free Guidance

As discussed in Sec. 5.2, to guide diffusion sampling towards particular classes, we replace $\epsilon_{\boldsymbol{\theta}}(\mathbf{x}_t, t)$ with

$$\hat{\epsilon}_{\boldsymbol{\theta}}(\mathbf{x}_t, t, c, w) = (1 + w)\epsilon_{\boldsymbol{\theta}}(\mathbf{x}_t, t, c) - w\epsilon_{\boldsymbol{\theta}}(\mathbf{x}_t, t), \tag{79}$$

where $w > 1.0$ is the "guidance scale", in the DDIM ODE. We experiment with classifier-free guidance on ImageNet. In Eq. (79) we re-use the conditional ImageNet score model $\epsilon_{\boldsymbol{\theta}}(\mathbf{x}_t, t, c)$ trained before (see App. C.1 for details), and train an additional unconditional ImageNet score model $\epsilon_{\boldsymbol{\theta}}(\mathbf{x}_t, t)$ using the exact same setup (and simply setting the class embedding to zero). We also re-use the conditional prediction head trained on top of the conditional ImageNet score model and train an additional prediction head for the unconditional model. Note that for both the score models as well as the prediction heads, we could share parameters between the models to reduce computational complexity [70]. The modified GENIE scheme for classifier-free guidance is then given as

$$\bar{\mathbf{x}}_{t_{n+1}} = \bar{\mathbf{x}}_{t_n} + (\gamma_{t_{n+1}} - \gamma_{t_n})\hat{\epsilon}_{\boldsymbol{\theta}}(\mathbf{x}_{t_n}, t_n, c, w) + \frac{1}{2}(\gamma_{t_{n+1}} - \gamma_{t_n})^2 \hat{\boldsymbol{k}}_{\boldsymbol{\psi}}(\mathbf{x}_{t_n}, t_n, c, w), \tag{80}$$

where

$$\hat{\boldsymbol{k}}_{\boldsymbol{\psi}}(\mathbf{x}_{t_n}, t_n, c, w) = (1 + w)\boldsymbol{k}_{\boldsymbol{\psi}}(\mathbf{x}_{t_n}, t_n, c) - w\boldsymbol{k}_{\boldsymbol{\psi}}(\mathbf{x}_{t_n}, t_n). \tag{81}$$

### F.4 Encoding

To encode a data point $\mathbf{x}_0$ into latent space, we first "diffuse" the data point to $t = 10^{-3}$, i.e., $\mathbf{x}_t = \alpha_t \mathbf{x}_0 + \sigma_t \boldsymbol{\epsilon}$, $\boldsymbol{\epsilon} \sim \mathcal{N}(\mathbf{0}, \mathbf{I})$. We subsequently simulate the generative ODE (backwards) from $t = 10^{-3}$ to $t = 1$, obtaining the latent point $\mathbf{x}_1$.

To decode a latent point $\mathbf{x}_1$, we simulate the generative ODE (forwards) from $t = 1.0$ to $t = 10^{-3}$. We then denoise the data point, i.e., $\mathbf{x}_0 = \frac{\mathbf{x}_t - \sigma_t \boldsymbol{\epsilon}_{\boldsymbol{\theta}}(\mathbf{x}_t, t)}{\alpha_t}$. Note that denoising is generally optional to sample from DDMs; however, for our encoding-decoding experiment we always used denoising in the decoding part to match the inital "diffusion" in the encoding part.

### F.5 Latent Space Interpolation

We can use encoding to perform latent space interpolation of two data points $\mathbf{x}_0^{(0)}$ and $\mathbf{x}_0^{(1)}$. We first encode both data points, following the encoding setup from App. F.4, and obtain $\mathbf{x}_1^{(0)}$ and $\mathbf{x}_1^{(1)}$, respectively. We then perform spherical interpolation of the latent codes:

$$\mathbf{x}_1^{(b)} = \mathbf{x}_1^{(0)} \sqrt{1-b} + \mathbf{x}_2^{(1)} \sqrt{b}, \quad b \in [0, 1]. \tag{82}$$

Subsequently, we decode the latent code $\mathbf{x}_1^{(b)}$ following the decoding setup from App. F.4. In Fig. 13, we show latent space interpolations for LSUN Church-Outdoor and LSUN Bedrooms.

### F.6 Extended Quantitative Results

In this section, we show additional quantitative results not presented in the main paper. In particular, we show results for all four hyperparameter combinations (binary choice of AFS and binary choice of denoising) for methods evaluated by ourselves. For these methods (i.e., GENIE, DDIM, S-PNDM, F-PNDM, Euler–Maruyama), we follow the **Synthesis Strategy** outlined in Sec. 5, with the exception that we use linear striding instead of quadratic striding for S-PNDM [63] and F-PNDM [63]. To apply quadratic striding to these two methods, one would have to derive the Adams–Bashforth methods for non-constant step sizes which is beyond the scope of our work.

Results can be found in Tabs. 8 to 13. As expected, AFS can considerably improve results for almost all methods, in particular for NFEs $\leq 15$. Denoising, on the other hand, is more important for larger NFEs. For our Cats models, we initially found that denoising hurts performance, and therefore did not further test it in all settings.

**Recall Scores.** We quantify the sample diversity of GENIE and other fast samplers using the recall score [111]. In particular, we follow DDGAN [67] and use the improved recall score [112]; results on CIFAR-10 can be found in Tab. 6. As expected, we can see that for all methods recall scores suffer as the NFEs decrease. Compared to the baselines, GENIE achieves excellent recall scores, being on par with F-PNDM for NFE$\geq 15$. However, F-PNDM cannot be run for NFE$\leq 10$ (due to its additional Runge–Kutta warm-up iterations). Overall, these results confirm that GENIE offers strong sample diversity when compared to other common samplers using the same score model checkpoint.

**Striding Schedule Grid Search.** As discussed in Sec. 5 the fixed quadratic striding schedule (for choosing the times $t$ for evaluating the model during synthesis under fixed NFE budgets) used in GENIE may be sub-optimal, in particular for small NFEs. To explore this, we did a small grid search over three different striding schedules. As described in App. C.2.4, the quadratic striding schedule can be written as $t_n = \left(1.0 - (1.0 - \sqrt{t_{\text{cutoff}}}) \frac{n}{N}\right)^2$, and easily be generalized to

$$t_n = \left(1.0 - (1.0 - t_{\text{cutoff}}^{1/\rho}) \frac{n}{N}\right)^\rho, \rho > 1. \tag{83}$$

In particular, besides the quadratic schedule $\rho = 2$, we also tested the two additional values $\rho = 1.5$ and $\rho = 2.5$. We tested these schedules on GENIE as well as DDIM [58]; note that the other two comptetive baselines, S-PNDM [63] and F-PNDM [63], rely on linear striding, and therefore a grid search is not applicable. We show results for GENIE and DDIM in Tab. 7; for each combination of solver and NFE we applied the best synthesis strategy (whether or not we use denoising and/or the analytical first step) of quadratic striding ($\rho = 2.0$) also to $\rho = 1.5$ and $\rho = 2.5$. As can be seen in the table, $\rho = 1.5$ improves for both DDIM and GENIE for NFE=5 (over the quadratic schedule

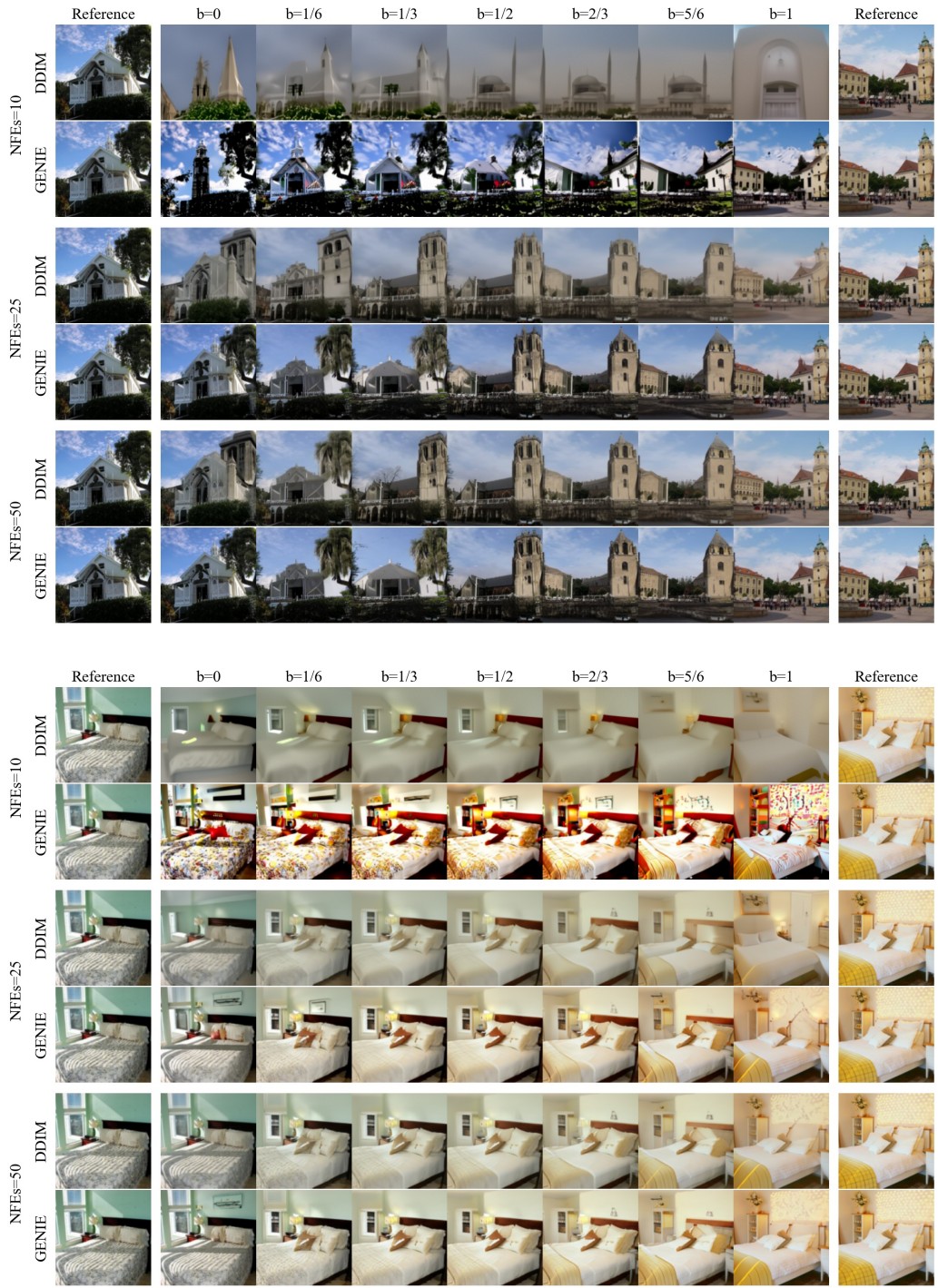

Figure 13: Latent space interpolations for LSUN Church-Outdoor (Top) and LSUN Bedrooms (Bottom). Note that $b = 0$ and $b = 1$ correspond to the decodings of the encoded reference images. Since this encode-decode loop is itself not perfect, the references are not perfectly reproduced at $b = 0$ and $b = 1$.

Table 6: Unconditional CIFAR-10 generative performance, measured in Recall (higher values are better). All methods use the same score model checkpoint.

| Method | AFS | Denoising | NFEs=5 | NFEs=10 | NFEs=15 | NFEs=20 | NFEs=25 |
|---|---|---|---|---|---|---|---|
| GENIE (ours) | ✗ | ✗ | 0.28 | 0.48 | 0.54 | 0.56 | 0.56 |
| | ✗ | ✓ | 0.21 | 0.45 | 0.52 | 0.56 | 0.57 |
| | ✓ | ✗ | 0.27 | 0.47 | 0.53 | 0.56 | 0.56 |
| | ✓ | ✓ | 0.19 | 0.46 | 0.53 | 0.55 | 0.56 |
| DDIM [58] | ✗ | ✗ | 0.10 | 0.27 | 0.38 | 0.43 | 0.46 |
| | ✗ | ✓ | 0.07 | 0.24 | 0.35 | 0.42 | 0.46 |
| | ✓ | ✗ | 0.08 | 0.27 | 0.38 | 0.43 | 0.46 |
| | ✓ | ✓ | 0.04 | 0.24 | 0.36 | 0.42 | 0.45 |
| S-PNDM [63] | ✗ | ✗ | 0.06 | 0.30 | 0.43 | 0.49 | 0.52 |
| | ✗ | ✓ | 0.02 | 0.25 | 0.39 | 0.46 | 0.50 |
| | ✓ | ✗ | 0.11 | 0.33 | 0.45 | 0.50 | 0.53 |
| | ✓ | ✓ | 0.06 | 0.29 | 0.41 | 0.47 | 0.51 |
| F-PNDM [63] | ✗ | ✗ | N/A | N/A | 0.55 | 0.57 | 0.58 |
| | ✗ | ✓ | N/A | N/A | 0.52 | 0.56 | 0.57 |
| | ✓ | ✗ | N/A | N/A | 0.55 | 0.58 | 0.59 |
| | ✓ | ✓ | N/A | N/A | 0.54 | 0.56 | 0.57 |
| Euler–Maruyama | ✗ | ✗ | 0.00 | 0.00 | 0.00 | 0.02 | 0.08 |
| | ✗ | ✓ | 0.00 | 0.00 | 0.00 | 0.03 | 0.06 |
| | ✓ | ✗ | 0.00 | 0.00 | 0.00 | 0.03 | 0.09 |
| | ✓ | ✓ | 0.00 | 0.00 | 0.00 | 0.03 | 0.09 |

Table 7: Unconditional CIFAR-10 generative performance (measured in FID) using our GENIE and DDIM [58] with different striding schedules using exponents $\rho \in \{1.5, 2.0, 2.5\}$.

| Method | $\rho$ | NFEs=5 | NFEs=10 | NFEs=15 | NFEs=20 | NFEs=25 |
|---|---|---|---|---|---|---|
| GENIE | 1.5 | **11.2** | **5.28** | 5.03 | 4.35 | 3.97 |
| | 2.0 | 13.9 | 5.97 | **4.49** | **3.94** | 3.67 |
| | 2.5 | 17.8 | 7.19 | 4.57 | **3.94** | **3.64** |
| DDIM | 1.5 | **27.6** | 13.5 | 8.97 | 7.20 | 6.15 |
| | 2.0 | 29.7 | **11.2** | **7.35** | **5.87** | **5.16** |
| | 2.5 | 33.2 | 13.4 | 8.28 | 6.36 | 5.39 |

$\rho = 2$), whereas larger $\rho$ are preferred for larger NFE. The improvement of GENIE from 13.9 to 11.2 FID for NFE=5 is significant.

**Discretization Errors of GENIE compared to other Fast Samplers.** We compute discretization errors, in particular local and global truncation errors, of GENIE and compare to existing faster solvers. We are using the CIFAR-10 model. We initially sample 100 latent vectors $\mathbf{x}_T \sim \mathcal{N}(\mathbf{0}, \mathbf{I})$ and then, starting from those latent vectors, synthesize 100 approximate ground truth trajectories (GTTs) using DDIM with 1k NFEs (for that many steps, the discretization error is negligible; hence, we can treat this as a pseudo ground truth).

We then synthesize 100 sample trajectories for DDIM [58], S-PNDM [63], F-PNDM [63], and GENIE (for NFEs=$\{5, 10, 15, 20, 25\}$, similar to the main experiments) using the same latent vectors as starting points that were used to generate the GTTs. DDIM, S-PNDM, and F-PNDM are training-free methods that can be run on the exact same score model, which also our GENIE relies on. Thereby, we are able to isolate discretization errors from errors in the learnt score function. We then compute the average $L_2$-distance (in Inception feature space [104]) between the output image of the fast samplers and the "output" of the pseudo GTT. As can be seen in Fig. 14, GENIE outperforms the three other methods on all NFEs.

Comparing the local truncation error (LTE) of different higher-order solvers can unfortunately not be done in a fair manner. Similar to DDIM, GENIE only needs the current value and a single NFE to predict the next step. In contrast, multistep methods rely on a history of predictions and Runge–Kutta methods rely on multiple NFEs to predict the next step. Thus, we can only fairly compare the LTE of

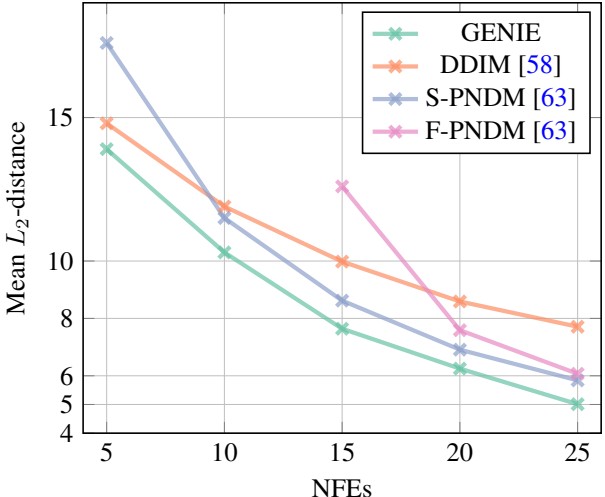

Figure 14: **Global Truncation Error:** $L_2$-distance of generated outputs by the fast samplers to the (approximate) ground truth (computed using DDIM with 1k NFEs) in Inception feature space [104]. Results are averaged over 100 samples.

GENIE to the LTE of DDIM. In particular, we compute LTEs at three starting times $t \in \{0.1, 0.2, .5\}$ (similar to what we did in Fig. 3). For each $t$, we then compare one step predictions for different step sizes $\Delta t$ against the ground truth trajectory ($L_2$-distance in data space averaged over 100 predictions; since we are not operating directly in image space at these intermediate $t$, using inception feature would not make sense here). As expected, we can see in Fig. 15 that GENIE has smaller LTE than DDIM for all starting times $t$.

## F.7   Extended Qualitative Results

In this section, we show additional qualitative comparisons of DDIM and GENIE on LSUN Church-Outdoor (Fig. 16), ImageNet (Fig. 17), and Cats (upsampler conditioned on test set images) (Fig. 18 and Fig. 19). In all figures, we can see that samples generated with GENIE generally exhibit finer details as well as sharper contrast and are less blurry compared to standard DDIM.

In Fig. 20 and Fig. 21, we show additional high-resolution images generated with the GENIE Cats upsampler using base model samples and test set samples, respectively.

## F.8   Computational Resources

The total amount of compute used in this research project is roughly 163k GPU hours. We used an in-house GPU cluster of V100 NVIDIA GPUs.

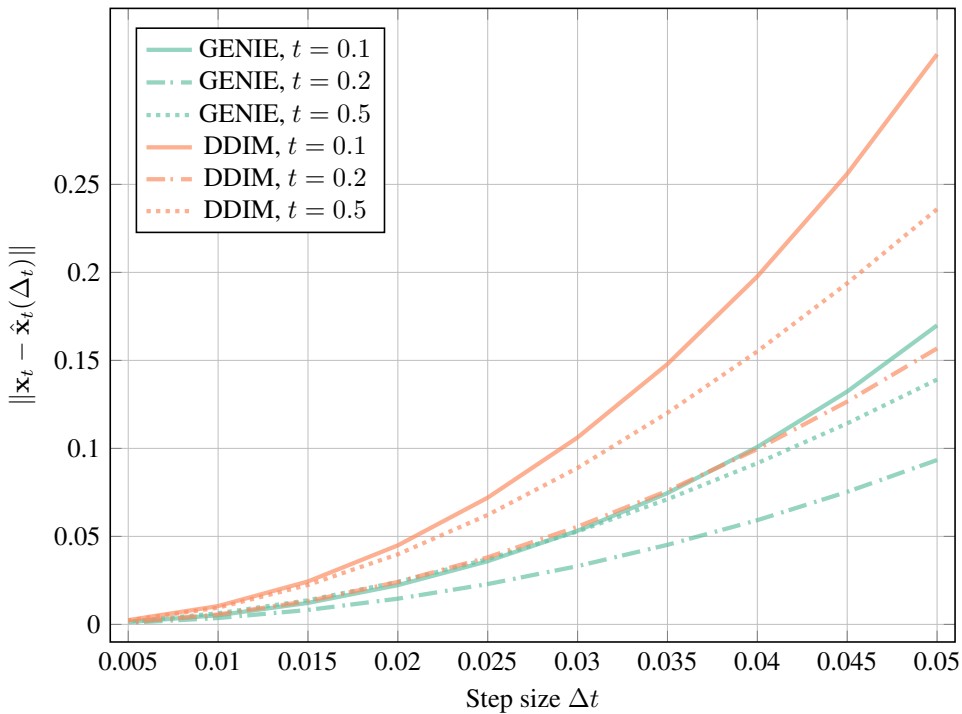

Figure 15: **Local Truncation Error:** Single step (local discretization) error, measured in $L_2$-distance to (approximate) ground truth (computed using DDIM with 1k NFEs) in data space and averaged over 100 samples, for GENIE and DDIM for three starting time points $t \in \{0.1, 0.2, 0.5\}$ (this is, the $t$ from which a small step with size $\Delta t$ is taken).

Table 8: Unconditional CIFAR-10 generative performance (measured in FID). Methods above the middle line use the same score model checkpoint; methods below all use different ones. (†): numbers are taken from literature. This table is an extension of Tab. 1.

| Method | AFS | Denoising | NFEs=5 | NFEs=10 | NFEs=15 | NFEs=20 | NFEs=25 |
|---|---|---|---|---|---|---|---|
| GENIE (ours) | ✗ | ✗ | 15.4 | **5.97** | 4.70 | 4.30 | 4.10 |
| | ✗ | ✓ | 23.5 | 6.91 | 4.74 | 4.02 | 3.72 |
| | ✓ | ✗ | 13.9 | 6.04 | 4.76 | 4.33 | 4.18 |
| | ✓ | ✓ | 17.9 | 6.27 | **4.49** | **3.94** | **3.67** |
| DDIM [58] | ✗ | ✗ | 30.1 | 11.6 | 7.56 | 6.00 | 5.27 |
| | ✗ | ✓ | 37.9 | 13.9 | 8.76 | 6.77 | 5.76 |
| | ✓ | ✗ | 29.7 | 11.2 | 7.35 | 5.87 | 5.16 |
| | ✓ | ✓ | 35.2 | 12.8 | 8.17 | 6.39 | 5.49 |
| S-PNDM [63] | ✗ | ✗ | 60.2 | 12.1 | 7.16 | 5.48 | 4.62 |
| | ✗ | ✓ | 101 | 17.2 | 10.8 | 8.74 | 7.62 |
| | ✓ | ✗ | 35.9 | 10.3 | 6.61 | 5.20 | 4.51 |
| | ✓ | ✓ | 56.8 | 14.9 | 10.2 | 8.37 | 7.35 |
| F-PNDM [63] | ✗ | ✗ | N/A | N/A | 12.1 | 6.58 | 4.89 |
| | ✗ | ✓ | N/A | N/A | 19.5 | 10.6 | 8.43 |
| | ✓ | ✗ | N/A | N/A | 10.3 | 5.96 | 4.73 |
| | ✓ | ✓ | N/A | N/A | 15.7 | 10.9 | 8.52 |
| Euler–Maruyama | ✗ | ✗ | 364 | 236 | 178 | 121 | 85.0 |
| | ✗ | ✓ | 391 | 235 | 191 | 129 | 89.9 |
| | ✓ | ✗ | 325 | 230 | 164 | 112 | 80.3 |
| | ✓ | ✓ | 364 | 235 | 176 | 120 | 83.6 |
| FastDDIM [64] (†) | ✗ | ✓ | - | 9.90 | - | 5.05 | - |
| Learned Sampler [66] (†) | ✗ | ✓ | **12.4** | 7.86 | 5.90 | 4.72 | 4.25 |
| Analytic DDIM (LS) [65] (†) | ✗ | ✓ | - | 14.0 | - | - | 5.71 |
| CLD-SGM [60] | ✗ | ✗ | 334 | 306 | 236 | 162 | 106 |
| VESDE-PC [57] | ✗ | ✓ | 461 | 461 | 461 | 461 | 462 |

Table 9: Conditional ImageNet generative performance (measured in FID).

| Method | AFS | Denoising | NFEs=5 | NFEs=10 | NFEs=15 | NFEs=20 | NFEs=25 |
|---|---|---|---|---|---|---|---|
| GENIE (ours) | ✗ | ✗ | 23.4 | 8.35 | 6.13 | 5.36 | 5.00 |
| | ✗ | ✓ | 35.4 | 7.59 | **5.23** | **4.48** | **4.13** |
| | ✓ | ✗ | 21.6 | 8.92 | 6.59 | 5.73 | 5.27 |
| | ✓ | ✓ | **20.2** | **7.41** | 5.36 | 4.68 | 4.27 |
| DDIM [58] | ✗ | ✗ | 39.0 | 14.5 | 9.47 | 7.57 | 6.64 |
| | ✗ | ✓ | 39.8 | 11.1 | 7.17 | 5.83 | 5.19 |
| | ✓ | ✗ | 37.4 | 14.7 | 9.73 | 7.86 | 6.92 |
| | ✓ | ✓ | 30.0 | 10.7 | 7.14 | 5.93 | 5.35 |
| S-PNDM [63] | ✗ | ✗ | 57.9 | 15.2 | 10.0 | 8.12 | 7.20 |
| | ✗ | ✓ | 60.6 | 12.2 | 8.69 | 7.59 | 6.94 |
| | ✓ | ✗ | 39.0 | 13.7 | 9.75 | 8.08 | 7.22 |
| | ✓ | ✓ | 35.5 | 11.2 | 8.54 | 7.52 | 6.94 |
| F-PNDM [63] | ✗ | ✗ | N/A | N/A | 13.9 | 9.45 | 7.87 |
| | ✗ | ✓ | N/A | N/A | 14.5 | 9.45 | 8.05 |
| | ✓ | ✗ | N/A | N/A | 12.5 | 9.01 | 7.74 |
| | ✓ | ✓ | N/A | N/A | 12.3 | 9.26 | 7.86 |

Table 10: Unconditional LSUN Bedrooms generative performance (measured in FID). Methods above the middle line use the same score model checkpoint; Learned Sampler uses a different one. (†): numbers are taken from literature.

| Method | AFS | Denoising | NFEs=5 | NFEs=10 | NFEs=15 | NFEs=20 | NFEs=25 |
|---|---|---|---|---|---|---|---|
| GENIE (ours) | ✗ | ✗ | 74.1 | 17.1 | 13.3 | 11.6 | 11.1 |
| | ✗ | ✓ | 115 | 11.4 | 7.18 | 5.80 | **5.35** |
| | ✓ | ✗ | 55.9 | 18.4 | 14.1 | 12.3 | 11.6 |
| | ✓ | ✓ | 47.3 | **9.29** | **6.83** | 5.79 | 5.40 |
| DDIM [58] | ✗ | ✗ | 69.6 | 27.1 | 19.0 | 15.8 | 14.2 |
| | ✗ | ✓ | 81.0 | 16.3 | 9.18 | 7.12 | 6.20 |
| | ✓ | ✗ | 62.1 | 27.1 | 19.3 | 16.3 | 14.6 |
| | ✓ | ✓ | 42.5 | 12.5 | 8.21 | 6.77 | 6.05 |
| S-PNDM [63] | ✗ | ✗ | 70.4 | 22.1 | 15.7 | 13.5 | 12.4 |
| | ✗ | ✓ | 88.9 | 12.2 | 8.40 | 7.33 | 6.80 |
| | ✓ | ✗ | 48.0 | 20.2 | 15.2 | 13.4 | 12.4 |
| | ✓ | ✓ | 45.0 | 10.8 | 8.14 | 7.23 | 6.71 |
| F-PNDM [63] | ✗ | ✗ | N/A | N/A | 36.1 | 18.5 | 14.6 |
| | ✗ | ✓ | N/A | N/A | 26.8 | 9.85 | 7.86 |
| | ✓ | ✗ | N/A | N/A | 29.4 | 17.5 | 14.3 |
| | ✓ | ✓ | N/A | N/A | 18.9 | 9.27 | 7.69 |
| Learned Sampler [66] (†) | ✗ | ✓ | **29.2** | 11.0 | - | **4.82** | - |

Table 11: Unconditional LSUN Church-Outdoor generative performance (measured in FID). Methods above the middle line use the same score model checkpoint; Learned Sampler uses a different one. (†): numbers are taken from literature.

| Method | AFS | Denoising | NFEs=5 | NFEs=10 | NFEs=15 | NFEs=20 | NFEs=25 |
|---|---|---|---|---|---|---|---|
| GENIE (ours) | ✗ | ✗ | 97.2 | 25.4 | 15.9 | 11.6 | 9.57 |
|  | ✗ | ✓ | 147 | 13.7 | 11.7 | 8.52 | 7.28 |
|  | ✓ | ✗ | 47.8 | 13.6 | 10.6 | 9.17 | 8.28 |
|  | ✓ | ✓ | 60.3 | **10.5** | **7.44** | **6.38** | **5.84** |
| DDIM [58] | ✗ | ✗ | 81.5 | 28.5 | 16.7 | 11.9 | 9.9 |
|  | ✗ | ✓ | 110 | 25.3 | 11.5 | 8.53 | 7.35 |
|  | ✓ | ✗ | 44.0 | 17.4 | 12.5 | 10.2 | 9.07 |
|  | ✓ | ✓ | 45.8 | 12.8 | 8.44 | 6.97 | 6.28 |
| S-PNDM [63] | ✗ | ✗ | 59.4 | 18.7 | 13.3 | 11.4 | 10.4 |
|  | ✗ | ✓ | 87.5 | 14.8 | 9.54 | 7.98 | 7.21 |
|  | ✓ | ✗ | 40.7 | 17.0 | 12.8 | 11.2 | 10.3 |
|  | ✓ | ✓ | 48.8 | 12.9 | 9.10 | 7.82 | 7.12 |
| F-PNDM [63] | ✗ | ✗ | N/A | N/A | 15.5 | 12.0 | 10.6 |
|  | ✗ | ✓ | N/A | N/A | 15.7 | 9.78 | 7.99 |
|  | ✓ | ✗ | N/A | N/A | 15.2 | 11.8 | 10.4 |
|  | ✓ | ✓ | N/A | N/A | 12.6 | 9.29 | 7.83 |
| Learned Sampler [66] (†) | ✗ | ✓ | **30.2** | 11.6 | - | 6.74 | - |

Table 12: Cats (base model) generative performance (measured in FID).

| Method | AFS | NFEs=10 | NFEs=15 | NFEs=20 | NFEs=25 |
|---|---|---|---|---|---|
| GENIE (ours) | ✗ | **12.2** | **8.74** | **7.40** | 6.84 |
|  | ✓ | 13.3 | 9.07 | 7.76 | **6.76** |
| DDIM [58] | ✗ | 12.7 | 9.89 | 8.66 | 7.98 |
|  | ✓ | 13.6 | 10.0 | 8.73 | 7.87 |
| S-PNDM [63] | ✗ | 12.8 | 11.6 | 10.8 | 10.4 |
|  | ✓ | 12.5 | 11.3 | 10.7 | 10.2 |
| F-PNDM [63] | ✗ | N/A | 12.8 | 10.4 | 10.6 |
|  | ✓ | N/A | 11.8 | 10.4 | 10.3 |

Table 13: Cats (upsampler) generative performance (measured in FID).

| Method | AFS | NFEs=5 | NFEs=10 | NFEs=15 |
|---|---|---|---|---|
| GENIE (ours) | ✗ | 7.03 | 4.93 | **4.83** |
|  | ✓ | **5.53** | **4.90** | 4.91 |
| DDIM [58] | ✗ | 11.3 | 7.16 | 5.99 |
|  | ✓ | 9.47 | 6.64 | 5.85 |
| S-PNDM [63] | ✗ | 16.7 | 12.1 | 8.83 |
|  | ✓ | 14.6 | 11.0 | 9.01 |
| F-PNDM [63] | ✗ | N/A | N/A | 12.9 |
|  | ✓ | N/A | N/A | 11.7 |

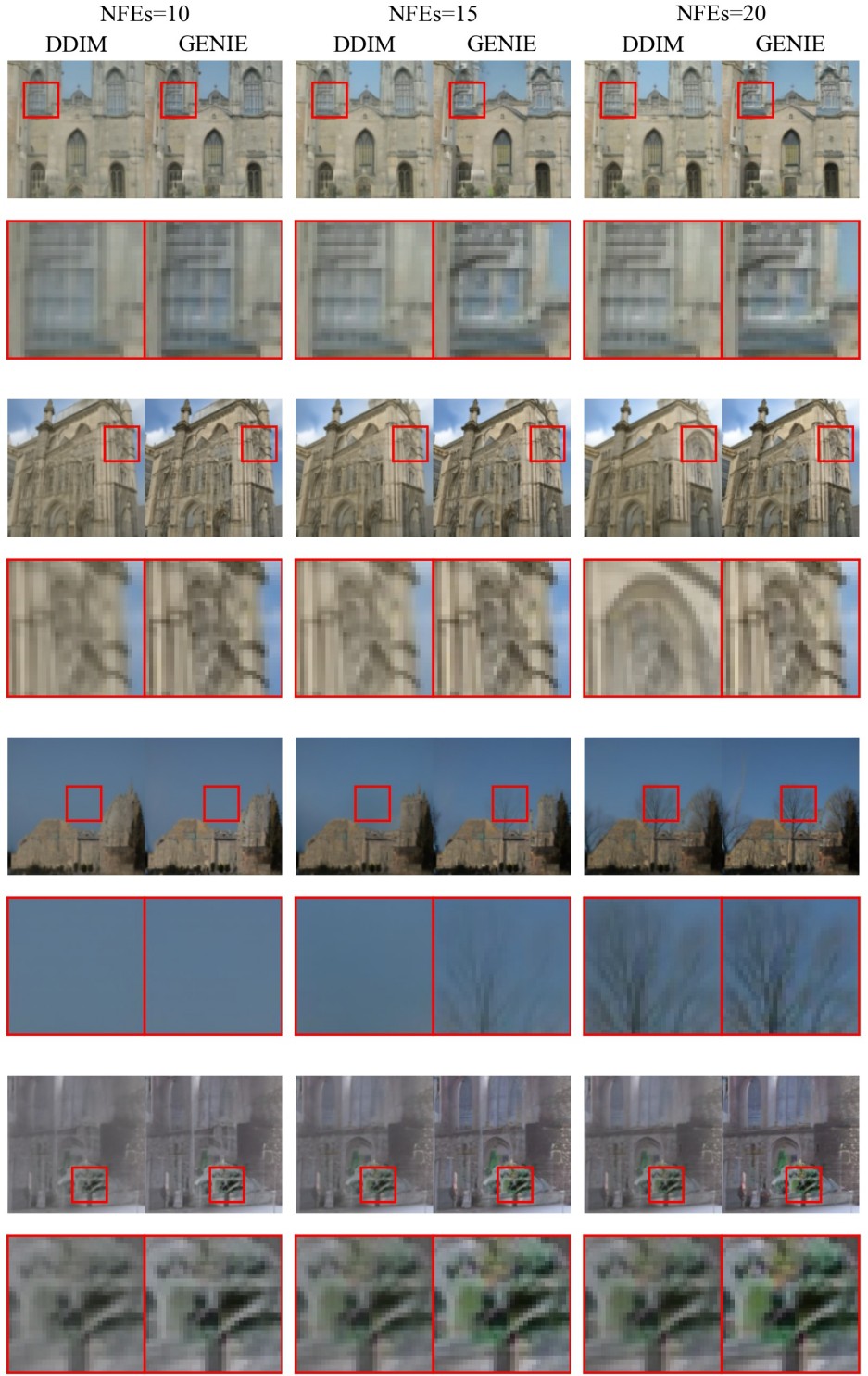

Figure 16: Additional samples on LSUN Church-Outdoor with zoom-in on details. GENIE often results in sharper and higher contrast samples compared to DDIM.

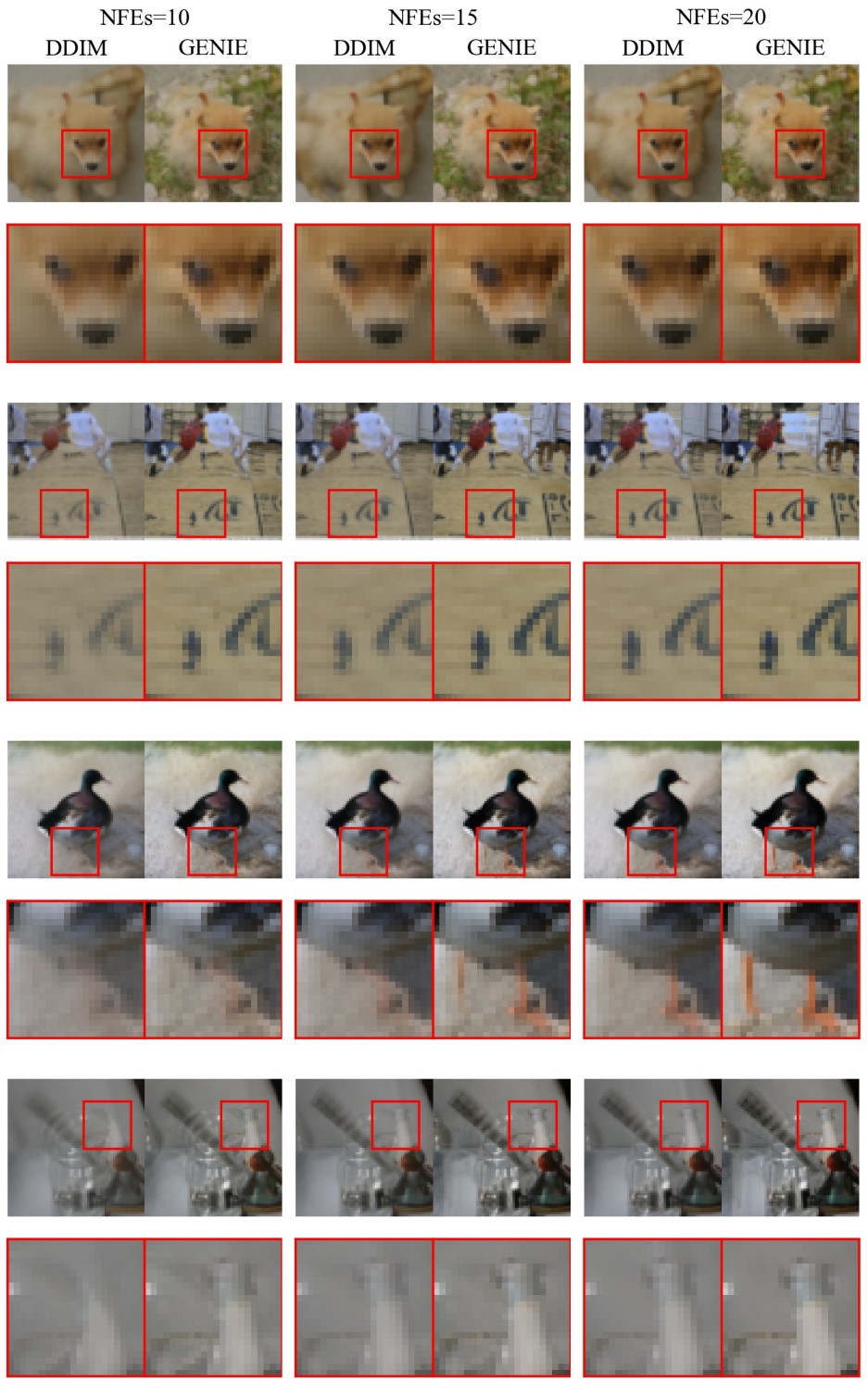

Figure 17: Additional samples on ImageNet with zoom-in on details. GENIE often results in sharper and higher contrast samples compared to DDIM.

DDIM GENIE

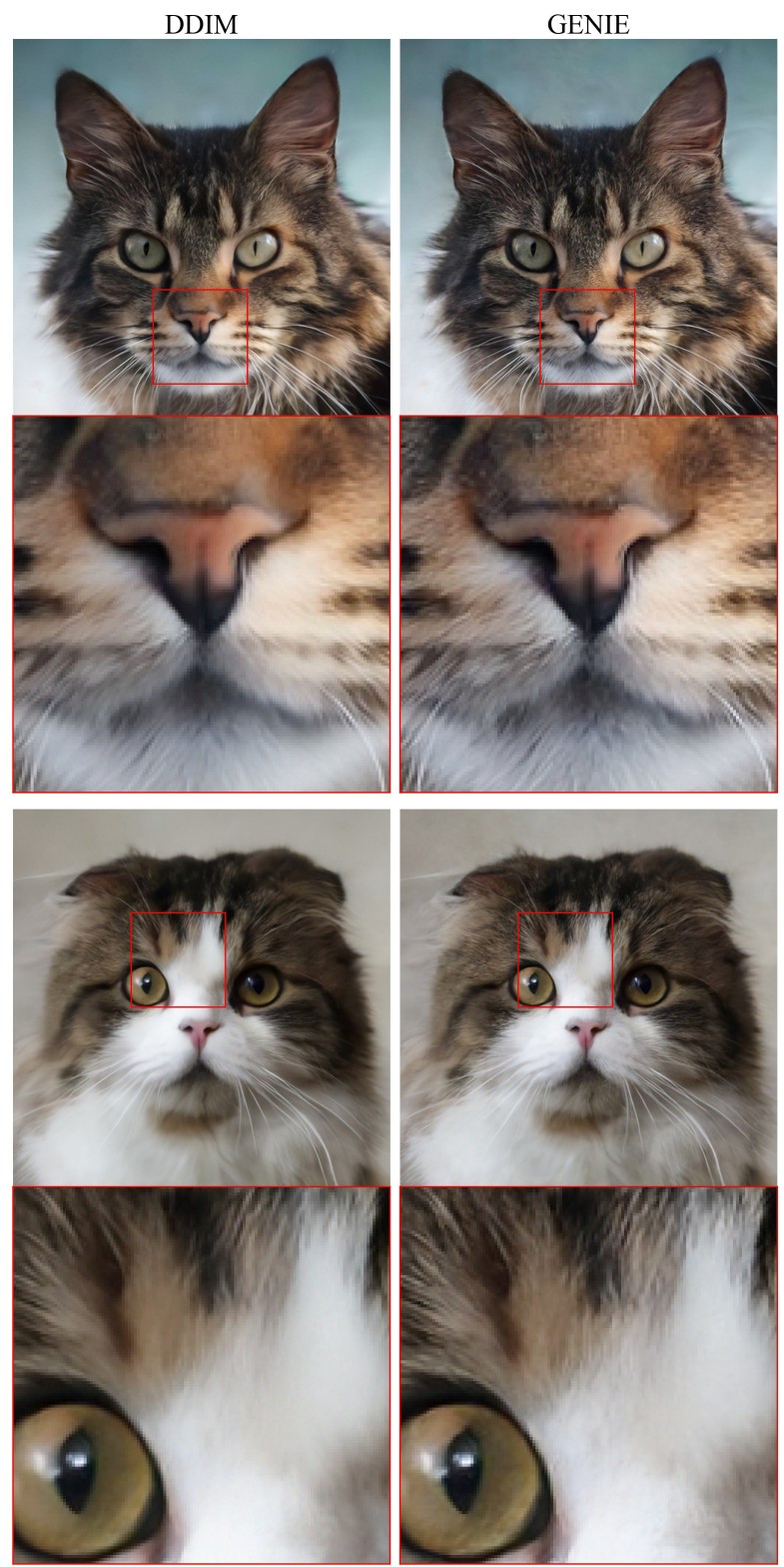

Figure 18: Additional samples on Cats with zoom-in on details. GENIE often results in sharper and higher contrast samples compared to DDIM.

DDIM  GENIE

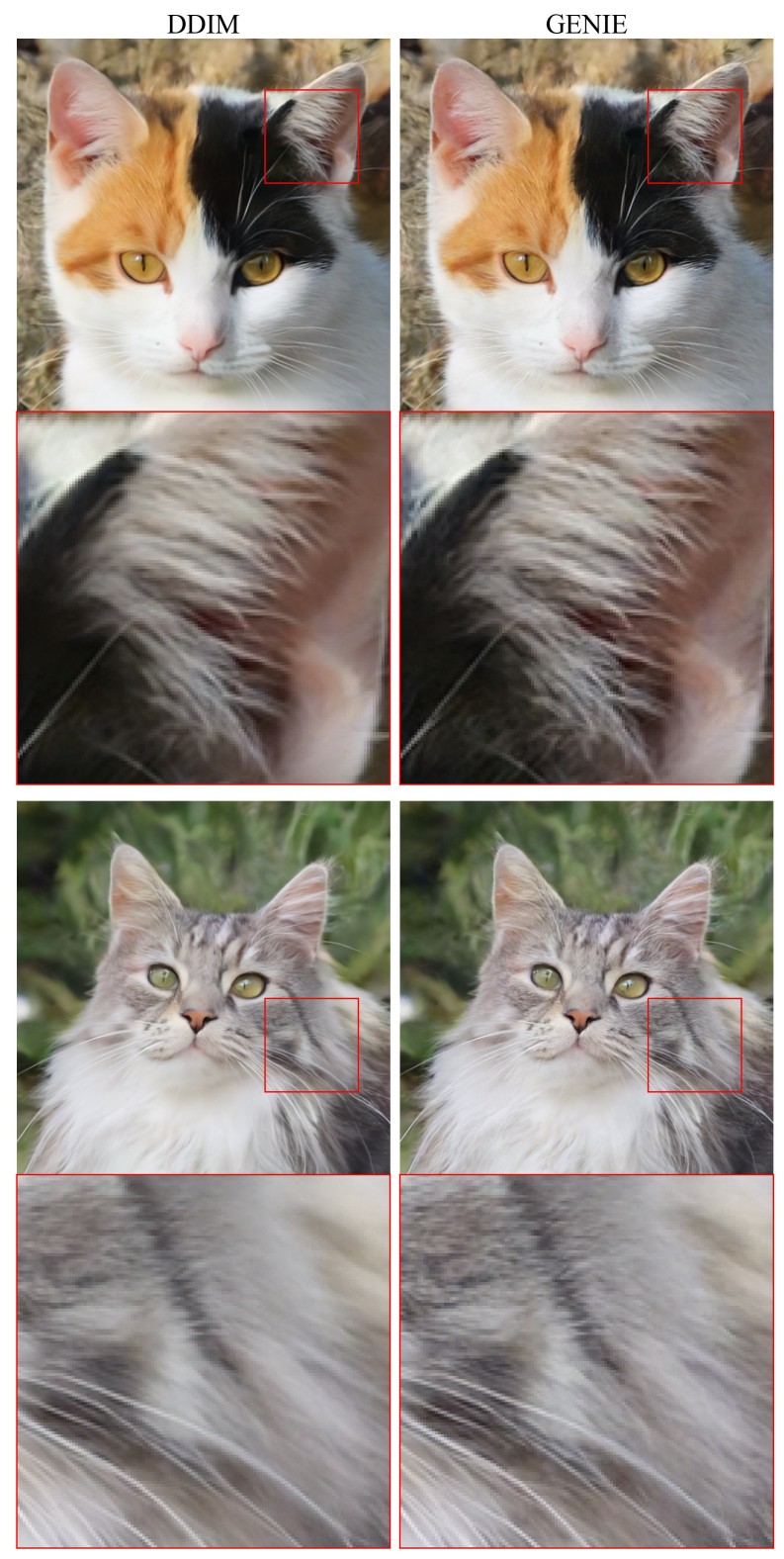

Figure 19: Additional samples on Cats with zoom-in on details. GENIE often results in sharper and higher contrast samples compared to DDIM.

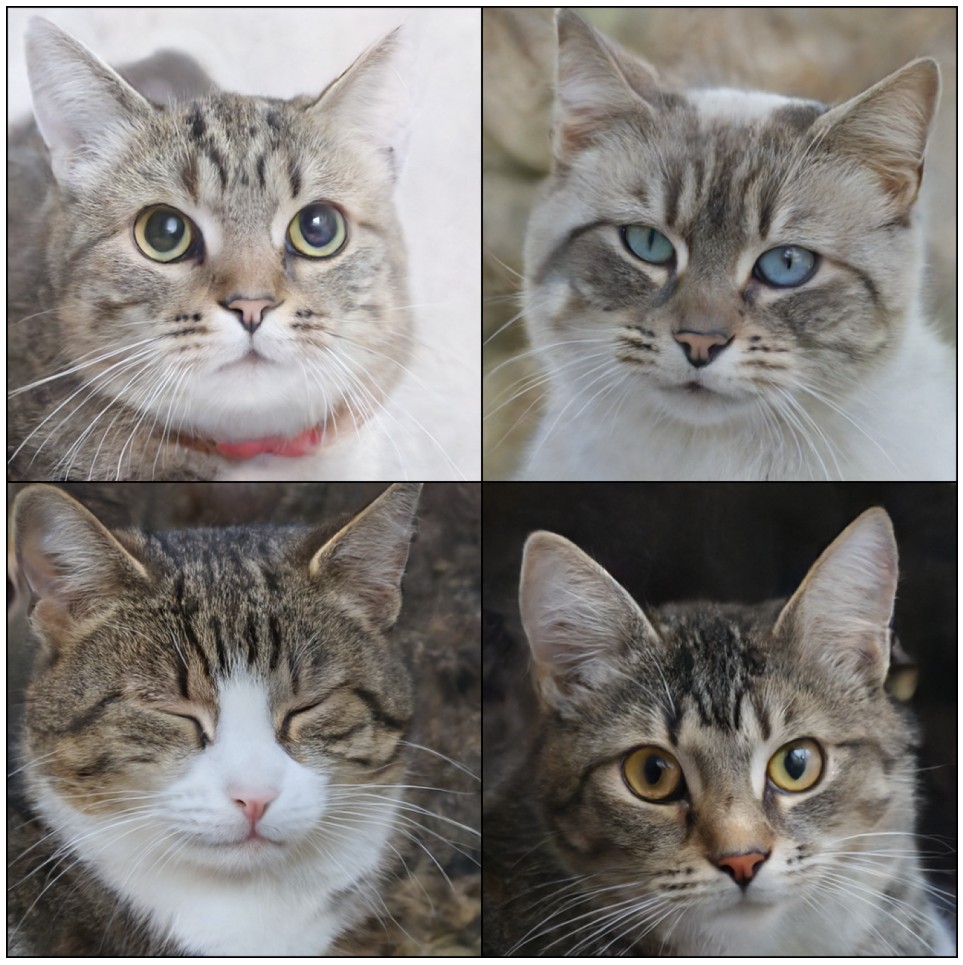

Figure 20: End-to-end samples on Cats. The GENIE base model uses 25 function evaluations and the GENIE upsampler only uses five function evaluations. An upsampler evaluation is roughly four times as expensive as a base model evaluation.

# G Miscellaneous

## G.1 Connection to Bao et al. [96]

The concurrent Bao et al. [96] learn covariance matrices for diffusion model sampling using prediction heads somewhat similar to the ones in GENIE. Specifically, both Bao et al. [96] and GENIE use small prediction heads that operate on top of the large first-order score predictor. However, we would like to stress multiple differences: (i) Bao et al. [96] learn the DDM's sampling covariance matrices, while we learn higher-order ODE gradients. More generally, Bao et al. [96] rely on stochastic diffusion model sampling, while we use the ODE formulation. (ii) Most importantly, in our case we can resort to directly learning the low-dimensional JVPs without low-rank or diagonal matrix approximations or other assumptions. Similar techniques are not directly applicable in Bao et al. [96]'s setting. In detail, this is because in their case the relevant matrices (obtained after Cholesky or another applicable decomposition of the covariance) do not act on regular vectors but random noise variables. In other words, instead of using a deterministic JVP predictor (which takes $\mathbf{x}_t$ and $t$ as inputs), as in GENIE, Bao et al. [96] would require to model an entire distribution for each $\mathbf{x}_t$ and $t$ without explicitly forming high-dimensional Cholesky decomposition-based matrices, if they wanted to do something somewhat analogous to GENIE's novel JVP-based approach. As a consequence, Bao et al. [96] take another route to keeping the dimensionality of the additional network outputs manageable in practice. In particular, they resort to assuming a diagonal covariance matrix in their experiments. By directly learning JVPs, we never have to rely on such potentially limiting assumptions. (iii) Experimentally,

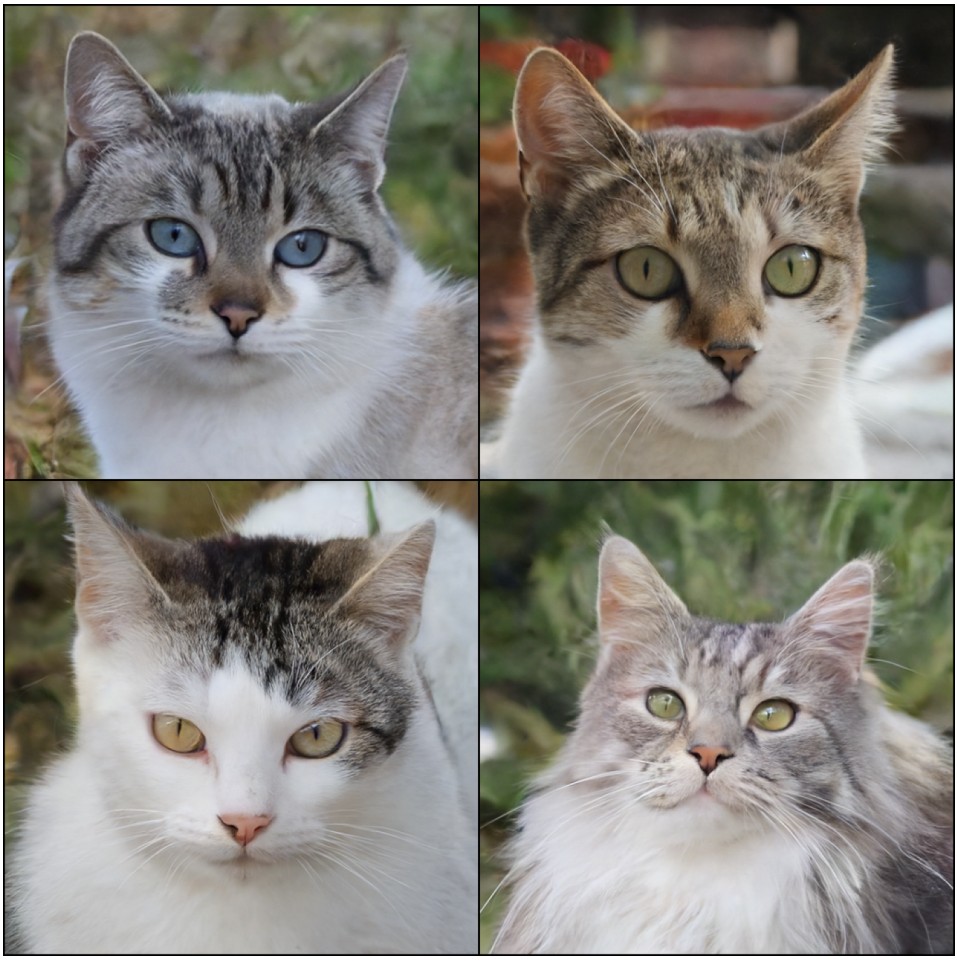

Figure 21: Upsampling $128 \times 128$ test set images using the GENIE upsampler with only five function evaluations.

Bao et al. [96] also consider fast sampling with few neural network calls. However, GENIE generally outperforms them (see, for example, their CIFAR10 results in their Table 2 for 10 and 25 NFE). This might indeed be due to the assumptions made by Bao et al. [96], which we avoid. Furthermore, their stochastic vs. our deterministic sampling may play a role, too.

### G.2 Combining GENIE with Progressive Distillation

We speculate that GENIE could potentially be combined with Progressive Distillation [69]: In every distillation stage of [69], one could quickly train a small GENIE prediction head to model higher-order ODE gradients. This would then allow for larger and/or more accurate steps, whose results represent the distillation target (teacher) in the progressive distillation protocol. This may also reduce the number of required distillation stages. Overall, this could potentially speed up the cumbersome stage-wise distillation and maybe also lead to an accuracy and performance improvement. In particular, we could replace the DDIM predictions in Algorithm 2 of [69] with improved GENIE predictions.

Note that this approach would not be possible with multistep methods as proposed by Liu et al. [63]. Such techniques could not be used here, because they require the history of previous predictions, which are not available in the progressive distillation training scheme.

We leave exploration of this direction to future work.