# OpenReview forum: "GENIE: Higher-Order Denoising Diffusion Solvers"
_NeurIPS.cc/2022/Conference — NeurIPS 2022 Accept_

### Official Review · Reviewer_bcTa · 2022-07-08

**Rating:** 7
**Confidence:** 4
**Soundness:** 3 good
**Presentation:** 3 good
**Contribution:** 3 good

**Summary:**

In this work, the authors proposed Higher-Order Denoising Diffusion Solvers (GENIE) using truncated Taylor methods,  which significantly accelerates image synthesis. The solver relies on higher-order score functions. To obtain the Jacobian-vector products (JVPs) for the higher-order score function, the authors proposed to extract them from the first-order score network via automatic differentiation and using a small additional head on top of the first-order score network.  The authors provide empirical  results to verify the advantages of the GENIE.

**Questions:**

It would be better to quantify the discretization error of the GENIE compared to the existing higher order methods.

**Ethics Review Area:**

["I don’t know"]

**Limitations:**

Still, the slow convergence is considered a limitation.  Aside from the higher-order score function, it would be better to consider other type of acceleration method for the reverse sampling procedure. I

**Strengths And Weaknesses:**

Strengths: The derivation of the higher-order score function using automatic differentiation and small additional head is novel. The empirical results are persuading.

Weakness:  The training scheme for the higher-order score function appears heuristic, which directly minimizes the differences. Given that the score training comes from more mathematical reasoning from the variational lower bound, the suggested heuristic derivation may need more theoretical justification from that aspect.

---

> ### Author Response · Authors · 2022-08-02
> **Thank you for questions and feedback (1)**
>
> We thank the reviewer for their thoughtful feedback. We are glad the reviewer recognizes the novelty of our work and believes our numerical results to be convincing. Below we address specific questions and comments.
>
> - **"The training scheme for the higher-order score function appears heuristic[...]"** We respectfully disagree with the reviewer in this point and would like to emphasize that GENIE is indeed a consistent and principled approach to developing a higher-order ODE solver for sampling from diffusion models.
>
>     GENIE's design consists of two parts: (1) We are building on the second Truncated Taylor Method (TTM), which is a well-studied ODE solver (see [59]) with provable local and global truncation errors (see also App. G.1). Therefore, if during inference we had access to the ground truth second-order ODE derivatives, which are required for the second TTM, GENIE would simply correspond to the *exact* second TTM.
>
>     (2) In principle, we could calculate the exact second-order derivatives *during inference* using automatic differentiation. However, this is too slow for competitive sampling speeds, as it requires additional backward passes through the first-order score network. Therefore, in practice, we use the learned prediction heads $\mathbf{k}_\psi(\mathbf{x}_t, t)$. Consequently, if $\mathbf{k}_\psi(\mathbf{x}_t, t)$ modeled the ground truth second-order derivatives exactly, i.e.  $\mathbf{k}_\psi(\mathbf{x}_t, t) = d\_{\gamma_t}{\mathbf{\epsilon}_\theta(\mathbf{x}_t, t)}$ for all $\mathbf{x}_t$ and $t$, we would obtain a rigorous second-order solver based on the TTM, following (1) above.
>
>     In practice, distillation will not be perfect. However, given the above analysis, optimizing a neural network $\mathbf{k}_\psi(\mathbf{x}_t, t)$ towards $d\_{\gamma_t}{\mathbf{\epsilon}_\theta(\mathbf{x}_t, t)}$ is well motivated and theoretically grounded. In particular, *during training* we are calculating *exact* ODE gradients using automatic differentiation on the first-order score model as distillation targets. Therefore, in the limit of infinite neural network capacity and perfect optimization, we could in theory minimize our distillation objective function Eq. (15) perfectly and obtain $\mathbf{k}_\psi(\mathbf{x}_t, t) = d\_{\gamma_t}{\mathbf{\epsilon}_\theta(\mathbf{x}_t, t)}$.
>
>     Also recall that regular denoising score matching itself, on which all diffusion models rely, follows the exact same argument. In particular, denoising score matching also minimizes a "simple" (weighted) $L_2$-loss between a trainable score model $\mathbf{s}_\theta(\mathbf{x}_t, t)$ and the spatial derivative of the log-perturbation kernel, i.e., $\nabla\_{\mathbf{x}_t} \log p_t(\mathbf{x}_t \mid \mathbf{x}_0)$. From this perspective, denoising score matching itself also simply tries to "distill" (spatial) derivatives into a model. If we perfectly optimized the denoising score matching objective, we would obtain a diffusion model that models the data distribution exactly, but in practice, similar to GENIE, we never achieve that due to imperfect optimization and finite-capacity neural networks. Nevertheless, denoising score matching similarly is a well-defined and principled method, precisely because of that theoretical limit in which the distribution can be reproduced exactly.
>
>     We would also like to point out that other, established higher-order methods for diffusion model sampling with the generative ODE, such as linear multistep methods [63], make approximations, too, which can be worse in fact. In particular, multistep methods *always* approximate higher-order derivatives in the TTM using finite differences which is crude for large step sizes, as can be seen in Fig. 3 (*bottom*). From this perspective, if our distillation is sufficiently accurate, GENIE can be expected to be more accurate than such multistep methods.
>
>     To clarify the reviewer's concern, we added this discussion in App. G.2, and we also slightly modified the paragraph **Learning Objective** in Sec. 3.

---

> > ### Author Response · Authors · 2022-08-02
> > **Thank you for questions and feedback (2)**
> >
> > - **"It would be better to quantify the discretization error of the GENIE compared to the existing higher order methods."** We follow the reviewer's suggestions and compute discretization errors (local and global truncation errors) on the CIFAR-10 model; see App. G.6. We initially sample 100 latent vectors $\mathbf{x}_T \sim \mathcal{N}(0, I)$ and then, starting from those latent vectors, synthesize 100 approximate ground truth trajectories (GTTs) using DDIM with 1k NFEs (for that many steps, the discretization error is negligible; hence, we can treat this as a pseudo ground truth).
> >
> >     We then synthesize 100 sample trajectories for DDIM [68], S-PNDM [63], F-PNDM [63], and GENIE (for NFEs=$\\{5, 10, 15, 20, 25\\}$, similar to the main experiments) using the same latent vectors as starting points that were used to generate the GTTs. DDIM, S-PNDM, and F-PNDM are training-free methods that can be run on the exact same score model, which also our GENIE relies on. Thereby, we are able to isolate discretization errors from errors in the learnt score function. We then compute the average $L_2$-distance (in Inception feature space [98]) between the output image of the fast samplers and the ``output'' of the pseudo GTT. As can be seen in Fig. 15, GENIE outperforms the three other methods on all NFEs.
> >
> >     Comparing the local truncation error (LTE) of different higher-order solvers can unfortunately not be done in a fair manner. Similar to DDIM, GENIE only needs the current value and a single NFE to predict the next step. In contrast, multistep methods rely on a history of predictions and Runge--Kutta methods rely on multiple NFEs to predict the next step. Thus, we can only fairly compare the LTE of GENIE to the LTE of DDIM. In particular, we compute LTEs at three starting times $t \in \\{0.1, 0.2, .5\\}$ (similar to what we did in Fig. 3). For each $t$, we then compare one step predictions for different step sizes $\Delta t$ against the ground truth trajectory ($L_2$-distance in data space averaged over 100 predictions; since we are not operating directly in image space at these intermediate $t$, using inception feature would not make sense here). As expected, we can see in Fig. 16 that GENIE has smaller LTE than DDIM for all starting times $t$.
> >
> > - **"Still, the slow convergence is considered a limitation. Aside from the higher-order score function, it would be better to consider other type of acceleration method for the reverse sampling procedure."** As stated in our conclusion, we agree with the reviewer that one limitation of GENIE is that it is still slightly slower than approaches that abandon the differential equation framework of DDMs altogether, such as [69]. This, however, comes at the considerable cost of preventing important applications such as guided sampling and image encoding (as we discussed in Secs. 4 and 5.2). To overcome this limitation, in future work we plan to leverage even higher-order gradients to accelerate rigorous ODE-based sampling from DDMs even further. Furthermore, one could consider combining higher-order score functions with multistep methods. We leave this exploration to future work. In an orthogonal direction, it might also be interesting to potentially combine brute-force fast sampling methods such as [69] with GENIE (we added a section on this in App. G.4).
> >
> > If our reply is satisfactory and the reviewer's concerns have been successfully addressed, we would like to kindly ask the reviewer to consider raising their score accordingly. Otherwise, we will be happy to further discuss.

---

> > > ### Comment · Reviewer_bcTa · 2022-08-06
> > > **Thanks for your detailed response**
> > >
> > > Your revision have properly addressed my concerns so I will rasie my rating accordingly.

---

### Official Review · Reviewer_LSQr · 2022-07-10

**Rating:** 6
**Confidence:** 4
**Soundness:** 3 good
**Presentation:** 3 good
**Contribution:** 3 good

**Summary:**

This paper proposes a high order method to solve the DDM ODE. To overcome the expansive computation of the JVP appeared in the high order term, this paper uses NN to distill the high order term. This paper also uses parameter sharing to avoid to much computation of NN. Experiments show GEINE has a good speed up performance.

**Questions:**



**Limitations:**



**Strengths And Weaknesses:**

Strengths:

Although the idea of high order solver already exists in diffusion models, this work further improves the performance of the high order solver. Therefore, the significance is sufficient in my opinion. Besides, this paper is also clearly written, and sufficient experiments are provided.


Weakness \& Questions:

1. How are the coefficients in Eq.(14) constructed? It does not directly match the coefficients in Eq.(12).

2. The $x_t$ in $d_{\gamma_t} \epsilon_\theta(x_t, t)$ follows the ODE. However, $x_t$ from $d_{\gamma_t} \epsilon_\theta(x_t, t)$ in Eq.(15) comes from $x_t = \alpha_t x_0 + \sigma_t \epsilon$. Does $d_{\gamma_t} \epsilon_\theta(x_t, t)$ in Eq.(15) actually denotes the function in the r.h.s. of Eq.(12), where $x_t$ can be arbitrary. If does, perhaps it would be better to use a new notation for the function in the r.h.s. of Eq.(12).

3. The author has mentioned prior high order solvers are suboptimal. The key difference that makes GENIE perform better can be discussed more. For example, both aiming to solve Eq.(8), why the NN approximation is better than other approximations in prior solvers?

4. I suggest to cite and discuss [1], which also learns an extra head at the top of the UNet to speed up sampling of diffusion models.


[1] Bao et al., Estimating the Optimal Covariance with Imperfect Mean in Diffusion Probabilistic Models

---

> ### Author Response · Authors · 2022-08-02
> **Thank you for questions and feedback (1)**
>
> We thank the reviewer for their thoughtful feedback. We are glad the reviewer recognizes our contribution as significant in the area of (higher-order) solvers for diffusion models. Below we address specific questions and comments.
>
> - **"How are the coefficients in Eq.~(14) constructed? [...]"** The coefficients in Eq. (14) are constructed by using the so-called *single data point* assumption: $p_t(\mathbf{x}_t) = \mathcal{N}(0, \sigma_t^2 I)$, which leads to $\mathbf{\epsilon}_\theta(\mathbf{x}_t, t) = \frac{\mathbf{x}_t}{\sigma_t}$. We refer to the derivation, which can be found in App. C.2.3, in the paragraph **Mixed Network Parameterization** of Sec. 3.1. However, we would like to briefly summarize the high-level intuitions here: We assume that the (diffused) data distribution is given by a single (perturbed) data point, in which case we can calculate the first-order score as well as the higher-order ODE gradient terms analytically along the entire diffusion process. After this analytic calculation, certain prefactors for the different terms in Eq. (12) appear. Even though in practice the empirical data distribution is complex, the *scale* of the distribution will be similar and hence using the emerging prefactors in front of the neural networks that parametrize the different higher-order terms in practice will ensure that the outputs of these neural nets all vary on well-behaved scales. This is discussed in detail in App. C.2.3. Furthermore, we show the superior numerical performance of this parametrization in Sec. 5.3.
>
> - **"[...] Does $d\_{\gamma_t} \mathbf{\epsilon}_\theta(\mathbf{x}_t, t)$ in Eq. (15) actually denotes the function in the r.h.s. of Eq. (12), where $\mathbf{x}_t$ can be arbitrary. [...]'"** This is an excellent question. The short answer is yes, they are the same, but a more nuanced reply: While Eq. (12) is a general expression that is theoretically valid for any $\mathbf{x}_t$, in practice these $\mathbf{x}_t$ correspond to those $\mathbf{x}_t$ that are encountered while sampling from the diffusion model with the generative ODE (Eq. (6)). The ODE trajectories traverse the space of the diffused data distribution. Therefore, we seek to model $d\_{\gamma_t} \mathbf{\epsilon}_\theta(\mathbf{x}_t, t)$ for all possible $\mathbf{x}_t$ from the diffused data distribution (such that we can then use the model for $d\_{\gamma_t} \mathbf{\epsilon}_\theta(\mathbf{x}_t, t)$ in the ODE solver). This is exactly what we aim to achieve by the minimization in Eq. (15), where the $\mathbf{x}_t$ are sampled from the diffusion process. Consequently, the inputs $\mathbf{x}_t$ in Eq. (12) and Eq. (15) are coming from the same distribution and also the $d\_{\gamma_t} \mathbf{\epsilon}_\theta(\mathbf{x}_t, t)$ are the same in Eq. (12) and Eq. (15). Hence, we did not change the notation.
>
>     That being said, we made several small changes to the manuscript to avoid confusion. For instance, we changed the LHS of Eq. (12) from $d\_{\gamma_t} \mathbf{\epsilon}_\theta$ to $d\_{\gamma_t} \mathbf{\epsilon}_\theta(\mathbf{x}_t, t)$, we partially rewrote the relevant paragraph **Learning Objective** of Sec. 3.1, and we now directly use diffused data points $\alpha_t \mathbf{x}_0 + \sigma_t \epsilon$ in Eq. (15) to make explicit that this minimization objective is trained with diffused data. Furthermore, the added App G should provide additional insights as well.
>
>     We are happy to incorporate any further suggestions from the reviewer.
>
> - **"[...] why the NN approximation [in Eq.(8)] is better than other approximations in prior solvers? [...]"** Assuming infinite capacity of the neural network used for distillation and perfect optimization, we would have a perfect second Truncated Taylor Method (TTM), this is, after distillation we would recover the exact second-order terms with our neural network. This implies that we have a principled method that is exact in the limit of perfect optimization (also see Apps. G.1 and G.2). We outline the advantage of such a perfect second TTM to approximations (such as linear multistep methods) in the paragraph **Comparison to Multistep Methods** in Sec. 3. Furthermore, we added App. G.5, where we compare discretization errors of GENIE to prior (fast) solvers: as can be seen in Fig. 15, GENIE outperforms all other methods, including multistep methods. Multistep methods, such as [63], *always* approximate higher-order derivatives in the TTM using finite differences which are crude for large step sizes. Moreover, we added App. G.1 where we now derive and discuss theoretical error bounds for Truncated Taylor Methods in more detail, as well as App. G.2 where we lay out in detail why GENIE is a principled and theoretically-grounded approach.

---

> > ### Author Response · Authors · 2022-08-02
> > **Thank you for questions and feedback (2)**
> >
> > - **"I suggest to cite and discuss [Bao et al.] [...]"** We thank the reviewer for bringing this work to our attention.  Indeed, when learning their sampling covariance matrices the way Bao et al. parametrize their prediction head on top of a larger backbone network is similar to how in GENIE we use small prediction heads for distillation on top of the large first-order score predictor.
> >
> >     However, we would also like to stress multiple differences: (i) Bao et al. learn the diffusion model's sampling covariance matrices, while we learn higher-order ODE gradients. More generally, Bao et al. rely on stochastic diffusion model sampling, while we use the ODE formulation. (ii) In our case, we can resort to directly learning the low-dimensional Jacobian-vector products (JVPs) without low-rank or diagonal matrix approximations or other assumptions. Similar techniques are not easily applicable in Bao et al.'s setting and therefore they make potentially limiting assumptions in practice to keep the dimensionality of the additional network outputs manageable (discussed in more detail in App. G.5). In particular, they resort to assuming a diagonal covariance matrix in their experiments. By directly learning JVPs, we never have to rely on such assumptions. (iii) Experimentally, Bao et al. also consider fast sampling with few neural network calls. However, GENIE generally outperforms them (see, for example, their CIFAR10 results in their Table 2 for 10 and 25 NFE). This might indeed be due to the assumptions made by Bao et al., which we avoid. Furthermore, their stochastic vs. our deterministic sampling may play a role, too.
> >
> >     We gladly cite and discuss this work. For now, we added a section on this work in App. G.5, but we will move this into the main paper for the camera-ready version when there is more space available. Finally, we would like to point out that this is concurrent work (Bao et al.'s work was posted on ArXiv on June 15, 2022, after the NeurIPS submission deadline).
> >
> > If our reply is satisfactory and the reviewer's questions have been successfully addressed, we would like to kindly ask the reviewer to consider raising their score accordingly. Otherwise, we will be happy to further discuss.

---

### Official Review · Reviewer_kzgD · 2022-07-11

**Rating:** 6
**Confidence:** 3
**Soundness:** 3 good
**Presentation:** 3 good
**Contribution:** 2 fair

**Summary:**

This paper proposes a higher-order denoising diffusion solver called GENIE to speed up the sampling process for DPM.

In particular, the idea of the paper is inspired by the fact that DDIM is a first-order discretization of the probability ODE. Naturally, the paper attempts to use higher-order methods to solve the ODE with smaller errors in the same step or equivalently, obtain a good result with much fewer steps.

This paper considers the second-order method in particular. The resulting sampling process requires the second-order score functions under the hood, which is also natural.

The paper tackles the problem by distilling the terms related to the second-order score functions (approximated by the jacobian of a score network). Namely, learning neural networks to predict the jacobian vector product terms appeared in the sampling process.

The proposed method is compared to training-free methods under FID on several standard benchmarks.

**Questions:**

See weakness.

**Limitations:**

I did not see a direct negative societal impact.

**Strengths And Weaknesses:**

Strengths:

1. At a very high level, I agree that introducing higher-order methods into DPMs can be helpful.

2. The presentation is clear and the related work is properly discussed.

Weakness:

1. The way to deal with the second-order score functions is not that principled.

2. It would be much better if the authors could provide some theoretical benefits of the higher-order methods.

3. Likelihood results are missing. It is well known that the likelihood and FID are complementary metrics for generative modeling. It is necessary to evaluate all models under both metrics.

4. The baselines in Table 1 are mainly training-free methods. Why not include [69] in Table 1? [69] is more comparable to GENIE since it also trains additional models for distillation.

5. Why GENIE is not that good with NFE=5 in Table 1? I did not see an analysis in depth.

---

> ### Author Response · Authors · 2022-08-02
> **Thank you for questions and feedback (1)**
>
> We thank the reviewer for their thoughtful feedback. We are glad that the reviewer found the presentation of our paper to be clear and the related work to be well-discussed. We also appreciate that the reviewer agrees with the motivation for our proposed method. Below we address specific questions and comments.
>
> - **"The way to deal with the second-order score functions is not that principled."** We respectfully disagree with the reviewer in this point and would like to emphasize that GENIE is indeed a consistent and principled approach to developing a higher-order ODE solver for sampling from diffusion models.
>
>     GENIE's design consists of two parts: (1) We are building on the second Truncated Taylor Method (TTM), which is a well-studied ODE solver (see [59]) with provable local and global truncation errors (see also App. G.1). Therefore, if during inference we had access to the ground truth second-order ODE derivatives, which are required for the second TTM, GENIE would simply correspond to the *exact* second TTM.
>
>     (2) In principle, we could calculate the exact second-order derivatives *during inference* using automatic differentiation. However, this is too slow for competitive sampling speeds, as it requires additional backward passes through the first-order score network. Therefore, in practice, we use the learned prediction heads $\mathbf{k}_\psi(\mathbf{x}_t, t)$. Consequently, if $\mathbf{k}_\psi(\mathbf{x}_t, t)$ modeled the ground truth second-order derivatives exactly, i.e.  $\mathbf{k}_\psi(\mathbf{x}_t, t) = d\_{\gamma_t}{\mathbf{\epsilon}_\theta(\mathbf{x}_t, t)}$ for all $\mathbf{x}_t$ and $t$, we would obtain a rigorous second-order solver based on the TTM, following (1) above.
>
>     In practice, distillation will not be perfect. However, given the above analysis, optimizing a neural network $\mathbf{k}_\psi(\mathbf{x}_t, t)$ towards $d\_{\gamma_t}{\mathbf{\epsilon}_\theta(\mathbf{x}_t, t)}$ is well motivated and theoretically grounded. In particular, *during training* we are calculating *exact* ODE gradients using automatic differentiation on the first-order score model as distillation targets. Therefore, in the limit of infinite neural network capacity and perfect optimization, we could in theory minimize our distillation objective function Eq. (15) perfectly and obtain $\mathbf{k}_\psi(\mathbf{x}_t, t) = d\_{\gamma_t}{\mathbf{\epsilon}_\theta(\mathbf{x}_t, t)}$.
>
>     Also recall that regular denoising score matching itself, on which all diffusion models rely, follows the exact same argument. In particular, denoising score matching also minimizes a "simple" (weighted) $L_2$-loss between a trainable score model $\mathbf{s}_\theta(\mathbf{x}_t, t)$ and the spatial derivative of the log-perturbation kernel, i.e., $\nabla\_{\mathbf{x}_t} \log p_t(\mathbf{x}_t \mid \mathbf{x}_0)$. From this perspective, denoising score matching itself also simply tries to "distill" (spatial) derivatives into a model. If we perfectly optimized the denoising score matching objective, we would obtain a diffusion model that models the data distribution exactly, but in practice, similar to GENIE, we never achieve that due to imperfect optimization and finite-capacity neural networks. Nevertheless, denoising score matching similarly is a well-defined and principled method, precisely because of that theoretical limit in which the distribution can be reproduced exactly.
>
>     We would also like to point out that other, established higher-order methods for diffusion model sampling with the generative ODE, such as linear multistep methods [63], make approximations, too, which can be worse in fact. In particular, multistep methods *always* approximate higher-order derivatives in the TTM using finite differences which is crude for large step sizes, as can be seen in Fig. 3 (*bottom*). From this perspective, if our distillation is sufficiently accurate, GENIE can be expected to be more accurate than such multistep methods.
>
>     To clarify the reviewer's concern, we added this discussion in App. G.2, and we also slightly modified the paragraph **Learning Objective** in Sec. 3.

---

> > ### Author Response · Authors · 2022-08-02
> > **Thank you for questions and feedback (2)**
> >
> > - **"It would be much better if the authors could provide some theoretical benefits of the higher-order methods."** The benefits of higher-order methods are discussed in the paragraph **The Benefit of Higher-Order Methods** in Sec. 3. However, the reviewer is explicitly pointing towards a *theoretical* analysis. Mathematically, the benefit of higher-order methods can be immediately seen from the Taylor expansion (Eqs. (8) and (9)), from which GENIE (and other higher-order methods) is derived. Specifically, the error of Eqs. (8) and (9) with respect to the ground truth ODE solution is $\mathcal{O}(h_n^{p+1})$ for the general Eq. (8), and $\mathcal{O}(h_n^3)$ for the second Truncated Taylor Method in Eq. (9), respectively. Considering that $h_n$ is a small time step, this means that using higher-order numerical methods will always lead to a more accurate approximation with respect to the exact ODE solution. In App G.1, we added a brief derivation and discussion on this. Specifically, we wrote down the explicit expression for the error made by higher-order methods with respect to the exact solution. We hope that this unambiguously clarifies why higher-order methods like GENIE are preferable when aiming to efficiently solve ODEs like the generative Probability Flow or the DDIM ODE of diffusion models.
> >
> > - **"Likelihood results are missing. [...]"** Likelihood is usually used as a metric to quantify **model** performance; more specifically, likelihood quantifies the probability of held-out test data under the learnt **model**, and a good likelihood score indicates that the model can generate diverse data.  However, in this work, we are not proposing a new diffusion model. Instead, we are proposing a new fast sampling scheme for a given diffusion model. In fact, previous works that also propose novel solvers or samplers for given diffusion models generally never provide any likelihood results, as this is not a solver but a model property (for instance [62, 64, 63, 69, 67, 66]).
> >
> >     Likelihood in diffusion models is usually calculated by solving the probability flow ODE jointly with the instantaneous change of variable formula from the neural ODE literature [75]. To obtain a reliable likelihood result, one needs to integrate the entire ODE *trajectory* carefully as the likelihood result depends on the trajectory itself. To do so, one should only use solvers that provide certain guarantees about the accuracy of the result, for example, adaptive step size solvers such as Runge--Kutta 4(5) [84] (this solver is generally used to calculate likelihood for diffusion models [57] in their ODE formulation). In contrast, fast samplers and solvers, like GENIE and [62, 64, 63, 69, 67, 66], are concerned with the quality of the *final output* only, not the trajectory. In this situation, it is natural to synthesize this final output as efficiently as possible as the trajectory itself is not of any importance (hence, this is at odds with the ODE-based likelihood calculation).
> >
> >     All that being said, a related relevant question which the reviewer might be interested in would be: "What is the diversity of the final generated samples using our novel solver?". A common metric to quantify output diversity of generative models is the recall score as proposed in [104]. In particular, we follow DDGAN [67] and use the improved recall score [105]. Consequently, we calculated recall scores for GENIE, and other samplers that rely on the same score model, for CIFAR-10 (see App. G.3, and in particular Tab. 9). We see that for all methods recall scores suffer as the NFEs decrease, but this is expected. Compared to the baselines, GENIE achieves excellent recall scores, being on par with F-PNDM for NFE$\geq$15. However, F-PNDM cannot be run for NFE$\leq$10 (due to its additional Runge--Kutta warm-up iterations). Overall, these results confirm that GENIE offers strong sample diversity when compared to other common samplers using the same score model checkpoint.

---

> > > ### Author Response · Authors · 2022-08-02
> > > **Thank you for questions and feedback (3)**
> > >
> > > - **"Why not include [69] in Table 1?"** As stated in Sec 5.1, we only compare to methods that solve the generative ODE/SDE, including training-free and training-based methods, such as the Learned Sampler [66]. Progressive Distillation [69] abandons the underlying ODE formulation and simply aims for brute-force distillation. As a result, [69] cannot easily perform image encoding or use methods such as classifier(-free) guidance [57, 70] (in fact, it would have to distill an entirely new model for each guidance weight setting). However, these are highly important capabilities for practical applications like image editing and conditional image synthesis. Furthermore, Progressive Distillation [69] involves many distillation stages of large neural networks, whereas GENIE only involves a *single* distillation stage of a very small neural network (we only add a few trainable layers to the fixed first-order score model). This makes GENIE much more user-friendly. Most importantly, GENIE is versatile and can also be used for encoding samples into latent space and it also works together with classifier- and classifier-free guidance, as we demonstrated in Sec. 5.2.
> > >
> > >     That being said, we acknowledge that [69] is extremely effective when aiming purely at very fast high quality sample generation without other applications in mind. Consequently, we consider [69] an orthogonal approach to GENIE. In fact, we would like to stress that our method could potentially be combined with [69]: In every distillation stage of [69], we could quickly train a small GENIE prediction head to model higher-order ODE gradients. This would then allow for larger and/or more accurate steps, whose results represent the distillation target (teacher) in the progressive distillation protocol. This may also reduce the number of required distillation stages. Overall, this could potentially speed up the cumbersome stage-wise distillation and maybe also lead to an accuracy and performance improvement. In particular, we could replace the DDIM prediction in Algorithm 2 of [69] with improved GENIE predictions. Note that this approach would not be possible with multistep methods as proposed by Liu et al. [63]. Such techniques could not be used here, because they require the history of previous predictions, which are not available in the progressive distillation training scheme. We included a new paragraph **Combining GENIE with Progressive Distillation** in App G.4
> > >
> > > - **"Why GENIE is not that good with NFE=5 in Table 1?""** GENIE outperforms all methods *on the same checkpoint* for NFE$=$5 in Tab 1. The reviewer is correct that the Learned Sampler (LS) [66] outperforms GENIE for NFE$=$5 in Tab~1. (using a different checkpoint). As stated in Sec 5.1, we believe that the LS's advantage in this setup is potentially due to the different model checkpoint (which may have been trained longer or with more careful hyperparameter tuning), LS learning an optimal time step striding schedule (whereas GENIE simply uses quadratic striding during sampling), or due to LS learning additional degrees of freedom, such as sampling variances, within a more general class of DDMs. In particular, the striding schedule becomes very important for small NFEs. A more careful optimization of the striding schedule for GENIE would likely also further improve our results; however, we leave this to future work, as in this paper we are focusing on the GENIE algorithm itself. In that regard, for larger NFEs, where the time step striding becomes less crucial, we generally outperform LS.
> > >
> > > Lastly, we want to again emphasize that we compare to methods that solve the generative ODE/SDE, including training-based methods such as Learned Sampler [66], and not only training-free methods. **If our reply is satisfactory and the reviewer's concerns have been successfully addressed, we would like to kindly ask the reviewer to consider raising their score accordingly. Otherwise, we will be happy to further discuss.**

---

> > > > ### Comment · Reviewer_kzgD · 2022-08-04
> > > > **Futher questions**
> > > >
> > > > Thanks for the feedback. I have further questions.
> > > >
> > > > 1. In the final algorithm, is the target of $k_{\psi}$ the exact solution or a finite-difference approximation? According to my understanding, it is the latter one. If so, there is an additional error caused by FD, making the analogy to DSM improper. Is the FD error considered in the theoretical benefits of GENIE?
> > > >
> > > > 2. I understand the benefits of exact high-order methods for smaller errors (without considering the computation issues). My concern is that with an additional training procedure (with one more approximation error term than typical DSM) and potentially one more FD error term, is GENIE provably better than first-order methods under practical assumptions?
> > > >
> > > > 3. In NFE=5 (the most interesting setting in my opinion), GENIE is worse than LS and the authors argue that this is because of the different checkpoints. I cannot prove whether the point is correct or not. My observation is that the relative performance drop from NFE=10 to NFE=5 of GENIE  is much larger than  (about 5 times) that from NFE-15 to NFE-10. Can you eliminate the possibility that GENIE is not that stable with NFE=5 because of the additional error terms mentioned in 2?
> > > >
> > > > 4. The authors argue that [67] has multiple learnable networks and it is not fair to compare GENIE with [67]. Does a similar argument hold for GENIE and other learning-free methods?

---

> > > > > ### Author Response · Authors · 2022-08-05
> > > > > **Reply to further questions (1)**
> > > > >
> > > > > We thank the reviewer for their prompt reply.
> > > > >
> > > > > * (1.) The target of $\mathbf{k}_\psi$ *is not a finite-difference approximation*, but the *exact derivative* $d\_{\gamma_t} \epsilon_\theta$, which can be computed using automatic differentiation. This is discussed, for example, in Sec. 3.1 (see, for instance, the paragraph **Learning Objective**) as well as in App. G.2 (of the revised appendix). We would also like to point the reviewer to our training pseudocode in Algorithm 1, which we updated to point this out more clearly (see App. C.2.4). In our PyTorch implementation of GENIE the JVP terms arising in Algorithm 1 can simply be computed using *torch.autograd.functional.jvp (https://pytorch.org/docs/stable/generated/torch.autograd.functional.jvp.html). To avoid any misunderstandings, automatic differentiation simply calculates the analytic, exact derivative, using the chain rule.
> > > > >
> > > > >     Consequently, there is no finite-difference approximation error during training and the training targets during distillation are *exact*, which makes the comparison to the training procedure of denoising score matching (App. G.2) meaningful, from our perspective.
> > > > >
> > > > >     Importantly, note that we cannot use $d\_{\gamma_t} \epsilon_\theta$ computed via automatic differentiation during *inference* because it would result in a significant computation overhead. Thus, we only use the exact automatic differentiation-based $d\_{\gamma_t} \epsilon_\theta$ during *training* and at inference time we use the fast $\mathbf{k}_\psi$ network to predict $d\_{\gamma_t} \epsilon_\theta$.
> > > > >
> > > > >     In summary, we want to strongly emphasize that **GENIE, neither during training nor inference, uses any finite difference approximations**. If the reviewer pointed out what led to this misunderstanding, we would gladly modify the paper to make this more clear.

---

> > > > > > ### Author Response · Authors · 2022-08-05
> > > > > > **Reply to further questions (2)**
> > > > > >
> > > > > > - (2.) To start with, let us address the quote "[...] potentially one more FD error term": as discussed above, we do not make use of any finite difference approximations in our algorithm.
> > > > > >
> > > > > >     Next, as discussed in the paragraph **Learning Objective** in Sec. 3 and in App. G.2 (of the revised appendix), we show that GENIE is consistent and principled, that is, in the limit of infinite neural network capacity and perfect optimization, we could in theory minimize our distillation objective function Eq. (15) perfectly and obtain $\mathbf{k}_\psi = d\_{\gamma_t} \epsilon_\theta$. In this case, GENIE would simply correspond to the exact second truncated Taylor method (TTM), a well-studied ODE solver (see [59]) with provable local and global truncation errors (see also App. G.1), and is therefore provably better than a first-order method.
> > > > > >
> > > > > >     Now, the reviewer is asking whether "GENIE is provably better than first-order methods under practical assumptions". In practice, distillation will not be perfect (however, we believe our results indicate that distillation works sufficiently well) and therefore we unfortunately cannot prove any error bounds for GENIE, as it would involve proving generalization bounds of neural networks using non-convex optimization, a generally unsolved problem in the literature. That being said, we believe this to be of little importance in practice: Our results clearly indicate that GENIE (approximate second order method) is better than DDIM (first order method) and even other higher order methods (S-PNDM and F-PNDM). Furthermore, by definition, (truncation) error bounds of ODE solvers only become meaningful in the small step size regime in which fast diffusion samplers are not operating anyways. To make this point even more clear, note that F-PNDM is based on a fourth-order multistep method. That is, in the small step size limit, F-PNDM will mostly likely always outperform GENIE. **However**, our results clearly show that for larger step sizes (for example, NFEs={5, 10, 15, 20, 25}) which we use in practice, GENIE outperforms F-PNDM.
> > > > > >
> > > > > >     It might be interesting to discuss this further: if the distillation in GENIE was exact, then we would have access to *exact* second-order ODE derivatives *without any errors*. However, the 4-th order F-PNDM, even though it is a 4-th order method in contrast to GENIE being only of second-order method, *still approximates even the second-order derivative only up to 4-th order* (by fitting a 4th order polynomial using past function evaluations). Hence, if our distillation is sufficiently accurate, then our second-order derivative approximation is likely better than that of even higher-order methods that rely on polynomial or finite-difference approximations, and this can potentially make a big difference when operating with large steps in the small-NFE limit (and we indeed outperform F-PNDM---this indicates that our distillation is indeed quite accurate, which is not all that surprising given the strong representational power of modern neural networks). To avoid misunderstandings, we do not intend to make any claims here about the relevance of the second-order derivative in general, but this example shows that even though a method might be of higher-order than another one, it may still perform worse when working with large steps where the prefactors in the local truncation error $\mathcal{O}(h^{n+1})$ (for an n-th order solver) matter significantly. Hence, the typical theoretical error analyses done in the ODE literature are of little relevance in our practical few-NFE situation.

---

> > > > > > > ### Author Response · Authors · 2022-08-05
> > > > > > > **Reply to further questions (3)**
> > > > > > >
> > > > > > > - (3.) To clarify, we never stated that LS's advantage is solely due to the different checkpoint. As discussed in our initial rebuttal and in Sec. 5.1 of our paper, the advantage could also be due to LS learning an optimal time step striding schedule (whereas GENIE simply uses quadratic striding during sampling) or due to LS learning additional degrees of freedom, such as sampling variances, within a more general class of DDMs.
> > > > > > >
> > > > > > >     In fact, *GENIE does outperforms LS [66] for the same fixed quadratic striding schedule*. As can be seen in Table 3 of LS [66], their best result for a fixed quadratic striding schedule is FID=14.26 ("GGDM + PRED" without "TIME", which corresponds to the time stepping optimization), whereas we obtain FID=13.9. However, since we compare to **any** method that solves the generative ODE/SDE, we naturally included their very best result FID=12.4 (obtained with a *learned* striding schedule) in our Table 1. For our work, learning the striding schedule was out of scope.
> > > > > > >
> > > > > > >     However, following the your comment and to analyze this matter deeper, we now quickly ran a small grid search over striding schedules for GENIE and DDIM (the other two competitive baselines, S-PNDM and F-PNDM, rely on linear striding by construction, and therefore no grid search is applicable) on CIFAR-10. This simple grid search was able to improve GENIE's FID for NFE=5 from 13.9 to **11.2** and for NFE=10 from 5.97 to **5.28**. We discussed the details of the grid search in the new section G.7 in the Appendix. Furthermore, we added two new lines to our Table 1 and also included both LS results (with and without learned striding). We can now see that under fair comparisons (only quadratic striding, or only optimized striding) we outperform LS in all NFE settings (note that in the interest of providing a quick reply, we have not updated Fig. 5 yet, but will do so soon).
> > > > > > >
> > > > > > >     Regarding "My observation is that the relative performance drop from NFE=10 to NFE=5 of GENIE is much larger than (about 5 times) that from NFE-15 to NFE-10.", we would like to point out that going from NFE=15 to NFE=10, $1/3$ of all steps are eliminated, whereas going from NFE=10 to NFE=5, 1/2 of all steps are eliminated. Consequently, it is very much expected that the performance decreases more significantly when going from NFE=10 to NFE=5. In fact, the same holds for all relevant baselines (LS, DDIM, S-PNDM).
> > > > > > >
> > > > > > >     Lastly, we want to address the quote "Can you eliminate the possibility that GENIE is not that stable with NFE=5 because of the additional error terms mentioned in 2?". We hope that we could clear up all concerns regarding "2." above already (most importantly, there are no finite-difference approximation errors anywhere in GENIE). Furthermore, with the additional grid search, we now showed that the striding schedule indeed becomes very important for NFE=5, and our simple grid search already outperforms LS [6] (please see updated Table 1). In conclusion, from our perspective there is no reason anymore to believe that specifically for NFE=5 GENIE becomes generally unstable. Also note that, as discussed previously, if distillation was exact we had an exact second-order solver, and this analysis is independent of the step size taken and the number of steps.

---

> > > > > > > > ### Author Response · Authors · 2022-08-05
> > > > > > > > **Reply to further questions (4)**
> > > > > > > >
> > > > > > > > - (4.) Given the previous discussion about [69], we believe that the reviewer is referring to Progressive Distillation (PG) [69] instead of DDGAN [67]? In the following, we make this assumption, but we kindly ask the reviewer to correct us if they actually meant DDGAN [67].
> > > > > > > >
> > > > > > > >     As stated in our inital rebuttal and in the paper, our criteria for comparison is not how much training a method involves but rather **whether or not the resulting method solves the generative ODE/SDE**. As stated in Sec. 4 and Sec. 5, being able to solve the generative ODE/SDE directly allows for applications such as classifier(-free) guidance [57, 70] and image encoding. These techniques play an important role in synthesizing photo-realistic images from DDMs [3, 4, 15, 17], as well as for image editing tasks [12, 17]. Therefore, we believe that practically relevant samplers should ideally be derived in such way that they allow for these applications, which was our goal with GENIE (which we validate and demonstrate extensively in Sec. 5.2).
> > > > > > > >
> > > > > > > >     In GENIE, we maintain the rigorous generative ODE/SDE framework, because *we only distill the second-order derivative*, and using that we can construct rigorous ODE solvers (using the second Truncated Taylor Method in our case). In PG, in contrast, *the entire sampling procedure* is approximated and all rigorous ODE concepts are abandoned.
> > > > > > > >
> > > > > > > >     We are convinced that PG [69] is currently the state-of-the-art method when one is interested in brute-force distillation and fast sampling at all costs. However, despite GENIE also involving (although considerably less) training, once training is finished we can directly use it for any of the above mentioned applications: for example, classifier-free guided sampling with any guidance scale $w$. On the other hand, consider we had brute-force distilled the diffusion model for a specific guidance scale $w_1$ using PG with training of many networks. If we now ever decided that we would actually prefer to use another guidance scale $w_2$, our initial model trained for $w_1$ is useless and we would have to go through the same, very computationally costly, procedure again. For GENIE, changing the guidance scale is as easy as specifying another number in the sampling script. We truly believe that this is a highly relevant aspect. Ultimately, the fast samplers that are developed by the community should ideally find their way into practical applications. And in essentially all large-scale diffusion models, guidance techniques are being used nowadays.
> > > > > > > >
> > > > > > > > **If our reply is satisfactory and the reviewer’s concerns have been successfully addressed, we would like to kindly ask the reviewer to consider raising their score accordingly. Otherwise, we will be happy to further discuss.**

---

> > > > > ### Comment · Reviewer_kzgD · 2022-08-06
> > > > > **Thanks for clarification.**
> > > > >
> > > > > Thanks for the response, which clarifies my misunderstanding. I'm happy to increase my rating accordingly.

---

### Official Review · Reviewer_6WXQ · 2022-07-14

**Rating:** 10
**Confidence:** 4
**Soundness:** 4 excellent
**Presentation:** 4 excellent
**Contribution:** 4 excellent

**Summary:**

The paper proposes a novel higher-order denoising diffusion solver, called GENIE, which is applicable particularly for denoising diffusion models with the variational preserving forward noising process. The GENIE solver is established from two novel approaches proposed in the paper.

First, the paper identifies that p-th order Taylor approximation (truncated Taylor Method, TTM) to DDIM ODE results in much simpler terms than the TTM of the equivalent probability flow ODE.
- Note that an equivalent probability flow ODE of a denoising SDE, aka denoising diffusions, is an ordinary differential equation whose marginals equal (in distribution) to the denoising SDE's marginals at every time step. Moreover, DDIM ODE is a scaled stochastic process (time-inhomogeneous manner) of the equivalent probability flow ODE.

More specifically, the authors show that the 2nd-order term in the DDIM ODE's Taylor approximation reduces to two Jacobian-vector products (JVP) plus one time-derivative; One JVP is a product of the Jacobian of the score function (wrt input) and the score function, and the Jacobian and input.

Second, instead of using the finite difference approximation of the Jacobian (of the score wrt input) during inference, it proposes to distill the JVPs (or a sum of two) by another network. The training of the JVP network is via minimizing the mean squared error to the finite difference approximation.
- Note that JVP will have the same dimensionality of the data. This is much lower than some previous approaches to learning higher order scores, which is $d^2$ for $d$-dimensional data.
To reduce the complexity of learning additional networks, the paper proposes a model both predicts score and JVP by sharing feature extractors.

In addition, the paper further introduces higher-order (>2) GENIE solvers.

For the experiments, the paper demonstrates the effectiveness of GENIE on multiple image generation benchmarks.

**Questions:**

N/A

**Strengths And Weaknesses:**

**Strength**

In my understanding, the paper's contributions are clear, and I also consider that the results are essential for several reasons:

1. The paper proposes a novel higher-order denoising diffusion solver applicable to pre-trained score-based generative models.
2. The paper motivates each part of the proposed method well so that readers can understand how each step contributes to the merits of GENIE.
3. I found that the paper has a well-organized structure that makes it clear to understand the proposed method. Significantly, the paper shares sufficient information to understand the new solver's behaviors, which will help readers understand relevant backgrounds and the proposed method.
    - For example, the first-order method to DDIM ODE is equivalent to the second-order with the "ideal derivative trick". Therefore, the first-order method to DDIM ODE performs better than the first-order method to the original probability flow ODE.

In general, the paper's contributions wrt the novelty are clear, and the proposed methods are well-defined. In addition, I found that the paper has a well-organized structure that makes it clear to understand the proposed methods. Thus, I'm inclined to accept the paper.

---

> ### Author Response · Authors · 2022-08-02
> **Thank you for questions and feedback**
>
> We would like to thank the reviewer for their thoughtful and positive feedback. We are glad that our contributions came across clearly and that we could successfully communicate their motivations, significance and novelty.
>
> In particular, we are delighted that we could convey the importance of learning $d$-dimensional Jacobian-vector products (JVPs), rather than $d^2$-dimensional higher order score functions, which we believe is one of the main contributions of our work. We are also glad that the reviewer highlights our side-analysis (App.~B.1) where we show that the DDIM solver (Euler's method applied to the DDIM ODE, a reparameterization of the Probablity Flow ODE) can be interpreted as an approximate second-order ODE solver (using the "ideal derivative trick" [61]); we believe that this interpretation partially explains why the DDIM solver has been so successful for synthesis in diffusion models. Lastly, we would like to clarify that the GENIE prediction heads are trained to predict the *exact* JVPs defined by the (first-order) score model and that, during training, we compute these exact JVPs via differentiation/backpropagation through the (first-order) score model. We do not rely on any finite difference approximations, neither during training nor during inference.

---

### Author Response · Authors · 2022-08-02
**Comment to all Reviewers**

We thank all reviewers for engaging in the review process. In our rebuttal replies, we attempted to address specific questions and comments as clearly and detailed as possible. In the main paper, apart from minor edits we only modified the paragraph **Learning Objective** in Sec. 3 more significantly. All other additions to the paper are currently placed in App. G due to space limitations. If the paper will be accepted, and another page will be made available for the main text, we will re-organize the content and potentially bring some of the context from the appendix into the main paper.

---

### Meta-Review · Area_Chair_uXmv · 2022-08-26

**Recommendation:** Accept
**Confidence:** Certain

**Metareview:**

There is overall consensus that the paper has significant contributions, good experimental validation and a clear presentation. The authors have addressed the majority of concerns raised by the reviewers during the author-reviewer discussions, leading to several scores being raised. I would like to thank the authors and reviewers for actively engaging in discussions. The recommendation is to accept this paper.

**Award:**

No

---

### Decision · Program_Chairs · 2022-09-14

Accept